# Tracking the Discriminative Axis: Dual Prototypes for Test-Time OOD Detection Under Coveriate Shift

## Abstract

For reliable deployment of deep-learning systems, out-of-distribution (OOD) detection is indispensable. In the real world, where test-time inputs often arrive as streaming mixtures of in-distribution (ID) and OOD samples under evolving covariate shifts, OOD samples are domain-constrained and bounded by the environment, and both ID and OOD are jointly affected by the same covariate factors. Existing methods typically assume a stationary ID distribution, but this assumption breaks down in such settings, leading to severe performance degradation. We empirically discover that, even under covariate shift, covariate-shifted ID (csID) and OOD (csOOD) samples remain separable along a discriminative axis in feature space. Building on this observation, we propose *DART*, a test-time, online OOD detection method that dynamically tracks dual prototypes—one for ID and the other for OOD—to recover the drifting discriminative axis, augmented with multi-layer fusion and flip correction for robustness. Extensive experiments on a wide range of challenging benchmarks, where all datasets are subjected to 15 common corruption types at severity level 5, demonstrate that our method significantly improves performance, yielding 15.32 pp AUROC gain and 49.15 pp FPR@95TPR reduction on ImageNet-C vs. iNaturalist-C compared to established baselines. These results highlight the potential of the test-time discriminative axis tracking for dependable OOD detection in dynamically changing environments.

## 1 Introduction

Deep neural networks (DNNs) achieve remarkable performance across applications such as image classification, object detection, medical imaging, autonomous driving, and speech recognition (Alam et al., 2020). These successes stem from large-scale datasets, high-performance hardware, and innovative model architectures (Deng et al., 2009; Krizhevsky et al., 2012; He et al., 2016; Vaswani et al., 2017), motivating deployment in real-world systems.

In practice, however, deployed models inevitably encounter test inputs that deviate from their training distributions. One form is **semantic shift**, where models face unknown semantics—commonly termed out-of-distribution (OOD) samples. Substantial progress has been made on OOD detection: existing methods typically assume either abstract characteristics (Hendrycks & Gimpel, 2016; Liu et al., 2020; Xu et al., 2023) or data-specific characteristics (Lee et al., 2018; Sun et al., 2022) of in-distribution (ID) data to distinguish ID from OOD. A second form is **covariate shift**, where data appears under new conditions such as changes in weather, illumination, or sensor noise (Moreno-Torres et al., 2012; Dockès et al., 2021). Most OOD methods implicitly assume stationary ID distributions as reference to separate ID and OOD, however, as shown in Figure 2, in practice they struggle under covariate shifts (Yang et al., 2024; 2021; 2023a) because shifting covariates alter the space geometry that their decision rules rely on.

We study **test-time OOD detection under covariate shift** in a realistic *streaming mixture* setting: test-time inputs arrive in mini-batches as mixtures of ID and OOD samples, and both are simultaneously exposed to the *same evolving covariate shifts* (e.g., a change in illumination). We denote these as **covariate-shifted ID (csID)** and **covariate-shifted OOD (csOOD)**. Within each mini-batch—as illustrated in Figure 1—spatial and temporal coherence arises from the task environment, so

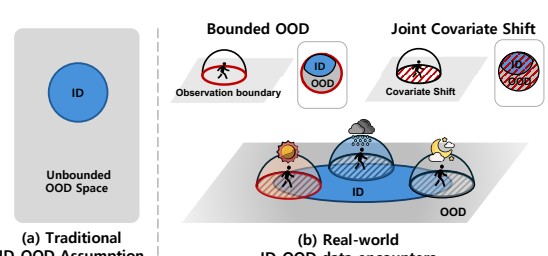

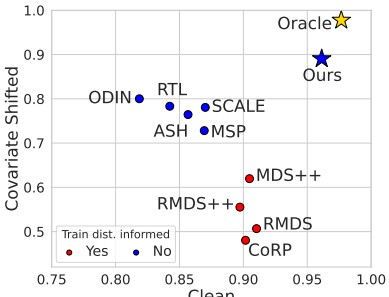

Figure 1: Comparison of traditional and real-world ID-OOD assumptions. (a) Traditional OOD detection assumes ID data (blue circle) exists within an unbounded OOD space (gray background). (b) In real-world scenarios, OOD data is bounded by physical and environmental constraints (observation boundary, top-left inset), limiting the space where OOD samples can occur. Furthermore, covariate shifts such as weather conditions can simultaneously affect both ID and OOD distributions (dashed regions), causing them to shift jointly in feature space.

Figure 2: AUROC comparison on both covariate shifted and clean ImageNet-based benchmark. Existing methods suffer under covariate shift, with train distribution–informed approaches dropping to around 0.5. In contrast, the oracle axis achieves consistently high performance regardless of shift, and our method effectively discovers this axis, attaining near-oracle results.

OOD samples are *domain-constrained* rather than arbitrary. For instance, in autonomous driving, encountering an unseen vehicle type is plausible OOD, whereas suddenly observing medical or satellite images is essentially impossible. Moreover, temporally correlated csID and csOOD typically undergo the *same* covariate shift, so their distributions co-evolve during deployment. In our setting, the test stream is unlabeled, the backbone is frozen, no training data are accessed at test time, and the algorithm maintains a small, bounded state.

In this practical scenario, we empirically observe a key insight that enables our approach. Across diverse datasets and shifts, we consistently observe: (i) *local coherence*—within short windows, csOOD samples organize into coherent groupings in representation space; and (ii) a *recoverable linear axis*—csID and csOOD remain approximately linearly separable along a dominant discriminative direction that *drifts* as covariates evolve. As shown in Figure 2 with annotation "Oracle", computing our method's OOD score with the optimal discriminative axis yields very high AUROC, demonstrating that separability in this direction can lead to strong detection performance. These observations suggest focusing on *tracking* the separation direction online rather than relying on a fixed, training-time score.

We propose **D**iscriminative **A**xis **R**eal-time **T**racker (*DART*), a **test-time**, **online OOD detection** method that continuously tracks a *discriminative axis* using a class-agnostic ID prototype and an OOD prototype *per feature layer*. At each step, incoming test samples update the prototypes via lightweight, stable rules, yielding the vector connecting them as the discriminative axis. Each sample is then scored by its relative position to this axis, producing a simple forward-pass detector aligned with the evolving feature space. To address the fact that covariate shifts affect DNN layers differently (Hendrycks & Dietterich, 2019; Yin et al., 2019), *DART* employs **multi-layer score fusion** to stabilize detection across heterogeneous, unpredictable shifts. The method requires only forward passes, no model weight update, no labels, and no access to training data at test time, which suits privacy-sensitive or on-device deployments where retraining is infeasible. For robustness, prototypes are initialized conservatively and updated with safeguards to prevent collapse.

Across challenging benchmarks, *DART* consistently delivers substantial gains over prior approaches. For example, on the ImageNet benchmark, *DART* achieves at least **9.04 percentage points (pp)** higher AUROC under covariate shift and **5.10pp** AUROC improvement on the clean setting, as shown in Figure 2, while attaining at least **40.15pp** and **19.06pp** FPR@95TPR reduction on the covariate shifted and clean datasets, respectively. Remarkably, the performance of *DART* comes close to that of the Oracle, highlighting the effectiveness of our approach.

Our contributions can be summarized as below:

- We formalize *test-time OOD detection under covariate shift* in a streaming mixture setting, distinguishing csID from csOOD and articulating realistic constraints (data stream, frozen backbone, small memory).

- We introduce **DART**, which *tracks dual prototypes online* to recover the drifting discriminative axis and fuses *multi-layer* scores for robustness to layer-specific covariate effects.

- We provide measurements and visualizations showing coherent csOOD groupings and approximately linear csID–csOOD separation within short test windows.

- We demonstrate consistent gains over strong post-hoc baselines on joint-shift suites, with large improvements in AUROC and FPR@95, using only forward passes and no retraining.

## 2 RELATED WORK

### 2.1 OUT-OF-DISTRIBUTION (OOD) DETECTION: TRAINING-DRIVEN VS. POST-HOC

Research on OOD detection can be broadly categorized into learning-based and post-hoc approaches.

**Training-driven approaches.** These methods modify training to enhance OOD separability, e.g., Outlier Exposure (OE) with auxiliary outliers (Hendrycks et al., 2018; Zhang et al., 2023a; Zhu et al., 2023) and $N+1$ classifiers that add an "unknown" class (Bendale & Boult, 2016; Shu et al., 2017; Chen et al., 2021). While effective, they require additional data or altered objectives and may misalign with deployment OODs, with potential side effects on ID accuracy. Some methods (Katz-Samuels et al., 2022; Yang et al., 2023b) update model parameters at test time via backpropagation, which introduces latency and can compromise ID accuracy under non-stationary streams.

**Post-hoc (training-free) approaches.** These operate on a frozen classifier without retraining and have gained widespread adoption due to their ease of use and compatibility with pretrained models. Categories include: output-based scoring (Hendrycks & Gimpel, 2016; Hendrycks et al., 2019a; Liu et al., 2020), distance-based methods (Lee et al., 2018; Ren et al., 2021; Mueller & Hein, 2025; Sun et al., 2022; Park et al., 2023), feature-based approaches (Liang et al., 2017; Wang et al., 2022; Sun et al., 2021; Djurisic et al., 2022; Xu et al., 2023; Zhang et al., 2022), and gradient-based methods (Huang et al., 2021; Behpour et al., 2023). Furthermore, *training distribution-informed* methods (e.g., Mahalanobis, ViM and KNN) *assume access to training statistics* (feature means, covariances, principal subspaces)—assumptions that can become invalid under test-time covariate drift and are often infeasible when training data are unavailable. In contrast, our method is post-hoc and relies solely on the unlabeled test stream, without training statistics.

**Post-hoc, test-time adaptive approaches.** An emerging line of work adapts OOD detection using test-time batches/streams *without* weight updates. RTL (Fan et al., 2024) uncovers a linear trend between OOD scores and features and fits a batch-level discriminator; OODD (Yang et al., 2025) maintains an online dynamic OOD dictionary to accumulate representative OOD features. These approaches are closer in spirit to our online setting but typically operate without explicitly addressing the time-varying covariate shift. *DART* differs by *tracking* the discriminative axis with *dual prototypes* (ID/OOD) *per layer* online across a stream; we maintain persistent, memory-light state that adapts smoothly to drift. Our goal is OOD detection under covariate drift in streaming mixtures; conventional test-time adaptation (TTA) methods that target closed-set robustness and/or adapt model weights (Wang et al., 2020; Niu et al., 2023) are orthogonal to this objective and not our focus.

### 2.2 COVARIATE SHIFT AND JOIN-SHIFT EVALUATION

**Covariate corruptions and natural shift.** Covariate shift—changes in input distributions with fixed labels—is commonly studied with corruption suites such as CIFAR-C and ImageNet-C (Hendrycks & Dietterich, 2019), which introduce noise, blur, weather, and other factors. Recent datasets emphasize natural sources of shift from environment and sensor variation (Baek et al., 2024; 2025).

**Joint semantic and covariate shift.** Full-spectrum OOD (FS-OOD) (Yang et al., 2023a)) evaluates semantic OOD while allowing covariate variation; OpenOOD unifies large-scale OOD evaluation and includes *joint-shift* settings (Yang et al., 2022; Zhang et al., 2023b). Dataset design such as NINCO (Bitterwolf et al., 2023) reduces ID contamination for clearer semantic separation. Our setting follows this trajectory but explicitly considers *streaming mixtures* where csID and csOOD experience the *same evolving covariates* (e.g., a sudden illumination change). *DART* is designed to adapt online in such scenarios via per-layer dual prototypes and multi-layer score fusion, without requiring training data, training statistics, or weight updates.

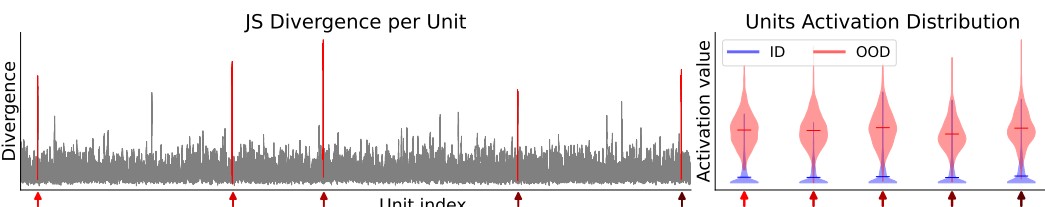

Figure 3: Unit-wise activation analysis. The left panel shows the JS divergence between ID and OOD activations, with arrows marking units of large divergence. The right panel visualizes the activation distributions of these units, where ID (blue) and OOD (red) are clearly separable.

## 3 METHOD

In this section, we introduce our method *DART* for online test-time OOD detection under covariate shift. We begin by revisiting a key motivation behind our approach: the empirical emergence of discriminative axis in pre-trained feature spaces. We then describe how the prototypes that define this axis are iteratively refined with incoming test batches. Finally, we explain why multi-layer fusion is essential to maintain robustness across unpredictable types of covariate shift.

### 3.1 FORMULATION SETUP

We first explain the notations for our problem. We define $\mathcal{D}_{ID}$ and $\mathcal{D}_{OOD}$ as the dataset for ID and OOD, observed by the OOD detection system. Then, let $\mathcal{B}_t = \{\mathbf{x}_{t,1}, \mathbf{x}_{t,2}, \ldots, \mathbf{x}_{t,N}\}$ denote an input batch received at test time, where each sample $\mathbf{x}_{t,i}$ may belong to one of two categories under covariate shift: covariate-shifted in-distribution (csID) or covariate-shifted out-of-distribution (csOOD). We denote the subset of csID samples as $\mathcal{B}_t^{\text{ID}}$ and the csOOD samples as $\mathcal{B}_t^{\text{OOD}}$, such that $\mathcal{B}_t = \mathcal{B}_t^{\text{ID}} \cup \mathcal{B}_t^{\text{OOD}}$.

Model is composed of multiple layers, and we extract intermediate feature representations from several of them. Let $f_l(\cdot)$ denote the feature mapping at layer $l$, where $l \in \mathcal{L} = \{1, 2, \ldots, L\}$. For a given input $\mathbf{x}$, we obtain a set of multi-layer features $\{\mathbf{z}^{(1)}, \mathbf{z}^{(2)}, \ldots, \mathbf{z}^{(L)}\}$, where $\mathbf{z}^{(l)} = f_l(\mathbf{x})$ represents the feature at layer $l$. Thus, for a csID sample $\mathbf{x}_t^{\text{ID}} \in \mathcal{B}_t^{\text{ID}}$ and a csOOD sample $\mathbf{x}_t^{\text{OOD}} \in \mathcal{B}_t^{\text{OOD}}$, their multi-layer feature sets are $\mathbf{Z}_t^{\text{ID}} = \{\mathbf{z}_t^{(1),\text{ID}}, \ldots, \mathbf{z}_t^{(L),\text{ID}}\}$ and $\mathbf{Z}_t^{\text{OOD}} = \{\mathbf{z}_t^{(1),\text{OOD}}, \ldots, \mathbf{z}_t^{(L),\text{OOD}}\}$, respectively.

### 3.2 ID–OOD SEPARABILITY IN FEATURE SPACE: EXISTENCE OF THE DISCRIMINATIVE AXIS

Prior works (Sun et al., 2021; Xu et al., 2023) have reported that ID and OOD samples exhibit distinct activation patterns in the feature space. In a similar spirit, we systematically examine unit-level activations from a distributional perspective. Specifically, we collect unit-wise activation distributions across multiple ID and OOD samples and compare them. Our analysis reveals that there exists certain units where distributions of ID and OOD samples diverge substantially, as evidenced by a large Jensen–Shannon (JS) divergence in Figure 3. Visualization via violin plots further demonstrates that ID and OOD activations can be sharply distinguished within those units.

Building upon this insight, we leverage these distributional differences to construct a unified discriminative direction. We compute prototype representations by averaging activations across all ID samples and all OOD samples respectively, yielding two representative points in the feature space[1]: $\mathbf{p}_{\text{ID}}$ and $\mathbf{p}_{\text{OOD}}$:

$$\mathbf{p}_{\text{ID}} = \frac{1}{|\mathcal{D}_{\text{ID}}|} \sum_{\mathbf{x} \in \mathcal{D}_{\text{ID}}} f(\mathbf{x}), \quad \mathbf{p}_{\text{OOD}} = \frac{1}{|\mathcal{D}_{\text{OOD}}|} \sum_{\mathbf{x} \in \mathcal{D}_{\text{OOD}}} f(\mathbf{x}) \tag{1}$$

Then, we define the connecting vector between these prototypes as the **discriminative axis**:

$$\mathbf{axis}_{\text{disc}}^{oracle} = \mathbf{p}_{\text{ID}} - \mathbf{p}_{\text{OOD}} \tag{2}$$

The key insight of this averaging operation is that it naturally implements an automatic weighting mechanism: units with substantial ID-OOD divergence are emphasized in the discriminative axis,

---

[1]While separability analysis may be conducted for each of the selected feature layers, but for clarity, we omit the layer index from the notation in this section.

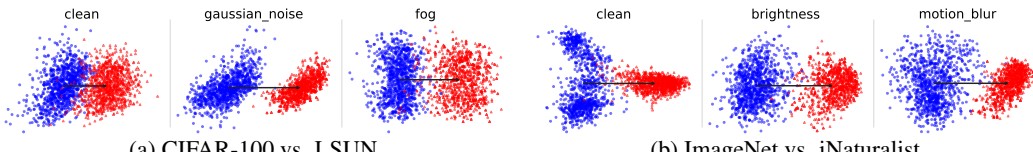

(a) CIFAR-100 vs. LSUN          (b) ImageNet vs. iNaturalist

Figure 4: Distribution of ID (blue dots) and OOD (red dots) samples at features space projected with the oracle discriminative axis as the horizontal axis.

while units with trivial divergence are suppressed. This occurs because discriminative units exhibit large differences between their ID and OOD mean activations, whereas non-discriminative units show similar mean values across both distributions. Figure 4 shows that when features are projected along this axis, ID and OOD samples consistently form distinct clusters. This separation persists regardless of the presence or type of covariate shift, indicating the existence of discriminative axis.

However, such a discriminative axis is a theoretical construct that presupposes knowledge of OOD distributions. Since the nature of OOD is inherently unknown before test-time, it is not feasible to predefine and fix such a discriminative axis in advance. This motivates the need to adaptively identify the optimal discriminative axis during test-time. To this end, we propose a method that *progressively identifies two prototypes*—one associated with ID samples and the other with OOD samples—whose connecting direction defines a discriminative axis that adapts to the evolving data stream.

### 3.3 BATCH-WISE PROTOTYPE REFINEMENT: TRACKING THE DISCRIMINATIVE AXIS

To craft and update the discriminative axis in an online manner, we refine prototypes through iterative pseudo-labeling and prototype updating. Refer to Appendix Sec. B.4 for the detailed algorithm.

Our method initializes and dynamically updates layer-specific prototypes for csID and csOOD based on current test batch features. Since true ID/OOD labels are unavailable during test-time, we rely on pseudo-labeling to distinguish between csID and csOOD samples. We employ Otsu algorithm (Otsu et al., 1975) to automatically determine optimal thresholds by *maximizing the between-class variance*, providing a principled way to separate samples based on their score distributions.

**Dual-Prototype Initialization.** For the initial batch, we use naive baseline score, Maximum Softmax Probability (MSP) as our reference score. We assign pseudo-labels using MSP with the Otsu-determined threshold, then compute initial prototypes as the mean feature vectors of their respective pseudo-labeled groups, i.e. $\bar{\mathbf{p}}_1^{\text{ID}} = \frac{1}{|\mathcal{S}_1^{\text{ID}}|} \sum_{i \in \mathcal{S}_1^{\text{ID}}} f(\mathbf{x}_{1,i})$, $\bar{\mathbf{p}}_1^{\text{OOD}} = \frac{1}{|\mathcal{S}_1^{\text{OOD}}|} \sum_{i \in \mathcal{S}_1^{\text{OOD}}} f(\mathbf{x}_{1,i})$.

**Dual-Prototype Tracking.** For subsequent batches, we design a more refined scoring function which better utilizes the built prototypes. We compute Euclidean distances between each sample and the dual prototypes from the previous timestep, then calculate a Relative Distance Score (RDS) that reflects each sample's position relative to both prototypes:

$$\text{RDS}(\mathbf{x}_{t,i}) = 1 - \frac{\|\mathbf{z}_{t,i} - \bar{\mathbf{p}}_{t-1}^{\text{ID}}\|_2}{\|\mathbf{z}_{t,i} - \bar{\mathbf{p}}_{t-1}^{\text{ID}}\|_2 + \|\mathbf{z}_{t,i} - \bar{\mathbf{p}}_{t-1}^{\text{OOD}}\|_2}, \quad (3)$$

where $\mathbf{z}_{t,i}$ denotes the feature of $i$-th sample in the batch $t$. The RDS formulation is inherently scale-invariant, making it robust to variations in feature magnitudes across different layers and architectures. Using the Otsu algorithm, we determine an optimal threshold to assign pseudo-labels based on this RDS score. We compute the new prototypes $\hat{\mathbf{p}}_t^{\text{ID}}, \hat{\mathbf{p}}_t^{\text{OOD}}$ as the mean feature vectors of their respective pseudo-labeled groups. To ensure prototype reliability, we incorporate Tukey's *outlier filtering* method (Tukey et al., 1977) to exclude samples that are too far from their assigned prototypes so that only the remaining samples contribute to the mean computation. Finally, we refine prototypes using *exponential moving average (EMA)* to maintain stability:

$$\bar{\mathbf{p}}_t^{\text{ID}} = \alpha \, \bar{\mathbf{p}}_{t-1}^{\text{ID}} + (1 - \alpha) \, \hat{\mathbf{p}}_t^{\text{ID}}, \quad \bar{\mathbf{p}}_t^{\text{OOD}} = \alpha \, \bar{\mathbf{p}}_{t-1}^{\text{OOD}} + (1 - \alpha) \, \hat{\mathbf{p}}_t^{\text{OOD}}. \quad (4)$$

This iterative process enables prototypes to progressively converge toward the true underlying distributions of csID and csOOD samples.

**Flip Correction.** Due to *DART*'s pseudo-labeling approach, incorrectly initialized prototypes can lead to catastrophic misplacement, with prototypes potentially drifting toward opposite sides of

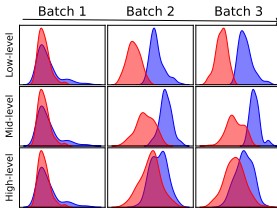
(a) RDS under Gaussian noise

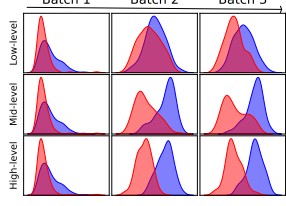
(b) RDS under defocus blur

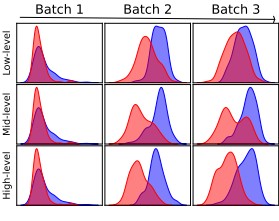
(c) RDS under jpeg comp.

Figure 5: Layer-wise RDS distributions across three covariate shift types. Each plot shows the RDS distribution of **csID** (**blue curve**, CIFAR-100) and **csOOD** (**red curve**, LSUN) samples at different network depths (low, mid, high-level) through three sequential batches. The visualizations reveal how different corruption types affect feature separability at specific network layers; under Gaussian noise, separability degrades in high-level layers, whereas under defocus blur, it degrades in low-level layers.

their desirable locations. To address this, we implement a "flip" detection mechanism that identifies prototype misalignments and automatically swaps them when necessary. We detect flips by comparing current prototypes with an auxiliary MSP-based prototype. A flip occurs when the csID prototype is significantly farther from the MSP-based reference than the csOOD prototype, while simultaneously showing lower cosine similarity. Formally, we swap prototypes when:

$$\|\bar{\mathbf{p}}_t^{\text{ID}} - \hat{\mathbf{p}}_{t,\text{MSP}}^{\text{ID}}\|_2 > 2\|\bar{\mathbf{p}}_t^{\text{OOD}} - \hat{\mathbf{p}}_{t,\text{MSP}}^{\text{ID}}\|_2 \quad \text{and} \quad \cos\left(\bar{\mathbf{p}}_t^{\text{ID}}, \hat{\mathbf{p}}_{t,\text{MSP}}^{\text{ID}}\right) < \cos\left(\bar{\mathbf{p}}_t^{\text{OOD}}, \hat{\mathbf{p}}_{t,\text{MSP}}^{\text{ID}}\right)$$

We use a weighted comparison (factor of 2) to impose a strict condition that prevents unintended flip detections, and a value we found works well across all datasets.

### 3.4 MULTI-LAYER SCORE FUSION

To enhance discriminative axis identification, we extend our approach to multi-layer features. Low-level features capture local patterns like textures and edges, while high-level features encode semantic concepts (Guo et al., 2016). However, covariate shifts can selectively disrupt different levels of visual information (Hendrycks & Dietterich, 2019; Yin et al., 2019)—blur corruptions primarily affect low-level features, while elastic transformations impair higher-level representations. As a result, different layers exhibit varying degrees of ID/OOD discriminability depending on the shift type, as illustrated in Figure 5. Since the nature of covariate shift is typically *unknown beforehand*, leveraging information from all feature levels through multi-layer fusion is essential for robust OOD detection.

As the prototypes are updated for each batch, we compute the $\text{RDS}_l$ for each selected layer $l$. To obtain the final OOD score for each sample $\mathbf{x}_{t,i}$ in the batch, we fuse the RDS values from the selected layers $\mathcal{L}$ by taking their average, formally given by $\text{RDS}_{\text{final}}(\mathbf{x}_{t,i}) = \frac{1}{|\mathcal{L}|} \sum_{l \in \mathcal{L}} \text{RDS}_l(\mathbf{x}_{t,i})$. Using the fused OOD score $\text{RDS}_{\text{final}}(\mathbf{x}_{t,i})$, we make the final OOD prediction.

## 4 EXPERIMENTS

### 4.1 EXPERIMENTAL SETUP

**Datasets.** Although *DART* is designed to adaptively handle challenging covariate shifts, we evaluate on both clean and covariate-shifted datasets as the shift may be weak or even absent in real-world scenarios. We use CIFAR-100 (Krizhevsky et al., 2009) and ImageNet (Deng et al., 2009) as ID datasets. For CIFAR-100, we use SVHN (Netzer et al., 2011), Places365 (Zhou et al., 2017), LSUN (Yu et al., 2015), iSUN (Xu et al., 2015), and Textures (Cimpoi et al., 2014) as OOD datasets. For ImageNet, we use ImageNet-O (Hendrycks et al., 2021), Places (Zhou et al., 2017), SUN (Xiao et al., 2010), iNaturalist (Van Horn et al., 2018), and Textures as OOD datasets. To simulate covariate shift, we apply 15 common corruption types (Hendrycks & Dietterich, 2019) at severity level 5 to both ID and OOD datasets, resulting in pairs such as CIFAR-100-C vs. SVHN-C.

**Models.** For CIFAR-100-based benchmarks, we use WideResNet-40-2 (Zagoruyko & Komodakis, 2016) pre-trained with AugMix (Hendrycks et al., 2019b) on clean CIFAR-100, which is available from RobustBench(Croce et al., 2020). For ImageNet-based benchmarks, we use the pre-trained RegNetY-16GF (Radosavovic et al., 2020) available from PyTorch. In addition, we also evaluate with

Table 1: OOD detection performance comparison with CIFAR-100-C and ImageNet-C csID. Results are the average of all 15 corruptions with severity level 5. (Best: **bolded**, Second-best: underlined)

| | | ImageNet-C | | | | | | | | | | | |
|---|---|---|---|---|---|---|---|---|---|---|---|---|---|
| Method | Training dist. informed | ImageNet-O-C | | Places-C | | SUN-C | | iNaturalist-C | | Textures-C | | Average | |
| | | FPR95 ↓ | AUROC ↑ | FPR95 ↓ | AUROC ↑ | FPR95 ↓ | AUROC ↑ | FPR95 ↓ | AUROC ↑ | FPR95 ↓ | AUROC ↑ | FPR95 ↓ | AUROC ↑ |
| MSP | NO | 85.01 | 62.18 | 74.41 | 73.10 | 73.64 | 73.65 | 53.91 | 82.27 | 70.88 | 72.85 | 71.57 | 72.81 |
| Energy | NO | 82.68 | 63.65 | 70.13 | 77.66 | 69.70 | 78.79 | 59.45 | 82.48 | 62.58 | 78.16 | 68.91 | 76.13 |
| Max logit | NO | 83.69 | 63.51 | 71.00 | 76.64 | 70.11 | 77.59 | 53.19 | 83.76 | 65.85 | 76.85 | 68.77 | 75.67 |
| GradNorm | NO | 85.09 | 58.28 | 68.15 | 78.13 | 64.77 | 81.23 | 49.29 | 85.45 | 51.70 | 84.11 | 63.80 | 77.44 |
| ViM | YES | 91.28 | 54.02 | 98.75 | 25.72 | 99.14 | 23.46 | 99.41 | 18.87 | 97.90 | 29.31 | 97.30 | 30.28 |
| KNN | YES | 92.60 | 53.73 | 98.17 | 30.68 | 98.23 | 29.94 | 98.69 | 24.10 | 90.21 | 50.39 | 95.58 | 37.77 |
| MDS$_{single}$ | YES | 92.67 | 50.34 | 99.11 | 21.17 | 99.40 | 18.89 | 99.68 | 14.49 | 98.29 | 26.37 | 97.83 | 26.25 |
| MDS$_{ensemble}$ | YES | 85.89 | 58.76 | 98.08 | 27.51 | 98.23 | 26.49 | 98.31 | 21.40 | 81.15 | 47.73 | 92.33 | 36.38 |
| ODIN | NO | 86.67 | 59.91 | 57.03 | 83.67 | 54.88 | 84.53 | 44.03 | 88.03 | 53.95 | 83.93 | 59.31 | 80.01 |
| ReAct | YES | 90.46 | 54.69 | 89.30 | 66.71 | 88.96 | 68.19 | 92.42 | 64.03 | 86.26 | 66.59 | 89.48 | 64.04 |
| SCALE | NO | 82.06 | 65.18 | 67.20 | 79.03 | 65.58 | 80.33 | 48.40 | 85.75 | 60.80 | 80.03 | 64.81 | 78.06 |
| ASH | NO | 82.45 | 63.90 | 69.82 | 77.91 | 69.26 | 79.08 | 58.88 | 82.76 | 62.00 | 78.55 | 68.48 | 76.44 |
| RTL | NO | 83.70 | 65.26 | 65.44 | 79.96 | 64.27 | 80.32 | 41.39 | 87.69 | 64.08 | 78.36 | 63.78 | 78.32 |
| NNGuide | YES | 88.94 | 56.31 | 81.97 | 69.88 | 81.05 | 73.03 | 74.75 | 76.53 | 58.93 | 79.60 | 77.13 | 71.07 |
| CoRP | YES | 93.02 | 52.30 | 95.18 | 46.32 | 95.22 | 46.33 | 95.99 | 43.54 | 93.05 | 51.72 | 94.49 | 48.04 |
| MDS++ | YES | 82.39 | 64.35 | 90.03 | 52.53 | 90.68 | 52.72 | 64.23 | 71.61 | 66.93 | 68.57 | 78.85 | 61.96 |
| RMDS | YES | 92.91 | 58.76 | 96.74 | 48.73 | 97.50 | 47.40 | 95.35 | 53.60 | 96.64 | 44.86 | 95.83 | 50.69 |
| RMDS++ | YES | 94.61 | 57.05 | 96.02 | 52.76 | 96.38 | 51.52 | 93.87 | 64.11 | 95.33 | 52.20 | 95.24 | 55.53 |
| *DART* | NO | **68.87** | **66.63** | **8.70** | **92.96** | **8.08** | **93.03** | **7.60** | **93.18** | **2.55** | **99.43** | **19.16** | **89.05** |
| | | CIFAR-100-C | | | | | | | | | | | |
| Method | Training dist. informed | SVHN-C | | Places365-C | | LSUN-C | | iSUN-C | | Textures-C | | Average | |
| | | FPR95 ↓ | AUROC ↑ | FPR95 ↓ | AUROC ↑ | FPR95 ↓ | AUROC ↑ | FPR95 ↓ | AUROC ↑ | FPR95 ↓ | AUROC ↑ | FPR95 ↓ | AUROC ↑ |
| MSP | NO | 91.48 | 58.40 | 87.30 | 64.23 | 86.85 | 64.49 | 88.40 | 62.74 | 91.91 | 58.24 | 89.19 | 61.62 |
| Energy | NO | 93.13 | 60.53 | 84.74 | 66.56 | 84.04 | 68.57 | 87.14 | 64.79 | 90.20 | 61.10 | 87.85 | 64.31 |
| Max logit | NO | 92.73 | 60.58 | 85.04 | 66.43 | 84.43 | 68.24 | 87.15 | 64.74 | 90.74 | 60.78 | 88.02 | 64.15 |
| GradNorm | NO | 96.07 | 47.75 | 92.56 | 52.99 | 94.89 | 45.14 | 94.79 | 45.32 | 84.59 | 62.52 | 92.58 | 50.74 |
| ViM | YES | 77.24 | 72.64 | 90.23 | 59.74 | 86.41 | 64.41 | 87.65 | 62.40 | 93.15 | 55.11 | 86.94 | 62.86 |
| KNN | YES | 88.45 | 66.38 | 85.63 | 63.99 | 83.87 | 70.22 | 87.06 | 65.25 | 90.83 | 60.93 | 87.17 | 65.35 |
| MDS$_{single}$ | YES | 89.04 | 61.64 | 91.30 | 57.42 | 84.94 | 65.60 | 86.68 | 62.82 | 95.16 | 48.58 | 89.42 | 59.21 |
| MDS$_{ensemble}$ | YES | 63.57 | 79.33 | 92.49 | 48.49 | 73.78 | 60.77 | 73.95 | 58.87 | 75.16 | 57.51 | 75.79 | 60.99 |
| ODIN | NO | 79.07 | 70.05 | 88.23 | 63.54 | 89.96 | 59.13 | 90.48 | 59.18 | 77.69 | 70.79 | 85.09 | 64.54 |
| ReAct | YES | 95.08 | 55.32 | 88.78 | 62.51 | 86.39 | 67.46 | 88.07 | 64.76 | 91.90 | 60.40 | 90.04 | 62.09 |
| SCALE | NO | 88.88 | 66.46 | 85.55 | 66.91 | 86.30 | 65.89 | 88.06 | 64.21 | 81.14 | 70.69 | 85.99 | 66.83 |
| ASH | NO | 92.05 | 62.73 | 85.42 | 66.48 | 85.24 | 67.78 | 87.99 | 64.34 | 87.66 | 64.41 | 87.67 | 65.18 |
| RTL | NO | 89.27 | 58.60 | 84.59 | 64.62 | 83.65 | 66.99 | 86.90 | 63.60 | 89.30 | 57.65 | 86.74 | 62.29 |
| NNGuide | YES | 91.11 | 62.78 | 88.78 | 63.45 | 91.30 | 58.64 | 91.96 | 57.62 | 77.91 | 68.69 | 88.21 | 62.24 |
| CoRP | YES | 64.51 | **80.52** | 89.98 | 60.28 | 92.56 | 59.80 | 91.30 | 59.11 | 55.25 | **81.49** | 78.72 | 68.40 |
| MDS++ | YES | 95.15 | 62.58 | 87.40 | 64.87 | 87.25 | 66.75 | 89.85 | 62.90 | 95.16 | 59.59 | 90.96 | 63.34 |
| RMDS | YES | 88.78 | 64.07 | 86.67 | 64.65 | 79.57 | 71.82 | 82.68 | 67.91 | 94.23 | 55.06 | 86.34 | 64.70 |
| RMDS++ | YES | 91.52 | 65.51 | 85.70 | 65.59 | 82.73 | 70.26 | 86.08 | 66.38 | 93.78 | 58.41 | 87.96 | 65.23 |
| *DART* | NO | **48.60** | 79.82 | **68.66** | **68.00** | **44.14** | **80.29** | **50.76** | **79.75** | **51.48** | 80.60 | **52.73** | **77.69** |

Transformer-based models on CIFAR-100 benchmarks, all of which are fine-tuned with CIFAR-100. Results on ResNet-50 (He et al., 2016) for ImageNet benchmarks are also provided in the Appendix.

**Baseline Methods.** MSP (Hendrycks & Gimpel, 2016), Max Logit (Hendrycks et al., 2019a), Energy (Liu et al., 2020), ODIN (Liang et al., 2017), GradNorm (Huang et al., 2021), SCALE (Xu et al., 2023), ASH (Djurisic et al., 2022), RTL (Fan et al., 2024), like *DART*, do not require any precomputed statistics or storage from the training data. In contrast, KNN (Sun et al., 2022), ViM (Wang et al., 2022), ReAct (Sun et al., 2021), NNGuide (Park et al., 2023), CoRP (Fang et al., 2024), MDS (Lee et al., 2018) and its variants (Ren et al., 2021; Mueller & Hein, 2025) require pre-computation or storage of reference information from training samples. As MDS has been implemented in prior work using either single-layer or multi-layer, we compare against both variants.

## 4.2 OOD DETECTION RESULTS

### 4.2.1 RESULTS ON COVARIATE SHIFTED DATASET

As *DART* is designed to adapt to test-time covariate shift, it demonstrates its full potential in the covariate-shifted setting. As shown in Table 1, on ImageNet-C benchmark, *DART* achieves the best performance on every OOD dataset and both evaluation metrics, with an average FPR@95TPR reduction of **40.15 pp** and average AUROC gain of **9.04 pp** compared to the second-best (i.e., ODIN). On CIFAR-100-C, *DART* once again achieves the best average performance on both metrics: an average FPR@95TPR reduction of **23.06 pp** and an AUROC gain of **9.29 pp** compared to the second-best. These results highlight the robustness and adaptability of *DART* in the face of test-time covariate shift.

An important observation is that methods relying on prior information from training data tend to perform similarly to, or even worse than, baselines that do not use such priors. This suggests that the prior information which is typically beneficial for OOD detection on clean datasets may become misaligned with test-time distributions under covariate shift, leading to degraded performance.

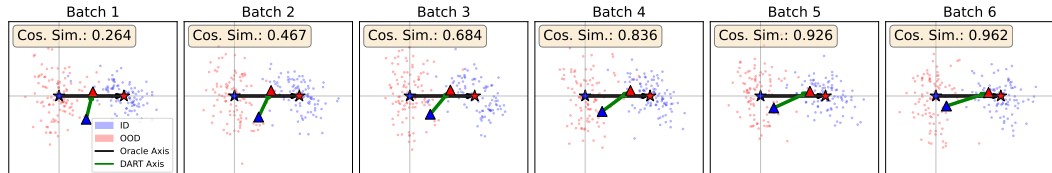

Figure 6: Progression of axis alignment at CIFAR-100-C vs. LSUN-C under impulse noise corruption

### 4.2.2 RESULTS ON CLEAN DATASET

Table 2 reports the OOD detection performance on the clean CIFAR-100 and ImageNet benchmark. Although *DART* mainly targets covariate-shifted environments, it consistently outperforms all baselines on the clean benchmarks, achieving the lowest average FPR@95TPR and the highest average AUROC. *DART* reduces FPR@95 by **19.06 pp** on ImageNet benchmark and **12.72 pp** on CIFAR-100 benchmark compared to the second-best, while improving AUROC by **5.10 pp** and **3.42 pp**, respectively. These results suggest that ID and OOD samples are well-separated in the feature space across most ID and OOD dataset combinations and that the ID and OOD prototypes, while not drastically shifted, are finely adjusted by *DART* toward more optimal axis for discrimination.

Importantly, baselines that leverage training samples to extract prior information before test time—Mahalanobis, KNN, and ViM—tend to outperform those not relying on such information. *DART* is a notable exception, performing the best without any prior information.

Table 2: OOD detection performance without covariate shift. All results are reported as the mean over all five OOD datasets for each ID set. (Best: **bolded**, Second-best: underlined)

| Method | ImageNet | | CIFAR-100 | |
|---|---|---|---|---|
| | FPR95 ↓ | AUROC ↑ | FPR95 ↓ | AUROC ↑ |
| MSP | 44.64 | 86.94 | 80.37 | 75.29 |
| Energy | 38.25 | 85.52 | 79.95 | 76.79 |
| Max logit | 37.05 | 86.43 | 79.91 | 77.06 |
| GradNorm | 80.50 | 57.26 | 94.92 | 43.62 |
| ViM | 56.58 | 87.60 | 71.94 | 75.61 |
| KNN | 84.79 | 75.87 | 71.48 | 81.16 |
| MDS$_{single}$ | 79.19 | 80.57 | 80.01 | 69.98 |
| MDS$_{ensemble}$ | 55.30 | 86.86 | 45.94 | 86.36 |
| ODIN | 60.84 | 81.86 | 81.11 | 69.50 |
| ReAct | 86.40 | 70.67 | 79.45 | 76.09 |
| SCALE | 35.01 | 87.02 | 76.60 | 77.53 |
| ASH | 37.77 | 85.67 | 80.44 | 76.76 |
| RTL | 42.30 | 84.24 | 64.11 | 80.76 |
| NNGuide | 70.60 | 70.05 | 87.55 | 66.29 |
| CoRP | 49.03 | 90.17 | 65.54 | 81.05 |
| MDS++ | 41.87 | 90.48 | 85.96 | 75.52 |
| RMDS | 46.28 | 91.03 | 67.90 | 82.51 |
| RMDS++ | 56.01 | 89.73 | 73.91 | 81.57 |
| *DART* | **15.95** | **96.13** | **33.12** | **89.78** |

### 4.2.3 RESULTS WITH TRANSFORMER ARCHITECTURES

We also conduct experiments on transformer-based architectures (Vaswani et al., 2017), specifically ViT-Tiny (Dosovitskiy et al., 2021; Winkawaks, 2023) and Swin-Tiny (Liu et al., 2021), to demonstrate the robustness of our method across different model architectures. As shown in the Table 3, *DART* consistently outperforms all baseline methods on both the covariate-shifted datasets and original datasets. Specifically, *DART* achieves **17.45pp** and **37.11pp** reductions in FPR@95 on covariate-shifted benchmarks for ViT-Tiny and Swin-Tiny, respectively. On clean benchmarks, it demonstrates **8.6pp** and **3.42pp** FPR@95 improvements for ViT-Tiny and Swin-Tiny, respectively. This result proves the superiority of our method and the emergence of discriminative axis in the feature space regardless of the underlying model architecture.

### 4.3 ABLATION STUDIES

**Progression of Axis-alignment.** Figure 6 demonstrates the online convergence capability of *DART* in discovering the oracle discriminative axis. The stars represent the global centroids of ID and OOD, with their connecting line forming the oracle discriminative axis. The triangles indicate the ID and OOD prototypes estimated by *DART* at each batch, whose connecting line represents the online discriminative axis. The cosine similarity between these two axes increases dramatically across batches, demonstrating *DART*'s ability to navigate the high-dimensional feature space and progressively align with the true discriminative direction.

**Impact of Multi-layer Fusion.** Figure 7a compares full *DART* against its single-layer variants on CIFAR-100 vs. LSUN, demonstrating the impact of multi-layer fusion. While *DART* achieves the highest average performance, the key advantage lies in *stability*. Single-layer variants occasionally outperform *DART* in specific settings (e.g., Block3 on original data, Block1 under Gaussian noise) but exhibit catastrophic failures under other corruptions due to varying covariate-shift impacts across layers. Therefore, *DART*'s multi-layer ensemble provides robustness under diverse covariate shifts.

**Impact of Flip Correction.** Figure 7b demonstrates the critical role of *DART*'s flip correction. We compare *DART* with *DART*-NoFlip (without flip correction) on CIFAR-100 vs iSUN for original and

Table 3: OOD detection performance comparison with ViT-Tiny and Swin-T architectures. We evaluate with CIFAR-100 ID, CIFAR-100-C csID and the corresponding OODs. All results are reported as the mean over all five OOD datasets. For covariate shifted datasets, results are the average of all 15 corruptions with severity level 5. (Best: **bolded**, Second-best: underlined)

| Method | Training dist. informed | Covariate shifted | | | | Clean | | | |
|---|---|---|---|---|---|---|---|---|---|
| | | ViT-Tiny | | Swin-T | | ViT-Tiny | | Swin-T | |
| | | FPR95 ↓ | AUROC ↑ | FPR95 ↓ | AUROC ↑ | FPR95 ↓ | AUROC ↑ | FPR95 ↓ | AUROC ↑ |
| MSP | NO | 89.56 | 57.03 | 85.24 | 63.39 | 70.36 | 79.79 | 60.16 | 84.31 |
| Energy | NO | 87.01 | 61.20 | 80.30 | 68.35 | 58.77 | 84.83 | 41.06 | 89.80 |
| Max Logit | NO | 87.58 | 60.65 | 81.90 | 67.73 | 60.27 | 84.54 | 42.11 | 89.52 |
| GradNorm | NO | 89.54 | 58.42 | 78.77 | 71.44 | 78.33 | 75.24 | 71.63 | 74.53 |
| ViM | YES | 89.49 | 58.75 | 96.94 | 45.76 | 58.27 | 85.06 | 93.00 | 66.26 |
| KNN | YES | 90.15 | 56.81 | 84.25 | 64.91 | 67.35 | 79.58 | 45.93 | 88.51 |
| MDS$_{single}$ | YES | 97.22 | 32.40 | 98.76 | 33.09 | 96.38 | 35.57 | 98.67 | 41.11 |
| MDS$_{ensemble}$ | YES | 61.73 | 65.95 | 98.60 | 33.75 | 29.18 | 88.40 | 98.42 | 45.53 |
| ODIN | NO | 87.53 | 60.52 | 84.88 | 63.63 | 80.25 | 71.22 | 91.20 | 63.65 |
| ReAct | YES | 85.94 | 62.01 | 76.76 | 71.14 | 56.86 | 81.79 | 41.78 | 89.51 |
| SCALE | NO | 88.84 | 58.99 | 76.68 | 71.57 | 61.75 | 83.66 | 44.84 | 88.50 |
| ASH | NO | 92.00 | 56.00 | 76.95 | 71.39 | 81.17 | 74.00 | 50.03 | 86.97 |
| RTL | NO | 84.42 | 58.69 | 83.71 | 63.62 | 40.20 | 87.75 | 50.90 | 84.23 |
| NNGuide | YES | 86.49 | 61.17 | 76.53 | 73.38 | 60.30 | 83.43 | 38.04 | 91.06 |
| CoRP | YES | 88.31 | 59.85 | 86.17 | 63.65 | 67.28 | 82.43 | 50.37 | 88.53 |
| MDS++ | YES | 79.15 | 66.25 | 76.01 | 69.80 | 46.70 | 87.12 | 35.28 | **91.54** |
| RMDS | YES | 82.63 | 63.56 | 91.33 | 58.14 | 47.98 | 86.93 | 54.00 | 87.65 |
| RMDS++ | YES | 81.30 | 64.05 | 88.13 | 61.81 | 48.11 | 86.71 | 47.01 | 88.90 |
| *DART* | NO | **44.28** | **68.60** | **38.90** | **81.28** | **20.58** | **94.31** | **31.86** | 88.03 |

(a) Impact of Multi-layer Feature Utilization

(b) Impact of Flip Correction across different corruption types.

Figure 7: Impact of *DART*'s individual components

corrupted datasets. While both perform identically on original data and under Gaussian noise, Shot noise reveals a drastic difference. Standard *DART* recovers via prototype flip and performs robustly, while *DART*-NoFlip suffers catastrophic degradation of detection capability due to reversed prototypes under Shot noise. The flip correction detects and rectifies these inversions by ensuring consistent prototype proximity. Similar effects with JPEG compression further validate the effectiveness.

## 5 CONCLUSION

In this work, we addressed the realistic challenge of OOD detection under test-time covariate shift, a scenario where existing methods often collapse. Our analysis revealed the consistent existence of a discriminative axis along which covariate-shifted ID and OOD samples remain separable. Building on this insight, we proposed *DART*, which dynamically tracks prototypes to recover the evolving discriminative axis with multi-layer fusion. Extensive experiments across diverse datasets and architectures confirmed its superiority over strong baselines, underscoring the promise of prototype-based axis tracking as a practical solution for reliable OOD detection in real-world environments.

**Limitations and Future Works.** While *DART* demonstrates strong performance, several avenues remain for improvement. The reliance on MSP-based initialization may impact performance when initial pseudo-labeling quality is poor, suggesting a need for more robust initialization strategies. Additionally, extending *DART* beyond vision tasks to other modalities presents an opportunity to validate the universality of discriminative axis tracking across different data representations.

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

# APPENDIX

## A EXPERIMENTAL DETAILS

### A.1 DATASETS DETAILS

**CIFAR-100** CIFAR-100 (Krizhevsky et al., 2009) consists of 60,000 color images of size $32 \times 32$ across 100 object classes, with 600 images per class. The dataset is divided into 50,000 training and 10,000 test samples. It includes diverse categories such as animals, vehicles, and everyday objects, and is commonly used for evaluating fine-grained image classification and representation learning. In our experiments, we use CIFAR-100 as the in-distribution dataset.

**SVHN** The Street View House Numbers (SVHN) dataset (Netzer et al., 2011) contains real-world digit images collected from Google Street View. It consists of over 600,000 images, each containing a single digit cropped from house number signs, with a resolution of $32 \times 32$. The dataset includes 10 classes (digits 0–9) and is known for its relatively low intra-class variability and high image quality. We use SVHN as an out-of-distribution dataset in our evaluation.

**LSUN** The Large-scale Scene UNderstanding (LSUN) dataset (Yu et al., 2015) contains millions of high-resolution images across various indoor and outdoor scene categories such as classroom, church, and bridge. In OOD detection benchmarks, a subset of LSUN is often used by resizing images to $32 \times 32$ resolution to match CIFAR-style inputs. In our experiments, we use the resized LSUN images as out-of-distribution samples.

**iSUN** The iSUN dataset (Xu et al., 2015) consists of natural scene images collected for saliency prediction, containing various indoor and outdoor environments. It includes around 6,000 images, which are typically resized to $32 \times 32$ for compatibility with CIFAR-based architectures. Due to its scene-centric content, iSUN is commonly used as an out-of-distribution dataset in image classification tasks. We follow prior works and use the resized version of iSUN for OOD evaluation.

**Textures** The Textures dataset (Cimpoi et al., 2014), also known as the Describable Textures Dataset (DTD), contains 5,640 texture images spanning 47 categories such as striped, dotted, and cracked. The images are collected "in the wild" and exhibit a wide range of fine-grained, low-level patterns. Its low semantic content and high texture diversity make it a challenging out-of-distribution benchmark.

**ImageNet** ImageNet-1K dataset (Deng et al., 2009) contains 1.28M training images and 50K validation images across 1,000 object categories.

**ImageNet-O** ImageNet-O (Hendrycks et al., 2021) is a curated out-of-distribution dataset containing 2,000 natural images that are semantically distinct from the 1,000 classes in ImageNet-1k. The images were collected to naturally lie outside the ImageNet taxonomy while maintaining comparable visual complexity. This dataset serves as a challenging benchmark for evaluating semantic OOD detection.

**SUN** The SUN dataset (Xiao et al., 2010) is a large-scale scene understanding benchmark containing over 130,000 images across a wide variety of indoor and outdoor environments. It covers hundreds of semantic scene categories such as kitchen, mountain, and library. The diversity and scene-centric nature of SUN make it a strong candidate for out-of-distribution evaluation.

**iNaturalist** The iNaturalist dataset (Van Horn et al., 2018) contains high-resolution images of fine-grained natural categories such as plants, insects, birds, and mammals, collected from citizen science platforms. Due to its distinct domain and taxonomic diversity, iNaturalist is widely used as an out-of-distribution benchmark in vision tasks. Its semantic gap from object-centric datasets makes it a challenging OOD evaluation setting.

**Common corruptions**   To evaluate robustness under covariate shift, we use a set of common image corruptions introduced by Hendrycks and Dietterich (Hendrycks & Dietterich, 2019). This benchmark includes 15 corruption types, grouped into noise (e.g., Gaussian noise, shot noise), blur (e.g., defocus, motion blur), weather (e.g., snow, fog), and digital distortions (e.g., JPEG compression, pixelation). We apply these corruptions to both in-distribution and out-of-distribution test samples to simulate realistic distribution shifts. Each corruption is applied at severity level 5, following the standard protocol used in prior robustness benchmarks.

## A.2   BASELINES DETAILS

We introduce the baselines compared with *DART* and specify the hyperparameter used for implementation. The hyperparameter settings mainly follow the settings from the original paper.

**MSP**   Maximum Softmax Probability (Hendrycks & Gimpel, 2016) uses the highest softmax output value as the confidence score, assuming in-distribution samples yield higher confidence. We extract this directly from the classifier's final layer.

**Energy**   Energy-based detection (Liu et al., 2020) computes $E(x) = -\log \sum_i \exp(f_i(x))$ from network logits, with lower values indicating in-distribution samples. $T = 1.0$ is used for temperature scaling.

**Max logit**   This method (Hendrycks et al., 2019a) uses the maximum pre-softmax logit value as the score, avoiding the normalization effect of softmax that may mask useful signals in relative logit magnitudes.

**GradNorm**   GradNorm (Huang et al., 2021) measures the gradient magnitude of the loss with respect to the penultimate layer features. OOD samples tend to produce larger gradient norms. We use a temperature of 1.0 for all experiments.

**ViM**   Virtual logit Matching (Wang et al., 2022) projects features into a null space and creates a virtual logit to enhance separation between ID and OOD samples. We set the dimension of the null space to 1000 for feature dimensions $\geq 1500$, to 512 for feature dimensions $\geq 768$, and to half the size of the feature dimensions otherwise.

**KNN**   K-nearest neighbors (Sun et al., 2022) measures the distance to k-nearest neighbors in feature space, with OOD samples typically farther from ID samples. We use L2 normalization for features and set k=50 for CIFAR-based experiments and k=200 for ImageNet-based experiments.

**Mahalanobis distance**   We implement both single-layer and ensemble versions of this method (Lee et al., 2018). The single-layer version models class-conditional feature distributions using Gaussian distributions and measures the distance to the nearest class distribution, using the penultimate layer features. The ensemble version combines layer-wise scores from multiple network layers using pre-computed weights. These weights are learned by utilizing FGSM-perturbed inputs (magnitude 0.001) as synthetic OOD data and applying logistic regression (regularization strength C=1.0, max iterations=1000) to determine the contribution of each layer's feature. We extract features for each layer or block depending on the model architecture.

**ODIN**   ODIN (Liang et al., 2018) enhances OOD detection by applying input perturbations with temperature scaling to create a larger gap between ID and OOD confidence scores. FGSM epsilon values are set as 0.002 for both CIFAR-100 and ImageNet.

**ReAct**   ReAct (Sun et al., 2021) truncates abnormally high hidden activations at test time, reducing model overconfidence on OOD data while preserving ID performance, thereby improving the separability between ID and OOD sample.

**SCALE**   SCALE (Xu et al., 2023) is a post-hoc OOD detection method that applies activation scaling to penultimate features, thereby enlarging the separation between ID and OOD energy scores.

**ASH**    ASH (Djurisic et al., 2022) prunes a large portion of late-layer activations (e.g., by top-K percentile) and either leaves the remaining values(ASH-P), binarizes them(ASH-B), or rescales them(ASH-S), then propagates the simplified representation through the network for scoring. We use ASH-P for performance comparison.

**RTL**    RTL (Fan et al., 2024) fits a linear regression between OOD scores and network features at test time, calibrating base detector outputs to improve detection performance through test-time adaptation.

**NNGuide**    NNGuide (Park et al., 2023) leverages k-nearest neighbor distances in the feature space, scaled by the model's confidence scores, to guide OOD detection by measuring how similar a test sample is to training samples while accounting for prediction confidence.

**CoRP**    CoRP (Fang et al., 2024) applies cosine normalization followed by Random Fourier Features approximation of a Gaussian kernel, then computes PCA reconstruction errors for OOD detection.

**MDS++**    MDS++ (Mueller & Hein, 2025) enhances the standard Mahalanobis Distance Score by applying L2 normalization to feature representations before computing class-conditional statistics, thereby improving the geometric separation between ID and OOD samples in the normalized feature space.

**RMDS**    RMDS (Relative Mahalanobis Distance Score) (Ren et al., 2021) computes relative Mahalanobis distances by comparing class-conditional scores against global background scores, effectively measuring how much a sample deviates from both class-specific and overall data distributions.

**RMDS++**    RMDS++ (Mueller & Hein, 2025) extends RMDS by incorporating L2 feature normalization before computing relative Mahalanobis distances, combining the benefits of normalized feature spaces with relative distance measurements to achieve more robust OOD detection.

### A.3    EVALUATION MODEL DETAILS

For CIFAR-100-based benchmark, we use the pre-trained WideResNet (Zagoruyko & Komodakis, 2016) with 40 layers and widen factor of 2 pretrained with AugMix (Hendrycks et al., 2019b) on clean CIFAR-100. The pretrained weights for this model is available from RobustBench (Croce et al., 2020).

For ImageNet-based benchmark, we use the pre-trained RegNetY-16GF (He et al., 2016) with the PyTorch checkpoint (Paszke et al., 2019), which is trained on ImageNet and widely used for OOD detection task.

For evaluation on transformer-based architectures, we train two models: ViT-Tiny and Swin-Tiny. Both models are initialized with ImageNet-pretrained weights provided by HuggingFace model hub. We then fine-tune the model weights and classifier on the CIFAR-100 dataset. Training continues until each model reaches its target accuracy threshold (80% for ViT-Tiny and 85% for Swin-Tiny), after which early stopping is applied.

### A.4    EVALUATION DETAILS

For evaluation, we construct each test-time batch to contain 100 in-distribution (ID) samples and 100 out-of-distribution (OOD) samples, resulting in a fixed batch size of 200. We sample a total of 100 such test batches for each experimental setting. For each batch, we compute the AUROC and FPR@95TPR metrics, and report the final performance by averaging the values across all batches.

### A.5    COMPUTE RESOURCES

All experiments were conducted using NVIDIA RTX 3090 and RTX 4090 GPUs.

# B  METHOD DETAILS

## B.1  OTSU ALGORITHM

To automatically determine a threshold that separates two distributions (e.g., ID and OOD) based on their scalar scores, we adopt Otsu algorithm (Otsu et al., 1975). Originally proposed for image binarization, Otsu algorithm selects the threshold that minimizes the intra-class variance (or equivalently maximizes the inter-class variance) when partitioning a set of scalar values into two groups.

Given a histogram of score values, the algorithm exhaustively searches for the threshold $\tau$ that minimizes the weighted sum of within-class variances:

$$\sigma_{\text{within}}^2(\tau) = \omega_0(\tau)\sigma_0^2(\tau) + \omega_1(\tau)\sigma_1^2(\tau), \tag{5}$$

where $\omega_0(\tau)$ and $\omega_1(\tau)$ are the probabilities of the two classes separated by threshold $\tau$, and $\sigma_0^2(\tau), \sigma_1^2(\tau)$ are the corresponding class variances. This approach allows for an adaptive and data-driven determination of the decision threshold, without requiring access to ground-truth labels or distributional assumptions.

In our method, Otsu algorithm is applied to the distribution of OOD scores computed over each test-time batch. This enables unsupervised, on-the-fly threshold selection for distinguishing ID and OOD samples, and plays a critical role in decision-making process during inference.

## B.2  TUKEY'S METHOD

To ensure robust prototype estimation, we apply outlier filtering prior to aggregating the feature representations of test samples. Specifically, we adopt Tukey's method, a non-parametric technique for identifying outliers based on the interquartile range (IQR) (Tukey et al., 1977).

Given a set of distance values (e.g., Euclidean distances between features and their assigned prototype), we first compute the lower quartile ($Q_1$) and upper quartile ($Q_3$). The interquartile range is then defined as:

$$\text{IQR} = Q_3 - Q_1. \tag{6}$$

A sample is identified as a potential outlier if its score $x$ satisfies:

$$x > Q_3 + 1.5 \cdot \text{IQR}. \tag{7}$$

We use Tukey's method with an IQR factor of 1.5 throughout our experiments.

This filtering step is applied independently to the distance scores within each test-time batch, effectively removing extreme values that may otherwise distort the prototype update.

## B.3  LAYER SELECTION FOR *DART*

While it is possible to utilize the output of all intermediate layers for multi-layer aggregation, doing so incurs additional computational overhead. To reduce this overhead while still capturing hierarchical representations, we select a subset of representative layers at coarse block granularity, as specified in Table 4.

Table 4: Included layers list for *DART*

| Model architecture | Included layers list |
|---|---|
| WideResNet-40-2 | `block1, block2, block3, fc` |
| RegNetY-16GF | `stem, trunk output {block1.block1-0, block1.block1-1, block2.block2-0 - block2.block2-3, block3.block3-0 - block3.block3-10, block4.block4-0}, fc` |
| ViT-Tiny | `vit.encoder.layer{0 - 11}, classifier` |
| Swin-T | `swin.encoder.layers{0.blocks.0, 0.blocks.1, 1.blocks.0, 1.blocks.1 2.blocks.0 - 2.blocks.5, 3.blocks.0, 3.blocks.1}, classifier` |
| ResNet-50 | `layer1, layer2, layer3, layer4, fc` |

**Algorithm 1** *DART*

---

**Require:** Pre-trained model $f$, layers $L = \{1, \dots, L\}$, EMA coefficient $\alpha$

1: **Initialization using the first batch $\mathcal{B}_1$:**
2: Let $o = (o_1, \dots, o_C)$ be the logit vector
3: Compute MSP: $\text{MSP}_{1,i} = \max_c \frac{\exp(o_c)}{\sum_{j=1}^{C} \exp(o_j)}$
4: Apply Otsu threshold $\tau_1$ on MSP scores for pseudo-labeling:
$$\hat{y}_i = \begin{cases} \text{ID} & \text{if } \text{MSP}_{1,i} \geq \tau_1 \\ \text{OOD} & \text{otherwise} \end{cases}$$
5: Partition features into $\mathcal{B}_1^{ID}$ and $\mathcal{B}_1^{OOD}$ based on $\hat{y}_i$
6: **for** each layer $l \in L$ **do**
7:     Initialize prototypes:
$$\bar{\mathbf{p}}_1^{(l),\text{ID}} = \frac{1}{|\mathcal{B}_1^{\text{ID}}|} \sum_{\mathbf{x}_{1,i} \in \mathcal{B}_1^{\text{ID}}} f_l(\mathbf{x}_{1,i}), \quad \bar{\mathbf{p}}_1^{(l),\text{OOD}} = \frac{1}{|\mathcal{B}_1^{\text{OOD}}|} \sum_{\mathbf{x}_{1,i} \in \mathcal{B}_1^{\text{OOD}}} f_l(\mathbf{x}_{1,i})$$
8: **end for**

9: **for** each batch $\mathcal{B}_t$ **do**
10:     **for** each layer $l \in L$ **do**
11:         **if** $t \mod n = 0$ **then**
12:             Apply flip correction if prototypes are misaligned. Refer to Section 3.3 in main paper for details.
13:         **end if**
14:         Extract features: $\mathbf{z}_{t,i}^{(l)} = f_l(\mathbf{x}_{t,i}), \ \mathbf{x}_{t,i} \in \mathcal{B}_t$
15:         Compute RDS:
$$\text{RDS}_i^{(l)} = 1 - \frac{\|\mathbf{z}_i^{(l)} - \bar{\mathbf{p}}_{t-1}^{ID,(l)}\|}{\|\mathbf{z}_i^{(l)} - \bar{\mathbf{p}}_{t-1}^{ID,(l)}\| + \|\mathbf{z}_i^{(l)} - \bar{\mathbf{p}}_{t-1}^{OOD,(l)}\|}$$
16:         Apply Otsu threshold $\tau_t$ on RDS scores for pseudo-labeling:
$$\hat{y}_i = \begin{cases} \text{ID} & \text{if } \text{RDS}_i^{(l)} \geq \tau_t \\ \text{OOD} & \text{otherwise} \end{cases}$$
17:         Partition features into $\mathcal{B}_t^{ID}$ and $\mathcal{B}_t^{OOD}$ based on $\hat{y}_i$
18:         Apply Tukey's method for outlier filtering:
            Let $s_i = \|z_{t,i}^{(l)} - p^{\text{proto}}\|_2$, where $p^{\text{proto}}$ is the corresponding prototype
            Filter out $z_{t,i}^{(l)}$ if $s_i > Q_3 + k \cdot \text{IQR}$
19:         Compute new centers:
$$\hat{\mathbf{p}}_t^{(l),\text{ID}} = \frac{1}{|\mathcal{B}_t^{\text{ID}}|} \sum_{\mathbf{x}_{t,i} \in \mathcal{B}_t^{\text{ID}}} f_l(\mathbf{x}_{t,i}), \quad \hat{\mathbf{p}}_t^{(l),\text{OOD}} = \frac{1}{|\mathcal{B}_t^{\text{OOD}}|} \sum_{\mathbf{x}_{t,i} \in \mathcal{B}_t^{\text{OOD}}} f_l(\mathbf{x}_{t,i})$$
20:         Update prototypes with EMA:
$$\bar{\mathbf{p}}_t^{ID,(l)} = \alpha \cdot \bar{\mathbf{p}}_t^{ID,(l)} + (1-\alpha) \cdot \hat{\mathbf{p}}_{t-1}^{ID,(l)}, \bar{\mathbf{p}}_t^{OOD,(l)} = \alpha \cdot \bar{\mathbf{p}}_t^{OOD,(l)} + (1-\alpha) \cdot \hat{\mathbf{p}}_{t-1}^{OOD,(l)}$$
21:     **end for**
22:     $\text{RDS}_{\text{multi}}(x_i) = \frac{1}{L} \sum_{l=1}^{L} \text{RDS}_i^{(l)}$
23: **end for**

---

## B.4 ALGORITHM OF *DART*

## C FULL RESULTS

Here we show the full results for all OOD datasets which was abbreviated as average in the main manuscript due to space limits.

### C.1 FULL CIFAR-100 RESULTS WITH WIDERESNET ON CLEAN DATASET

Table 5: OOD detection performance comparison with CIFAR-100 ID and the corresponding OODs. FPR@95TPR (%) is lower the better and AUROC (%) is higher the better. (Best: bolded, Second-best: underlined)

| Method | Training dist. informed | SVHN | | Places365 | | LSUN | | iSUN | | Textures | | Average | |
|---|---|---|---|---|---|---|---|---|---|---|---|---|---|
| | | FPR95 ↓ | AUROC ↑ | FPR95 ↓ | AUROC ↑ | FPR95 ↓ | AUROC ↑ | FPR95 ↓ | AUROC ↑ | FPR95 ↓ | AUROC ↑ | FPR95 ↓ | AUROC ↑ |
| MSP | NO | 79.32 | 77.27 | 80.37 | 74.99 | 78.39 | 76.91 | 81.23 | 74.56 | 82.55 | 72.70 | 80.37 | 75.29 |
| Energy | NO | 80.32 | 78.74 | 78.47 | 75.69 | 78.39 | 78.68 | 83.42 | 74.53 | 79.17 | 76.31 | 79.95 | 76.79 |
| Max logit | NO | 79.85 | 79.11 | 78.63 | 76.01 | 77.87 | 78.92 | 82.49 | 75.00 | 79.69 | 76.25 | 79.71 | 77.06 |
| GradNorm | NO | 96.80 | 44.47 | 94.81 | 51.36 | 98.22 | 32.12 | 98.48 | 30.97 | 86.31 | 59.18 | 94.92 | 43.62 |
| ViM | YES | 54.18 | 85.44 | 86.74 | 64.33 | 66.48 | 81.52 | 66.82 | 80.95 | 85.47 | 65.83 | 71.94 | 75.61 |
| KNN | YES | 63.34 | 86.06 | 80.09 | 73.38 | 66.57 | 84.85 | 72.86 | 80.60 | 74.54 | 80.89 | 71.48 | 81.16 |
| Mahalanobis$_{single}$ | YES | 77.75 | 75.58 | 90.94 | 59.26 | 69.56 | 80.02 | 70.64 | 78.54 | 91.15 | 56.50 | 80.01 | 69.98 |
| Mahalanobis$_{ensemble}$ | YES | 62.92 | 88.78 | 93.41 | 61.12 | **13.12** | **97.34** | 16.44 | 96.55 | 43.79 | 87.99 | 45.94 | 86.36 |
| ODIN | NO | 64.15 | 80.96 | 84.30 | 69.48 | 90.32 | 63.24 | 91.03 | 61.21 | 75.75 | 72.60 | 81.11 | 69.50 |
| ReAct | YES | 87.25 | 70.36 | 79.97 | 75.71 | 73.91 | 80.58 | 75.75 | 74.48 | 80.39 | 75.32 | 79.45 | 76.09 |
| SCALE | NO | 74.38 | 81.36 | 78.47 | 75.55 | 80.91 | 75.14 | 83.88 | 72.60 | 65.34 | 82.98 | 76.60 | 77.53 |
| ASH | NO | 79.20 | 79.65 | 79.96 | 75.11 | 81.64 | 77.12 | 85.45 | 73.20 | 75.96 | 78.72 | 80.44 | 76.76 |
| RTL | NO | 50.15 | 87.42 | 73.23 | 75.59 | 58.46 | 85.01 | 68.38 | 80.67 | 70.33 | 75.12 | 64.11 | 80.76 |
| NNGuide | YES | 86.85 | 71.34 | 87.71 | 67.73 | 94.66 | 59.74 | 95.50 | 57.33 | 73.01 | 75.32 | 87.55 | 66.29 |
| CoRP | YES | 43.43 | 91.59 | 86.30 | 68.31 | 82.53 | 77.18 | 80.56 | 77.13 | **34.89** | **91.04** | 65.54 | 81.16 |
| MDS++ | YES | 83.59 | 81.25 | 85.42 | 70.98 | 84.16 | 77.33 | 86.64 | 73.88 | 90.00 | 74.15 | 85.96 | 75.52 |
| RMDS | YES | 68.73 | 85.23 | 77.53 | 77.11 | 52.64 | 88.28 | 57.46 | 85.64 | 83.13 | 76.29 | 67.90 | 82.51 |
| RMDS++ | YES | 74.20 | 84.57 | 76.79 | **77.67** | 65.24 | 85.70 | 71.02 | 82.49 | 82.29 | 77.44 | 73.91 | 82.51 |
| *DART* | NO | **9.64** | **97.67** | **70.00** | 75.12 | 30.24 | 91.49 | **14.79** | **96.62** | 40.91 | 88.02 | **33.12** | **89.78** |

### C.2 FULL IMAGENET RESULTS WITH REGNET ON CLEAN DATASET

Table 6: OOD detection performance comparison with ImageNet ID and the corresponding OODs. FPR@95TPR (%) is lower the better and AUROC (%) is higher the better. (Best: bolded, Second-best: underlined)

| Method | Training dist. informed | IN-O | | Places | | SUN | | iNaturalist | | Textures | | Average | |
|---|---|---|---|---|---|---|---|---|---|---|---|---|---|
| | | FPR95 ↓ | AUROC ↑ | FPR95 ↓ | AUROC ↑ | FPR95 ↓ | AUROC ↑ | FPR95 ↓ | AUROC ↑ | FPR95 ↓ | AUROC ↑ | FPR95 ↓ | AUROC ↑ |
| MSP | NO | 52.87 | 83.02 | 54.76 | 84.30 | 52.16 | 84.92 | 20.33 | 95.25 | 43.08 | 87.23 | 44.64 | 86.94 |
| Energy | NO | 57.59 | 76.17 | 48.11 | 81.05 | 41.17 | 85.04 | 8.68 | 97.49 | 35.69 | 87.87 | 38.25 | 85.52 |
| Max logit | NO | 52.30 | 78.42 | 47.22 | 82.29 | 41.65 | 85.60 | 9.50 | 97.43 | 34.57 | 88.39 | 37.05 | 86.43 |
| GradNorm | NO | 93.95 | 34.15 | 87.71 | 52.32 | 79.27 | 62.51 | 74.90 | 63.44 | 66.67 | 73.87 | 80.50 | 57.26 |
| ViM | YES | 30.21 | 93.82 | 71.37 | 82.93 | 72.21 | 83.42 | 50.70 | 90.12 | 58.42 | 87.70 | 56.58 | 87.60 |
| KNN | YES | 69.58 | 85.71 | 97.60 | 69.55 | 96.05 | 71.65 | 99.40 | 66.69 | 61.32 | 85.77 | 84.79 | 75.87 |
| Mahalanobis$_{single}$ | YES | 48.14 | 90.43 | 87.04 | 76.70 | 90.16 | 76.11 | 88.65 | 78.79 | 81.98 | 80.81 | 79.19 | 80.57 |
| Mahalanobis$_{ensemble}$ | YES | 8.69 | 97.82 | 78.62 | 81.37 | 77.80 | 82.59 | 88.75 | 77.80 | 22.62 | 94.71 | 55.30 | 86.86 |
| ODIN | NO | 60.40 | 82.05 | 71.28 | 76.69 | 68.97 | 77.47 | 47.64 | 88.78 | 55.90 | 84.33 | 60.84 | 81.86 |
| ReAct | YES | 97.32 | 53.19 | 90.93 | 65.47 | 85.13 | 72.83 | 74.11 | 85.74 | 84.53 | 76.10 | 86.40 | 70.67 |
| SCALE | NO | 52.97 | 77.70 | 45.51 | 82.83 | 37.89 | 86.79 | 8.72 | 97.46 | 29.96 | 90.30 | 35.01 | 87.02 |
| ASH | NO | 57.04 | 76.29 | 47.82 | 81.17 | 40.78 | 85.21 | **8.43** | 98.01 | 34.80 | 88.15 | 37.77 | 85.67 |
| RTL | NO | 52.60 | 79.13 | 49.71 | 83.58 | 48.01 | 82.53 | 21.53 | 90.75 | 39.67 | 85.22 | 42.30 | 84.24 |
| NNGuide | YES | 87.68 | 49.90 | 85.03 | 61.03 | 76.69 | 68.44 | 49.44 | 87.67 | 54.18 | 83.23 | 70.60 | 70.05 |
| CoRP | YES | 42.37 | 93.23 | 68.17 | 83.92 | 62.14 | 86.35 | 33.19 | 94.68 | 39.27 | 92.67 | 49.03 | 90.17 |
| MDS++ | YES | 34.49 | 94.26 | 73.19 | 81.07 | 65.30 | 84.69 | 9.86 | **98.01** | 26.49 | 94.37 | 41.87 | 90.48 |
| RMDS | YES | 41.02 | 92.19 | 63.46 | 87.29 | 61.54 | 88.84 | 9.41 | 97.54 | 55.96 | 89.27 | 46.28 | 91.03 |
| RMDS++ | YES | 57.33 | 90.42 | 73.07 | 85.76 | 70.16 | 87.53 | 17.70 | 96.44 | 61.79 | 88.50 | 56.01 | 89.73 |
| *DART* | NO | **0.59** | **99.82** | **14.41** | **96.31** | **36.14** | **90.86** | 20.71 | 95.28 | **7.90** | **98.36** | **15.95** | **96.13** |

## C.3 FULL CIFAR-100 RESULTS WITH VIT-TINY

Table 7: OOD detection performance comparison with ViT-Tiny. We evaluate with CIFAR-100 ID, CIFAR-100-C csID and the corresponding OODs. (Best: **bolded**, Second-best: underlined)

| | | Covariate Shifted | | | | | | | | | | | |
|---|---|---|---|---|---|---|---|---|---|---|---|---|---|
| Method | Training dist. informed | SVHN-C | | Places365-C | | LSUN-C | | iSUN-C | | Textures-C | | Average | |
| | | FPR95 ↓ | AUROC ↑ | FPR95 ↓ | AUROC ↑ | FPR95 ↓ | AUROC ↑ | FPR95 ↓ | AUROC ↑ | FPR95 ↓ | AUROC ↑ | FPR95 ↓ | AUROC ↑ |
| MSP | NO | 88.32 | 54.65 | 89.98 | 57.41 | 89.13 | 60.20 | 90.96 | 57.75 | 89.39 | 55.15 | 89.56 | 57.03 |
| Energy | NO | 83.57 | 60.42 | 88.13 | 60.19 | 87.47 | 63.71 | 89.33 | 60.96 | 86.57 | 60.73 | 87.01 | 61.20 |
| Max logit | NO | 84.66 | 59.74 | 88.59 | 59.76 | 87.74 | 63.31 | 89.76 | 60.53 | 87.17 | 59.93 | 87.58 | 60.65 |
| GradNorm | NO | 89.95 | 62.39 | 89.89 | 56.74 | 92.07 | 54.39 | 92.37 | 53.47 | 83.40 | 65.13 | 89.54 | 58.42 |
| ViM | YES | 86.98 | 58.01 | 91.24 | 57.30 | 88.56 | 62.26 | 90.90 | 58.85 | 89.78 | 57.33 | 89.49 | 58.75 |
| KNN | YES | 90.16 | 56.65 | 90.34 | 54.45 | 90.01 | 60.53 | 92.23 | 55.79 | 87.99 | 56.64 | 90.15 | 56.81 |
| Mahalanobis$_{single}$ | YES | 97.65 | 32.82 | 96.87 | 34.13 | 96.77 | 33.42 | 97.27 | 31.31 | 97.54 | 30.30 | 97.22 | 32.40 |
| Mahalanobis$_{ensemble}$ | YES | 37.91 | **90.30** | 94.09 | 45.50 | 61.56 | 61.16 | 61.78 | 61.13 | 53.29 | **71.68** | 61.73 | 65.95 |
| ODIN | NO | 85.29 | 61.87 | 88.93 | 58.27 | 89.56 | 58.57 | 89.54 | 59.06 | 84.33 | 64.82 | 87.53 | 60.52 |
| ReAct | YES | 82.37 | 60.08 | 87.19 | 61.31 | 85.39 | 65.80 | 87.74 | 62.86 | 86.99 | 59.98 | 85.94 | 62.01 |
| SCALE | NO | 84.84 | 59.26 | 89.66 | 58.25 | 90.05 | 60.75 | 91.51 | 57.95 | 88.16 | 58.76 | 88.84 | 58.99 |
| ASH | NO | 92.41 | 58.90 | 91.56 | 55.26 | 93.51 | 52.40 | 93.88 | 51.33 | 88.62 | 62.12 | 92.00 | 56.00 |
| RTL | NO | 80.17 | 59.38 | 86.96 | 57.54 | 83.75 | 62.21 | 88.03 | 56.60 | 83.18 | 57.72 | 84.42 | 58.69 |
| NNGuide | YES | 86.45 | 59.03 | 87.05 | 59.69 | 86.04 | 64.58 | 88.36 | 61.35 | 84.53 | 61.21 | 86.49 | 61.17 |
| CoRP | YES | 88.45 | 59.62 | 89.92 | 56.64 | 87.28 | 63.92 | 89.22 | 59.97 | 86.69 | 59.10 | 88.31 | 59.85 |
| MDS++ | YES | 77.22 | 67.31 | 83.05 | 61.79 | 78.16 | 68.41 | 80.03 | 65.75 | 72.31 | 68.01 | 78.15 | 66.25 |
| RMDS | YES | 76.29 | 65.91 | 87.26 | 60.66 | 84.90 | 65.54 | 87.27 | 61.98 | 77.44 | 63.73 | 82.63 | 63.56 |
| RMDS++ | YES | 75.35 | 65.59 | 86.15 | 61.22 | 83.89 | 66.16 | 86.10 | 62.95 | 77.02 | 64.35 | 81.30 | 64.05 |
| *DART* | NO | **36.44** | 71.92 | **67.29** | 57.33 | **25.08** | **80.93** | **31.67** | **75.50** | 60.90 | 57.30 | **44.28** | **68.60** |
| | | Clean | | | | | | | | | | | |
| Method | Training dist. informed | SVHN | | Places365 | | LSUN | | iSUN | | Textures | | Average | |
| | | FPR95 ↓ | AUROC ↑ | FPR95 ↓ | AUROC ↑ | FPR95 ↓ | AUROC ↑ | FPR95 ↓ | AUROC ↑ | FPR95 ↓ | AUROC ↑ | FPR95 ↓ | AUROC ↑ |
| MSP | NO | 62.06 | 82.50 | 78.47 | 74.69 | 70.48 | 81.33 | 74.85 | 78.79 | 65.92 | 81.63 | 70.36 | 79.79 |
| Energy | NO | 41.79 | 89.13 | 73.81 | 77.71 | 61.09 | 86.02 | 66.13 | 83.60 | 51.04 | 87.68 | 58.77 | 84.83 |
| Max logit | NO | 44.96 | 88.65 | 74.40 | 77.59 | 61.78 | 85.80 | 67.05 | 83.33 | 53.15 | 87.33 | 60.27 | 84.54 |
| GradNorm | NO | 70.09 | 81.30 | 83.25 | 69.27 | 85.99 | 72.71 | 87.16 | 70.48 | 65.14 | 82.44 | 78.33 | 75.24 |
| ViM | YES | 47.34 | 88.18 | 71.80 | 78.03 | 57.30 | 87.18 | 63.52 | 84.49 | 51.37 | 87.41 | 58.27 | 85.06 |
| KNN | YES | 60.23 | 83.70 | 79.73 | 69.78 | 65.70 | 83.28 | 72.82 | 77.93 | 58.29 | 83.19 | 67.35 | 79.58 |
| Mahalanobis$_{single}$ | YES | 97.62 | 31.26 | 94.24 | 47.56 | 96.96 | 31.19 | 97.67 | 27.57 | 95.40 | 40.26 | 96.38 | 35.57 |
| Mahalanobis$_{ensemble}$ | YES | 13.93 | 97.14 | 94.58 | 55.26 | 3.03 | 99.06 | 3.93 | 98.93 | **30.44** | **91.63** | 29.18 | 88.40 |
| ODIN | NO | 75.65 | 75.47 | 87.48 | 64.34 | 84.41 | 68.70 | 83.64 | 69.79 | 70.07 | 77.78 | 80.25 | 71.22 |
| ReAct | YES | 40.18 | 89.58 | 72.80 | 78.53 | 57.23 | 84.84 | 62.69 | 84.64 | 51.40 | 87.38 | 56.86 | 81.79 |
| SCALE | NO | 44.17 | 88.43 | 73.71 | 77.68 | 67.15 | 83.85 | 71.51 | 81.28 | 52.19 | 87.07 | 61.75 | 83.66 |
| ASH | NO | 77.25 | 79.20 | 83.45 | 69.11 | 86.72 | 71.52 | 88.41 | 69.02 | 70.01 | 81.14 | 81.17 | 74.00 |
| RTL | NO | 26.70 | 89.94 | 67.55 | 77.48 | 27.65 | 93.41 | 39.20 | 89.62 | 39.88 | 88.30 | 40.20 | 87.75 |
| NNGuide | YES | 47.09 | 87.22 | 76.40 | 74.95 | 60.59 | 85.51 | 66.31 | 82.58 | 51.10 | 86.91 | 60.30 | 83.43 |
| CoRP | YES | 61.85 | 85.00 | 80.76 | 73.19 | 64.51 | 85.81 | 68.88 | 82.48 | 60.42 | 85.68 | 67.28 | 82.43 |
| MDS++ | YES | 39.14 | 88.84 | 66.54 | 78.57 | 42.39 | 90.18 | 48.78 | 87.41 | 36.66 | 90.59 | 46.70 | 87.12 |
| RMDS | YES | 36.21 | 89.32 | 62.15 | 80.43 | 47.61 | 88.92 | 55.13 | 85.97 | 38.81 | 90.00 | 47.98 | 86.93 |
| RMDS++ | YES | 36.30 | 89.09 | 62.64 | 80.08 | 48.16 | 88.67 | 55.21 | 85.72 | 38.26 | 89.98 | 48.11 | 86.71 |
| *DART* | NO | **3.24** | **99.23** | 62.77 | **81.73** | **0.85** | **99.80** | **1.04** | **99.74** | 35.00 | 91.04 | **20.58** | **94.31** |

## C.4 FULL CIFAR-100 RESULTS WITH SWIN-TINY

Table 8: OOD detection performance comparison with Swin-Tiny. We evaluate with CIFAR-100 ID, CIFAR-100-C csID and the corresponding OODs. (Best: **bolded**, Second-best: underlined)

| Covariate Shifted | | | | | | | | | | | | |
|---|---|---|---|---|---|---|---|---|---|---|---|---|
| Method | Training dist. informed | SVHN-C | | Places365-C | | LSUN-C | | iSUN-C | | Textures-C | | Average | |
| | | FPR95 ↓ | AUROC ↑ | FPR95 ↓ | AUROC ↑ | FPR95 ↓ | AUROC ↑ | FPR95 ↓ | AUROC ↑ | FPR95 ↓ | AUROC ↑ | FPR95 ↓ | AUROC ↑ |
| MSP | NO | 87.48 | 58.92 | 87.20 | 63.29 | 82.95 | 67.03 | 85.14 | 64.75 | 84.33 | 62.97 | 85.42 | 63.39 |
| Energy | NO | 81.88 | 67.29 | 86.20 | 64.20 | 78.30 | 71.63 | 81.65 | 68.08 | 73.47 | 70.53 | 80.30 | 68.35 |
| Max logit | NO | 83.72 | 66.10 | 86.30 | 64.38 | 79.40 | 71.16 | 82.44 | 67.79 | 77.63 | 69.22 | 81.90 | 67.73 |
| GradNorm | NO | 82.38 | 68.38 | 89.90 | 62.00 | 80.76 | 72.92 | 79.68 | 73.52 | 61.12 | **80.40** | 78.77 | 71.44 |
| ViM | YES | 95.84 | 51.26 | 94.98 | 50.17 | 97.60 | 46.77 | 98.04 | 43.00 | 98.22 | 37.61 | 96.94 | 45.76 |
| KNN | YES | 82.69 | 68.02 | 87.62 | 60.94 | 82.50 | 67.52 | 85.40 | 62.77 | 83.06 | 65.31 | 84.25 | 64.91 |
| Mahalanobis$_{single}$ | YES | 98.23 | 38.79 | 97.74 | 41.80 | 99.32 | 32.79 | 99.36 | 30.22 | 99.13 | 21.86 | 98.76 | 33.09 |
| Mahalanobis$_{ensemble}$ | YES | 97.17 | 44.72 | 97.94 | 39.09 | 99.35 | 31.32 | 99.38 | 29.27 | 99.14 | 24.34 | 98.60 | 33.75 |
| ODIN | NO | 80.78 | 66.20 | 89.74 | 59.86 | 91.86 | 58.65 | 91.08 | 59.38 | 70.96 | 74.06 | 84.88 | 63.63 |
| ReAct | YES | 78.42 | 69.44 | 85.84 | 65.20 | 76.79 | 73.35 | 78.67 | 71.44 | 64.09 | 76.25 | 76.76 | 71.14 |
| SCALE | NO | 79.13 | 69.57 | 86.21 | 65.09 | 76.71 | 73.93 | 78.22 | 72.28 | 63.15 | 76.96 | 76.68 | 71.57 |
| ASH | NO | 78.65 | 70.33 | 87.57 | 63.95 | 77.48 | 73.87 | 78.64 | 72.06 | 62.42 | 76.76 | 76.95 | 71.39 |
| RTL | NO | 86.72 | 57.38 | 86.72 | 62.68 | 79.41 | 69.61 | 83.11 | 65.84 | 82.61 | 62.58 | 83.71 | 63.62 |
| NNGuide | YES | 78.56 | 72.74 | 84.64 | 67.62 | 73.65 | 76.11 | 76.30 | 73.44 | 69.52 | 76.99 | 76.53 | 73.38 |
| CoRP | YES | 84.83 | 67.05 | 89.32 | 59.20 | 85.16 | 65.79 | 86.98 | 61.99 | 84.56 | 64.22 | 86.17 | 63.65 |
| MDS++ | YES | 72.36 | 75.29 | 88.15 | 60.55 | 78.41 | 69.66 | 79.70 | 66.70 | 61.44 | 76.79 | 76.01 | 69.80 |
| RMDS | YES | 91.82 | 59.47 | 91.80 | 56.76 | 89.66 | 61.30 | 91.74 | 57.40 | 91.61 | 55.75 | 91.33 | 58.14 |
| RMDS++ | YES | 89.90 | 63.59 | 90.80 | 58.82 | 85.84 | 64.50 | 88.39 | 60.72 | 85.74 | 61.41 | 88.13 | 61.81 |
| *DART* | NO | **44.90** | 70.99 | **57.62** | 75.48 | **14.28** | **95.53** | **24.97** | **87.74** | 52.74 | 76.66 | **38.90** | **81.28** |

| Clean | | | | | | | | | | | | |
|---|---|---|---|---|---|---|---|---|---|---|---|---|
| Method | Training dist. informed | SVHN | | Places365 | | LSUN | | iSUN | | Textures | | Average | |
| | | FPR95 ↓ | AUROC ↑ | FPR95 ↓ | AUROC ↑ | FPR95 ↓ | AUROC ↑ | FPR95 ↓ | AUROC ↑ | FPR95 ↓ | AUROC ↑ | FPR95 ↓ | AUROC ↑ |
| MSP | NO | 59.06 | 86.09 | 69.83 | 79.17 | 58.09 | 85.76 | 62.11 | 83.42 | 51.72 | 87.13 | 60.16 | 84.31 |
| Energy | NO | 35.88 | 91.83 | 58.64 | 83.16 | 38.61 | 91.40 | 44.35 | 88.95 | 27.80 | 93.67 | 41.06 | 89.80 |
| Max logit | NO | 37.54 | 91.54 | 58.73 | 83.04 | 39.41 | 91.10 | 45.02 | 88.66 | 29.86 | 93.28 | 42.11 | 89.52 |
| GradNorm | NO | 85.31 | 61.74 | 83.37 | 67.78 | 70.06 | 80.46 | 68.41 | 80.33 | 51.02 | 82.32 | 71.63 | 74.53 |
| ViM | YES | 83.71 | 80.08 | 93.84 | 62.22 | 96.72 | 63.50 | 97.19 | 59.52 | 93.54 | 65.96 | 93.00 | 66.26 |
| KNN | YES | 31.60 | 93.26 | 66.75 | 80.31 | 46.95 | 89.67 | 53.14 | 86.22 | 31.23 | 93.09 | 45.93 | 88.51 |
| Mahalanobis$_{single}$ | YES | 97.07 | 59.75 | 98.46 | 43.08 | 99.67 | 35.81 | 99.72 | 33.62 | 98.43 | 33.27 | 98.67 | 41.11 |
| Mahalanobis$_{ensemble}$ | YES | 96.51 | 64.83 | 98.40 | 43.33 | 99.57 | 38.31 | 99.65 | 36.58 | 97.99 | 44.61 | 98.42 | 45.53 |
| ODIN | NO | 89.89 | 67.25 | 95.52 | 59.39 | 97.62 | 57.25 | 97.04 | 57.87 | 75.93 | 76.50 | 91.20 | 63.65 |
| ReAct | YES | 42.84 | 90.56 | 58.78 | 82.17 | 38.80 | 91.42 | 42.60 | 89.57 | 25.89 | 93.83 | 41.78 | 89.51 |
| SCALE | NO | 49.51 | 87.94 | 62.13 | 81.02 | 40.86 | 91.00 | 44.03 | 89.32 | 27.67 | 93.21 | 44.84 | 88.50 |
| ASH | NO | 54.51 | 85.90 | 67.96 | 79.09 | 46.42 | 89.69 | 48.93 | 88.02 | 32.31 | 92.14 | 50.03 | 86.97 |
| RTL | NO | 43.25 | 87.35 | 70.99 | 74.76 | 44.41 | 87.35 | 51.83 | 84.40 | 44.03 | 87.30 | 50.90 | 84.23 |
| NNGuide | YES | 37.51 | 91.82 | **56.58** | 84.28 | 33.61 | 93.04 | 38.34 | 91.40 | 24.15 | 94.75 | 38.04 | 91.06 |
| CoRP | YES | 41.14 | 92.32 | 68.66 | 80.55 | 50.94 | 89.81 | 55.48 | 87.22 | 35.63 | 92.77 | 50.37 | 88.53 |
| MDS++ | YES | 30.97 | 93.67 | 56.68 | 83.95 | 33.31 | 92.98 | 39.00 | 90.82 | 16.43 | 96.28 | 35.28 | 91.54 |
| RMDS | YES | 52.84 | 89.57 | 64.26 | 82.37 | 52.70 | 88.97 | 59.42 | 86.09 | 40.78 | 91.23 | 54.00 | 87.65 |
| RMDS++ | YES | 47.14 | 90.76 | 58.64 | 83.56 | 44.55 | 90.16 | 50.92 | 87.55 | 33.78 | 92.48 | 47.01 | 88.90 |
| *DART* | NO | **7.27** | **97.88** | 77.97 | 64.13 | **6.31** | **98.52** | **6.68** | **98.49** | 61.06 | 81.14 | **31.86** | 88.03 |

## C.5 Full ImageNet Results with ResNet-50

Table 9: OOD detection performance comparison with ResNet-50 on ImageNet-based benchmark. Results on covariated shifted dataset are the average of all 15 corruptions with severity level 5. (Best: **bolded**, Second-best: underlined)

| | Covariate Shifted | | | | | | | | | | | | |
|---|---|---|---|---|---|---|---|---|---|---|---|---|---|
| Method | Training dist. informed | ImageNet-O-C | | Places-C | | SUN-C | | iNaturalist-C | | Textures-C | | Average | |
| | | FPR95 ↓ | AUROC ↑ | FPR95 ↓ | AUROC ↑ | FPR95 ↓ | AUROC ↑ | FPR95 ↓ | AUROC ↑ | FPR95 ↓ | AUROC ↑ | FPR95 ↓ | AUROC ↑ |
| MSP | NO | 86.14 | 55.60 | 86.79 | 63.17 | 84.84 | 65.27 | 77.89 | 69.62 | 88.52 | 54.85 | 84.84 | 61.70 |
| Energy | NO | 82.16 | 58.92 | 89.03 | 61.29 | 87.91 | 63.58 | 86.44 | 64.52 | 88.05 | 55.85 | 86.72 | 60.83 |
| Max logit | NO | 83.88 | 57.81 | 87.57 | 63.00 | 85.79 | 65.42 | 81.49 | 68.07 | 88.02 | 55.87 | 85.35 | 62.03 |
| GradNorm | NO | 76.86 | 64.58 | 70.84 | 77.06 | 65.23 | 80.51 | 51.36 | 85.62 | 67.11 | 74.14 | 66.28 | 76.38 |
| ViM | YES | 95.64 | 38.92 | 99.50 | 17.64 | 99.65 | 14.90 | 99.94 | 8.60 | 96.94 | 25.14 | 98.33 | 21.04 |
| KNN | YES | 83.81 | 63.63 | 90.93 | 55.23 | 91.12 | 56.45 | 95.20 | 45.36 | 61.31 | 73.33 | 84.47 | 58.80 |
| MDS$_{single}$ | YES | 95.14 | 38.80 | 99.52 | 18.75 | 99.67 | 15.95 | 99.93 | 9.60 | 96.20 | 27.69 | 98.09 | 22.16 |
| MDS$_{ensemble}$ | YES | 79.71 | 62.76 | 92.44 | 42.67 | 92.17 | 41.57 | 94.26 | 33.49 | 64.44 | 62.83 | 84.60 | 48.66 |
| ODIN | NO | 76.69 | 68.04 | 25.14 | 92.96 | 22.70 | 93.69 | 27.84 | 92.22 | 38.11 | 85.81 | 38.10 | 86.54 |
| ReAct | YES | 82.24 | 59.09 | 84.47 | 67.56 | 82.92 | 69.76 | 80.76 | 70.72 | 84.85 | 59.33 | 83.05 | 65.29 |
| SCALE | NO | 79.10 | 63.07 | 79.87 | 70.76 | 76.10 | 74.05 | 63.83 | 79.72 | 75.88 | 68.39 | 74.96 | 71.20 |
| ASH | NO | 80.51 | 61.30 | 87.83 | 63.20 | 86.48 | 65.88 | 82.64 | 68.22 | 84.77 | 60.05 | 84.45 | 63.73 |
| RTL | NO | 82.27 | 57.85 | 76.94 | 71.71 | 73.29 | 74.47 | 64.06 | 78.47 | 81.14 | 59.14 | 75.54 | 68.33 |
| NNGuide | YES | 73.91 | 67.80 | 68.11 | 77.74 | 62.29 | 81.42 | 50.54 | 85.23 | 57.13 | 78.16 | 62.40 | 78.11 |
| CoRP | YES | 75.89 | 73.38 | 75.07 | 73.16 | 71.42 | 76.37 | 67.32 | 78.15 | 44.54 | 86.27 | 66.85 | 77.47 |
| MDS++ | YES | **63.76** | **77.47** | 81.50 | 66.32 | 79.47 | 69.03 | 66.54 | 77.67 | 36.03 | **88.92** | 65.46 | 75.88 |
| RMDS | YES | 90.38 | 50.24 | 95.39 | 44.28 | 95.88 | 42.31 | 96.52 | 45.83 | 85.80 | 50.46 | 92.79 | 46.62 |
| RMDS++ | YES | 77.40 | 64.63 | 86.31 | 58.47 | 85.77 | 58.86 | 78.16 | 68.26 | 61.45 | 73.34 | 77.82 | 64.71 |
| *DART* | NO | 73.52 | 58.78 | **2.46** | **99.34** | **1.06** | **99.59** | **1.18** | **99.61** | 23.75 | 79.75 | **20.39** | **87.41** |
| | Clean | | | | | | | | | | | | |
| Method | Training dist. informed | ImageNet-O | | Places | | SUN | | iNaturalist | | Textures | | Average | |
| | | FPR95 ↓ | AUROC ↑ | FPR95 ↓ | AUROC ↑ | FPR95 ↓ | AUROC ↑ | FPR95 ↓ | AUROC ↑ | FPR95 ↓ | AUROC ↑ | FPR95 ↓ | AUROC ↑ |
| MSP | NO | 64.81 | 75.42 | 56.33 | 85.09 | 53.18 | 85.97 | 36.95 | 91.68 | 53.73 | 83.95 | 53.00 | 84.42 |
| Energy | NO | 55.18 | 82.01 | 42.09 | 89.64 | 34.38 | 91.54 | 25.30 | 94.38 | 33.96 | 90.49 | 38.18 | 89.61 |
| Max logit | NO | 55.09 | 81.85 | 42.79 | 78.62 | 35.50 | 91.39 | 23.88 | 95.05 | 35.07 | 90.28 | 38.47 | 87.44 |
| GradNorm | NO | 55.97 | 78.38 | 47.85 | 87.45 | 42.19 | 88.60 | 24.01 | 94.34 | 42.17 | 86.85 | 42.44 | 87.12 |
| ViM | YES | 56.20 | 77.62 | 42.62 | 86.49 | 31.60 | 90.55 | 21.81 | 94.30 | 33.31 | 89.08 | 37.11 | 87.61 |
| KNN | YES | 11.09 | 97.01 | 72.62 | 83.16 | 72.49 | 84.40 | 82.54 | 79.21 | 16.31 | 96.53 | 51.01 | 88.06 |
| MDS$_{single}$ | YES | 11.10 | 95.67 | 54.71 | 87.07 | 45.46 | 90.53 | 78.67 | 76.90 | 19.89 | 94.97 | 41.97 | 89.03 |
| MDS$_{ensemble}$ | YES | 30.76 | 88.27 | 95.87 | 62.81 | 95.47 | 62.19 | 97.37 | 50.94 | 49.44 | 86.03 | 73.78 | 70.05 |
| ODIN | NO | 18.70 | 94.49 | 94.09 | 64.38 | 93.09 | 64.67 | 96.72 | 51.10 | 29.60 | 91.41 | 66.44 | 73.21 |
| ReAct | YES | 16.84 | 95.44 | 36.62 | 89.54 | 34.89 | 89.80 | 38.99 | 88.92 | 35.50 | 88.30 | 32.57 | 90.40 |
| SCALE | NO | 50.55 | 84.83 | 37.29 | 91.44 | 30.29 | 93.06 | 17.02 | 96.40 | 31.67 | 92.22 | 33.36 | 91.59 |
| ASH | NO | 42.77 | 87.82 | 32.37 | 92.47 | 24.55 | 94.22 | **10.87** | **97.60** | 19.80 | 94.77 | 26.07 | 93.38 |
| RTL | NO | 56.98 | 75.34 | 42.41 | 86.56 | 36.48 | 87.81 | 23.85 | 91.29 | 38.71 | 85.14 | 39.69 | 85.23 |
| NNGuide | YES | 38.59 | 88.32 | **28.59** | **93.10** | 20.02 | **95.18** | 17.82 | 96.20 | 21.11 | 94.17 | 25.23 | **93.39** |
| CoRP | YES | 30.01 | 89.25 | 64.00 | 84.28 | 55.98 | 87.82 | 83.16 | 73.97 | 23.66 | 94.53 | 51.36 | 85.97 |
| MDS++ | YES | 14.92 | 97.16 | 61.42 | 84.92 | 49.94 | 88.97 | 33.84 | 93.72 | **1.69** | **99.49** | 32.36 | 92.85 |
| RMDS | YES | 61.88 | 88.56 | 87.73 | 79.15 | 88.16 | 80.24 | 65.98 | 90.30 | 51.95 | 88.58 | 71.14 | 84.97 |
| RMDS++ | YES | 56.42 | 86.76 | 75.65 | 82.07 | 73.61 | 83.96 | 36.83 | 93.50 | 29.65 | 91.97 | 54.43 | 87.65 |
| *DART* | NO | **0.60** | **99.83** | 33.90 | 89.31 | **15.90** | 95.77 | 14.29 | 96.46 | 23.68 | 94.78 | **17.67** | **95.23** |

## D Effect of EMA α

We employ an exponential moving average (EMA) to update the prototype, where the smoothing coefficient $\alpha$ is treated as an only hyperparameter of *DART*. To examine the sensitivity of our method to this hyperparameter, we conduct an ablation study on the CIFAR-100-C vs. SVHN-C and ImageNet-C vs. iNaturalist-C benchmark, and report the results as AUROC(%) in Table **??**. Results in Table 10 demonstrates that our method is robust to the choice of the EMA coefficient $\alpha$.

Table 10: The effect of EMA $\alpha$ on the performance of *DART*

| $\alpha$ | 0.5 | 0.6 | 0.7 | 0.8 | 0.9 | std. |
|---|---|---|---|---|---|---|
| CIFAR-100-C vs. SVHN-C | 78.79 | 78.68 | 78.84 | 78.58 | 78.11 | ± 0.29 |
| ImageNet-C vs. iNaturalist-C | 93.18 | 93.18 | 93.18 | 93.18 | 93.19 | ± 0.00 |

# E   EXTENDED RDS DISTRIBUTION VISUALIZATIONS

In the main paper, we present representative visualizations of RDS distributions for a subset of corruption types, to illustrate how ID and OOD samples are separated across different feature levels. These visualizations support our observation that the most discriminative layer can vary depending on the type of corruption. For completeness, we provide the full set of visualizations covering all 15 corruption types in this appendix.

## E.1   RDS DISTRIBUTION VISUALIZATIONS

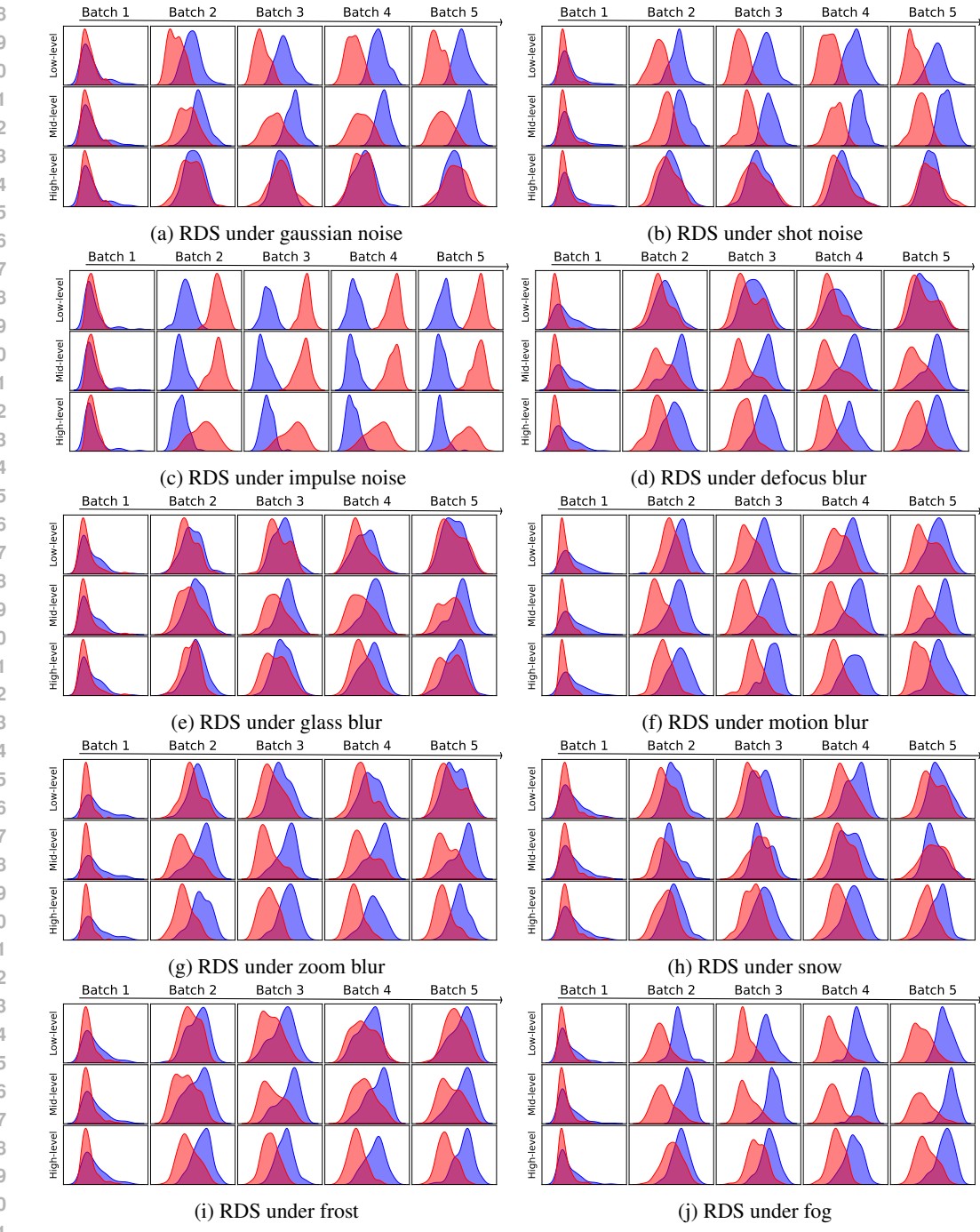

(a) RDS under gaussian noise

(b) RDS under shot noise

(c) RDS under impulse noise

(d) RDS under defocus blur

(e) RDS under glass blur

(f) RDS under motion blur

(g) RDS under zoom blur

(h) RDS under snow

(i) RDS under frost

(j) RDS under fog

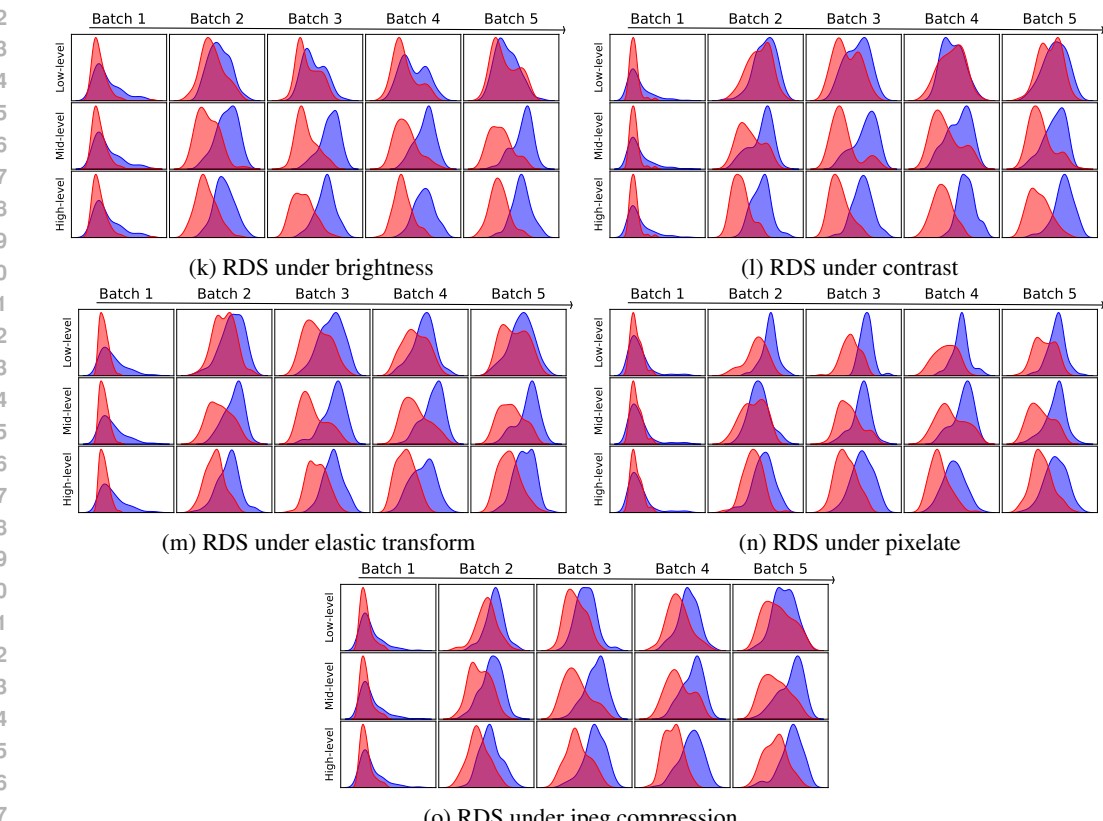

Figure 8: Full corruptions visualizations for RDS distributions. Each plot shows at different network depths (low, mid, high-level) through five sequential batches. The visualizations reveal how different corruption types affect RDS separability at specific network layers.

# F    THEORETICAL ANALYSIS

In this section, we provide a mathematical derivation of the feature amplification phenomenon observed in OOD data. We demonstrate how the mismatch between the frozen statistics of Batch Normalization (BN) derived from ID data and the statistics of OOD data leads to an explosion in feature magnitude for specific units.

## F.1    SETUP AND DEFINITIONS

Consider a specific channel (or neuron) index $k$ in a hidden layer. Let $x$ denote the input to this layer (or the output of the previous layer). We define the pre-activation feature $Z_k$ as a random variable:

$$Z_k(x) = w_k^\top x + b_k, \tag{8}$$

where $w_k$ and $b_k$ are the weight vector and bias for unit $k$, respectively.

## F.2    THE SILENT UNIT ASSUMPTION

We focus on a specific type of unit, which we term a *Silent Unit*. This unit typically corresponds to a feature that is either irrelevant for classifying In-Distribution (ID) data or represents a direction orthogonal to the ID manifold. Consequently, its activation on ID data is minimal and stable.

Mathematically, let $\mathcal{D}_{ID}$ be the ID dataset and $P_{ID}$ be the underlying ID distribution. We assume the variance of $Z_k$ over $\mathcal{D}_{ID}$ is very small and of the same order as the BN stability constant $\epsilon$ (vanishing but non-zero variance). The running statistics, $\mu_{ID}$ and $\sigma_{ID}^2$, which are frozen after training, are

given by:

$$\mu_{ID} = \mathbb{E}_{x \sim P_{ID}}[Z_k(x)], \tag{9}$$

$$\sigma_{ID}^2 = \text{Var}_{x \sim P_{ID}}[Z_k(x)] = \mathcal{O}(\epsilon). \tag{10}$$

### F.3 Inference Logic of Batch Normalization

During the inference phase, BN normalizes the input using the frozen ID statistics. The normalized output $\hat{Z}_k(x)$ is computed as:

$$\hat{Z}_k(x) = \gamma_k \cdot \frac{Z_k(x) - \mu_{ID}}{\sqrt{\sigma_{ID}^2 + \epsilon}} + \beta_k, \tag{11}$$

where $\gamma_k$ and $\beta_k$ are learnable affine parameters, and $\epsilon$ is a small constant for numerical stability (e.g., $10^{-5}$).

Crucially, due to the silent unit assumption ($\sigma_{ID}^2 = \mathcal{O}(\epsilon)$), the scaling factor $\lambda_k$ satisfies:

$$\lambda_k \triangleq \frac{1}{\sqrt{\sigma_{ID}^2 + \epsilon}} = \Theta\left(\frac{1}{\sqrt{\epsilon}}\right). \tag{12}$$

For typical choices such as $\epsilon = 10^{-5}$, this corresponds to a large constant amplification.

### F.4 Feature Amplification on OOD Data

Now, consider a bounded OOD input $x_{OOD} \sim P_{OOD}$. Since $x_{OOD}$ does not share the specific semantic structure of ID data that suppresses the activation of unit $k$, $Z_k(x_{OOD})$ follows a distribution determined by random projection in the high-dimensional feature space.

Let $\Delta$ be the deviation of the OOD pre-activation from the ID mean:

$$\Delta = |Z_k(x_{OOD}) - \mu_{ID}|. \tag{13}$$

Unlike ID data, OOD data is not concentrated around $\mu_{ID}$, implying that $\Delta$ is a non-negligible value of order $\mathcal{O}(1)$ with respect to $\epsilon$.

We now compare the magnitude of the normalized feature $|\hat{Z}_k|$ for ID and OOD inputs (assuming $\beta_k \approx 0$ for simplicity):

**Case 1: ID Data.** For $x \sim P_{ID}$, the deviation of $Z_k(x)$ from $\mu_{ID}$ is controlled by the standard deviation:

$$|Z_k(x_{ID}) - \mu_{ID}| = \mathcal{O}(\sigma_{ID}) = \mathcal{O}(\sqrt{\epsilon}). \tag{14}$$

Using $\sigma_{ID}^2 = \mathcal{O}(\epsilon)$, the denominator satisfies $\sqrt{\sigma_{ID}^2 + \epsilon} = \Theta(\sqrt{\epsilon})$, and thus the normalized activation remains stable:

$$|\hat{Z}_k(x_{ID})| = |\gamma_k| \cdot \frac{\mathcal{O}(\sqrt{\epsilon})}{\Theta(\sqrt{\epsilon})} = \Theta(|\gamma_k|) = \mathcal{O}(1). \tag{15}$$

**Case 2: OOD Data.** For $x \sim P_{OOD}$, the deviation $\Delta$ is constant with respect to $\epsilon$. However, the denominator remains of order $\sqrt{\epsilon}$ due to the silent unit statistics. This leads to an explosion in magnitude:

$$|\hat{Z}_k(x_{OOD})| \approx |\gamma_k| \cdot \frac{\Delta}{\sqrt{\sigma_{ID}^2 + \epsilon}} = \Omega\left(\frac{1}{\sqrt{\epsilon}}\right). \tag{16}$$

Consequently, for sufficiently small $\epsilon$, we obtain

$$|\hat{Z}_k(x_{OOD})| = \Omega\left(\frac{1}{\sqrt{\epsilon}}\right), \tag{17}$$

which is much larger than the ID magnitude (e.g., two orders of magnitude larger when $\epsilon = 10^{-5}$). This amplified signal passes through the ReLU activation function (if positive), resulting in abnormally high activation values for OOD data compared to ID data.

**Remark.** *This theoretical framework elucidates the phenomenon observed in Fig 3, where specific units exhibit significantly higher activation values for OOD samples. As derived above, the variance mismatch amplifies the OOD signals in silent units, thereby facilitating the clear separation between ID and OOD distributions in the feature space.*

### F.5 VALIDITY UNDER SHARED COVARIATE SHIFT

We next argue that the above amplification mechanism remains valid even when ID and OOD inputs undergo covariate shift (e.g., changes in weather, illumination, or sensor characteristics).

Let $e$ denote an environment configuration that induces a covariate shift via a transformation $T_e$ on the input space, and assume that this transformation is applied identically to ID and OOD samples:

$$x_e = T_e(x), \quad x \sim P_{ID} \text{ or } x \sim P_{OOD}. \tag{18}$$

The pre-activation of unit $k$ in environment $e$ can then be written as

$$Z_k^{(e)}(x) = w_k^\top x_e + b_k = w_k^\top T_e(x) + b_k = Z_k(x) + \delta_k(e, x), \tag{19}$$

where $\delta_k(e, x) \triangleq w_k^\top(T_e(x) - x)$ captures the effect of the covariate shift along the $k$-th direction.

We assume that the covariate shift transformation $T_e$ is bounded in the input space, i.e., there exists a constant $B_{\text{env}}$ such that

$$\|T_e(x) - x\|_2 \le B_{\text{env}} \quad \text{for all } x \text{ in the support of } P_{ID} \text{ and } P_{OOD}. \tag{20}$$

Then the induced perturbation along unit $k$ satisfies

$$|\delta_k(e, x)| = |w_k^\top(T_e(x) - x)| \le \|w_k\|_2 \|T_e(x) - x\|_2 \le \|w_k\|_2 B_{\text{env}}. \tag{21}$$

Thus we can choose

$$C_{\text{env}} \triangleq \|w_k\|_2 B_{\text{env}}, \tag{22}$$

which is a finite constant independent of the BN hyperparameter $\epsilon$.

Importantly, BN still uses the frozen statistics $(\mu_{ID}, \sigma_{ID}^2)$ computed from the original ID distribution (before the shift). Thus, the normalization scale

$$\sqrt{\sigma_{ID}^2 + \epsilon} = \Theta(\sqrt{\epsilon}) \tag{23}$$

is unchanged.

**ID under covariate shift.** For $x \sim P_{ID}$, we have

$$|Z_k^{(e)}(x_{ID}) - \mu_{ID}| = |Z_k(x_{ID}) - \mu_{ID} + \delta_k(e, x_{ID})| \le |Z_k(x_{ID}) - \mu_{ID}| + |\delta_k(e, x_{ID})|. \tag{24}$$

Under the silent-unit assumption, $|Z_k(x_{ID}) - \mu_{ID}| = \mathcal{O}(\sqrt{\epsilon})$, while the covariate shift contribution is bounded by $C_{\text{env}}$. Hence

$$|Z_k^{(e)}(x_{ID}) - \mu_{ID}| \le \mathcal{O}(\sqrt{\epsilon}) + C_{\text{env}}. \tag{25}$$

Dividing by $\sqrt{\sigma_{ID}^2 + \epsilon} = \Theta(\sqrt{\epsilon})$ yields a bounded normalized activation:

$$|\hat{Z}_k^{(e)}(x_{ID})| = |\gamma_k| \cdot \frac{|Z_k^{(e)}(x_{ID}) - \mu_{ID}|}{\sqrt{\sigma_{ID}^2 + \epsilon}}, \tag{26}$$

which remains a finite constant determined by $(\gamma_k, C_{\text{env}}, \epsilon)$ and does not diverge with the OOD amplification discussed below.

**OOD under covariate shift.** For $x \sim P_{OOD}$, recall that in the original environment we have

$$\Delta_0 \triangleq |Z_k(x_{OOD}) - \mu_{ID}| = \mathcal{O}(1), \tag{27}$$

reflecting the fact that OOD samples are not concentrated around the ID mean. Under the shared covariate shift, we obtain

$$|Z_k^{(e)}(x_{OOD}) - \mu_{ID}| = |Z_k(x_{OOD}) - \mu_{ID} + \delta_k(e, x_{OOD})| \ge \Delta_0 - |\delta_k(e, x_{OOD})| \ge \Delta_0 - C_{\text{env}}. \tag{28}$$

For bounded shifts with $C_{\text{env}} < \Delta_0$, the deviation remains of constant order, i.e.,

$$|Z_k^{(e)}(x_{OOD}) - \mu_{ID}| = \Omega(1). \tag{29}$$

After BN normalization, this yields

$$|\hat{Z}_k^{(e)}(x_{OOD})| = |\gamma_k| \cdot \frac{|Z_k^{(e)}(x_{OOD}) - \mu_{ID}|}{\sqrt{\sigma_{ID}^2 + \epsilon}} = \Omega\left(\frac{1}{\sqrt{\epsilon}}\right), \tag{30}$$

which is still much larger than the ID magnitude. Therefore, even when ID and OOD undergo the same covariate shift, the mismatch between frozen ID statistics and OOD activations, combined with the silent-unit scaling, continues to produce systematically amplified responses for OOD data.

**Remark on Layer Normalization.** The above derivation characterizes feature amplification under Batch Normalization, which relies on frozen ID statistics at test time. However, many modern architectures in our experimental setup (e.g., ViT, Swin-Transformer) primarily use Layer Normalization instead of Batch Normalization. It is therefore natural to ask whether a similar unit-wise OOD amplification effect can also arise under Layer Normalization. To clarify why this may plausibly occur, we briefly recall how LN affects the per-sample feature energy.

Let $h(x) \in \mathbb{R}^d$ be a feature vector with components $h_i(x)$. LN computes the sample mean and variance as

$$\mu(x) = \frac{1}{d} \sum_{i=1}^{d} h_i(x), \qquad \sigma^2(x) = \frac{1}{d} \sum_{i=1}^{d} \big( h_i(x) - \mu(x) \big)^2, \tag{31}$$

and produces normalized activations

$$\tilde{h}_i(x) = \frac{h_i(x) - \mu(x)}{\sqrt{\sigma^2(x) + \epsilon}}, \tag{32}$$

(optionally followed by an affine transform with $\gamma_i$ and $\beta_i$). Ignoring the small stability constant $\epsilon$ for clarity, we obtain

$$\frac{1}{d} \sum_{i=1}^{d} \tilde{h}_i(x)^2 = \frac{1}{d} \sum_{i=1}^{d} \frac{\big( h_i(x) - \mu(x) \big)^2}{\sigma^2(x)} = \frac{1}{\sigma^2(x)} \cdot \frac{1}{d} \sum_{i=1}^{d} \big( h_i(x) - \mu(x) \big)^2 = 1, \tag{33}$$

so that

$$\sum_{i=1}^{d} \tilde{h}_i(x)^2 = d. \tag{34}$$

Thus, LN enforces a fixed per-sample variance (and hence a fixed "energy" $\sum_i \tilde{h}_i^2$) and effectively redistributes this energy across feature dimensions for each input $x$. Here, by "energy" we simply refer to the squared $\ell_2$-norm (or variance) of the normalized feature vector, not to the energy-based OOD score (e.g., negative log-sum-exp of logits) commonly used in energy-based OOD detection.

While our formal analysis focuses on Batch Normalization, it is plausible that an analogous mechanism can operate under Layer Normalization. In particular, under LN the per-sample variance is fixed, and OOD inputs may induce large excursions along directions that remain nearly silent for ID data. In such cases, the fixed energy budget would be concentrated on these silent directions, potentially leading to much larger normalized activations on OOD samples in the corresponding units than on ID samples. A rigorous theoretical and empirical study of this LayerNorm case—for example, characterizing how per-sample variance redistribution interacts with silent directions in high-dimensional feature spaces—is an interesting direction that we leave for future work.

## G    CROSS-DOMAIN EVALUATION

In the main body of the paper, we reported results under evaluation settings where each test stream contains a single OOD dataset and a single type of covariate shift. In this appendix, we further evaluate our method in a more challenging regime with multiple OOD datasets and multiple covariate shifts.

### G.1    MIXED OOD

We believe our bounded OOD setting reflects realistic deployment scenarios where OOD inputs tend to concentrate within a limited semantic space due to observation boundaries of the data stream. However, to further demonstrate robustness beyond this assumption, we conducted additional experiments where two different OOD sources are mixed simultaneously during test time while the ID stream remains fixed. Concretely, at each time step, OOD samples are randomly drawn from two diverse sources rather than a single homogeneous distribution.

Results in Table 11 show that *DART* maintains the best detection performance even under this mixed-OOD scenario, indicating that our OOD prototype successfully finds the discriminative axis when OOD samples span multiple semantic categories. We observe performance degradation only in

Table 11: Mixed OOD evaluation

| Method | ImageNet-O + Places | | Places + SUN | | SUN + Textures | |
|---|---|---|---|---|---|---|
| | FPR@95TPR | AUROC | FPR@95TPR | AUROC | FPR@95TPR | AUROC |
| MSP | 48.43 | 83.70 | 48.65 | 84.26 | 43.65 | 85.52 |
| Energy | 51.56 | 78.42 | 44.24 | 82.57 | 38.31 | 85.65 |
| Max Logit | 48.14 | 80.20 | 42.85 | 83.49 | 37.29 | 86.24 |
| GradNorm | 86.49 | 43.25 | 77.26 | 57.44 | 67.54 | 66.80 |
| ODIN | 62.28 | 79.44 | 66.10 | 76.78 | 60.37 | 80.13 |
| SCALE | 47.62 | 80.09 | 40.36 | 84.39 | 33.92 | 87.81 |
| RTL | 46.82 | 81.46 | 44.77 | 82.94 | 40.64 | 83.98 |
| DART | **19.23** | **94.45** | **25.96** | **91.15** | **18.65** | **94.80** |

an extreme case where all five OOD sets are mixed simultaneously—a scenario that fundamentally violates our bounded OOD assumption. However, we believe such extreme mixing rarely occurs in practice, and our assumption holds within realistic deployment boundaries.

## G.2 CONTINUAL OOD

We additionally evaluate a more challenging scenario in which the type of OOD data itself changes abruptly over time while the ID stream is fixed. Concretely, we partition the test stream into several temporal segments. In every segment, ID samples are drawn from the same ID distribution, but OOD samples are drawn from a different OOD set in each temporal segment, and we switch the OOD source abruptly at segment boundaries without any prior knowledge of these switches.

Table 12: Continual OOD evaluation

| Method | ImageNet-O → Places | | Places → SUN | | SUN → Textures | |
|---|---|---|---|---|---|---|
| | FPR@95TPR | AUROC | FPR@95TPR | AUROC | FPR@95TPR | AUROC |
| MSP | 48.99 | 83.63 | 47.93 | 84.36 | 43.77 | 85.52 |
| Energy | 51.07 | 78.46 | 44.29 | 82.38 | 38.26 | 85.65 |
| Max Logit | 47.58 | 80.29 | 43.53 | 83.40 | 37.29 | 86.24 |
| GradNorm | 86.24 | 43.53 | 76.87 | 57.52 | 67.52 | 66.80 |
| ODIN | 62.14 | 79.40 | 66.44 | 76.84 | 60.41 | 80.11 |
| SCALE | 47.35 | 80.15 | 40.91 | 84.31 | 33.98 | 87.81 |
| RTL | 48.19 | 79.64 | 45.50 | 80.94 | 39.65 | 83.97 |
| DART | **12.05** | **95.49** | **23.40** | **93.51** | **18.80** | **94.81** |

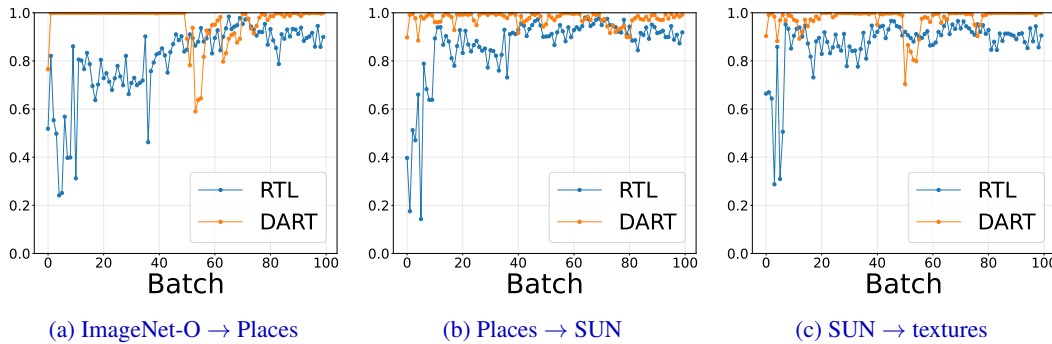

(a) ImageNet-O → Places     (b) Places → SUN     (c) SUN → textures

Figure 9: AUROC vs. batch

Results in Table 12 indicate that *DART* continues to exhibit stable OOD detection performance across segments, with only modest fluctuations when the OOD set changes, whereas non-adaptive

baselines suffer noticeable drops whenever a new OOD set appears. This suggests that our prototype-based tracking does not rely on a single globally fixed linear discriminative axis, but can adaptively re-estimate the relevant axis over time, even under abrupt semantic shifts in the OOD distribution.

### G.3 MIXED COVARIATE SHIFT

We conduct additional experiments where a test batch contains samples from multiple different covariate shifts. We designed two specific scenarios: a mixture of two shifts and a mixture of all 15 shifts.

In the setting with the mixture of two shifts, we evaluated settings where two different covariate shift types are mixed within a single batch. In this scenario, *DART* maintains robust performance, significantly outperforming all baseline methods by a large margin, as shown in Table 13.

Table 13: Performance comparison under mixed-shift scenarios

| Method | Original + Gaussian noise | | Gaussian noise + Snow | | Snow + JPEG compression | |
| | *FPR@95TPR* | *AUROC* | *FPR@95TPR* | *AUROC* | *FPR@95TPR* | *AUROC* |
|---|---|---|---|---|---|---|
| MSP | 82.63 | 67.43 | 85.01 | 62.49 | 83.07 | 68.22 |
| Energy | 72.01 | 76.77 | 83.91 | 69.82 | 89.72 | 66.83 |
| Max Logit | 77.19 | 73.34 | 85.40 | 67.10 | 85.73 | 68.66 |
| GradNorm | 54.72 | 83.51 | 65.33 | 81.94 | 72.16 | 79.49 |
| ODIN | 66.85 | 77.72 | 45.20 | 83.46 | 12.08 | 97.39 |
| SCALE | 66.99 | 78.21 | 80.27 | 71.68 | 79.15 | 73.94 |
| RTL | 79.84 | 67.73 | 81.92 | 65.32 | 79.86 | 71.65 |
| DART | **22.04** | **93.93** | **0.85** | **99.62** | **0.79** | **99.72** |

While unrealistic in real-world data accumulation, to push the limits of our method, we also test a more extreme scenario involving a mixture of all 15 covariate shifts and report in Table 14. In this challenging setting, the performance gap between *DART* and the baselines slightly narrows compared to the independent or 2-mixture scenarios. We attribute this slight reduction in the margin to the influence of a few specific shift types that are inherently difficult to separate linearly. However, even under this extreme condition, *DART* consistently maintains the top-ranking performance (Rank 1) among all compared methods.

Table 14: Performance comparison under the mixture of all 15 shifts (Extreme Case)

| Method | *FPR@95TPR* | *AUROC* |
|---|---|---|
| MSP | 86.64 | 62.71 |
| Energy | 86.27 | 63.02 |
| Max Logit | 86.46 | 63.67 |
| GradNorm | 69.55 | 77.45 |
| ODIN | 25.45 | 92.16 |
| SCALE | 78.20 | 71.19 |
| RTL | 81.27 | 65.63 |
| DART | **19.68** | **92.70** |

### G.4 CONTINUAL COVARIATE SHIFT

We additionally evaluate on streams where the covariate shift explicitly evolves over time. Concretely, we construct time-varying streams in which the corruption type changes (e.g., from "clean" to "gaussian noise", from "gaussian noise" to "snow", and from "snow" to "jpeg compression"), while samples within each segment remain temporally correlated. This reflects evolving acquisition conditions rather than a single static corruption. In Table 15, *DART* maintains strong OOD detection performance after each environment change.

Table 15: Continual covariate shift evaluation

| Method | Clean → Gaussian noise | | Gaussian noise → Snow | | Snow → JPEG compression | |
|---|---|---|---|---|---|---|
| | *FPR@95TPR* | *AUROC* | *FPR@95TPR* | *AUROC* | *FPR@95TPR* | *AUROC* |
| MSP | 68.33 | 77.90 | 78.90 | 71.78 | 76.98 | 72.87 |
| Energy | 60.76 | 81.66 | 78.54 | 77.16 | 75.70 | 76.83 |
| Max Logit | 62.29 | 80.57 | 77.78 | 75.59 | 75.45 | 76.31 |
| GradNorm | 74.72 | 69.77 | 71.01 | 78.73 | 79.89 | 72.13 |
| ODIN | 68.45 | 79.06 | 67.32 | 80.97 | 59.59 | 84.23 |
| SCALE | 61.05 | 81.35 | 76.72 | 77.30 | 70.10 | 78.89 |
| RTL | 69.69 | 75.57 | 82.20 | 70.10 | 76.42 | 73.90 |
| DART | **15.70** | **96.15** | **0.81** | **99.75** | **0.98** | **99.69** |

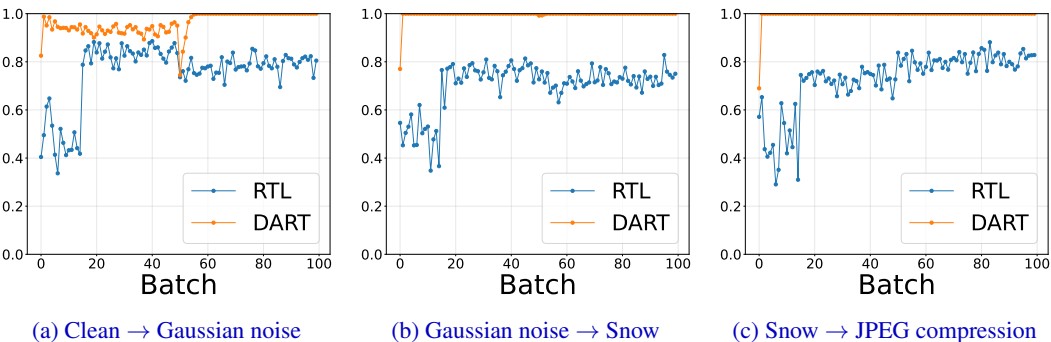

(a) Clean → Gaussian noise     (b) Gaussian noise → Snow     (c) Snow → JPEG compression

Figure 10: AUROC vs. batch

## H SYSTEM OVERHEAD

We measure wall-clock inference time for all methods on the same device, after the backbone forward pass, and only for the OOD-score computation (Figure 11). Concretely, we report the total time required to process 100 mini-batches of size 200 (20k test samples in total), using RegNetY-16GF. Under this protocol, DART falls into the group of fast methods: it is markedly faster than recent baselines such as RTL, NNGuide, and MDS-based variants, which require regression fitting, KNN-style searches, or repeated Mahalanobis evaluations.

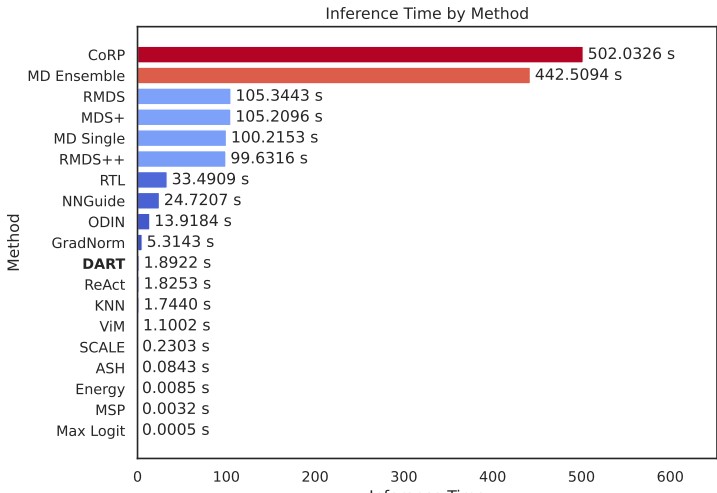

Figure 11: Inference time

## I  IMPACT OF FLIP CORRECTION

To quantitatively assess the impact of flip correction, we compare with a variant that does not perform flip correction, DART-NoFlip, using CIFAR-100-C as the csID dataset. As shown in Table 16, flip correction improves the performance of our method.

Table 16: Performance comparison between *DART*-NoFlip and *DART*

| Method | SVHN-C | | Places365-C | | LSUN-C | | iSUN-C | | Textures-C | | Average | |
|---|---|---|---|---|---|---|---|---|---|---|---|---|
| | FPR95 ↓ | AUROC ↑ | FPR95 ↓ | AUROC ↑ | FPR95 ↓ | AUROC ↑ | FPR95 ↓ | AUROC ↑ | FPR95 ↓ | AUROC ↑ | FPR95 ↓ | AUROC ↑ |
| *DART*-NoFlip | 51.14 | 71.84 | 70.14 | 66.08 | 44.17 | 80.15 | 52.30 | 78.58 | 55.37 | 75.01 | 54.62 | 74.33 |
| *DART* | 48.60 | 79.82 | 68.66 | 68.00 | 44.14 | 80.29 | 50.76 | 79.75 | 51.48 | 80.60 | 52.73 | 77.69 |

To further illustrate the effect on performance over time, we additionally analyze CIFAR-100-C vs Textures-C on a per-corruption basis. Figure 12 visualizes the detection performance over time (per-batch AUROC) before and after applying flip correction. For several corruptions—glass blur, snow, fog, and contrast—we observe that once flip correction is triggered, the dual prototype axis is realigned toward the oracle discriminative direction, leading to an abrupt jump and sustained improvement in performance. For shot noise, flip correction occurs early in the stream at the 20-th batch, after which the subsequent batches exhibit much more stable and higher performance compared to *DART*-NoFlip.

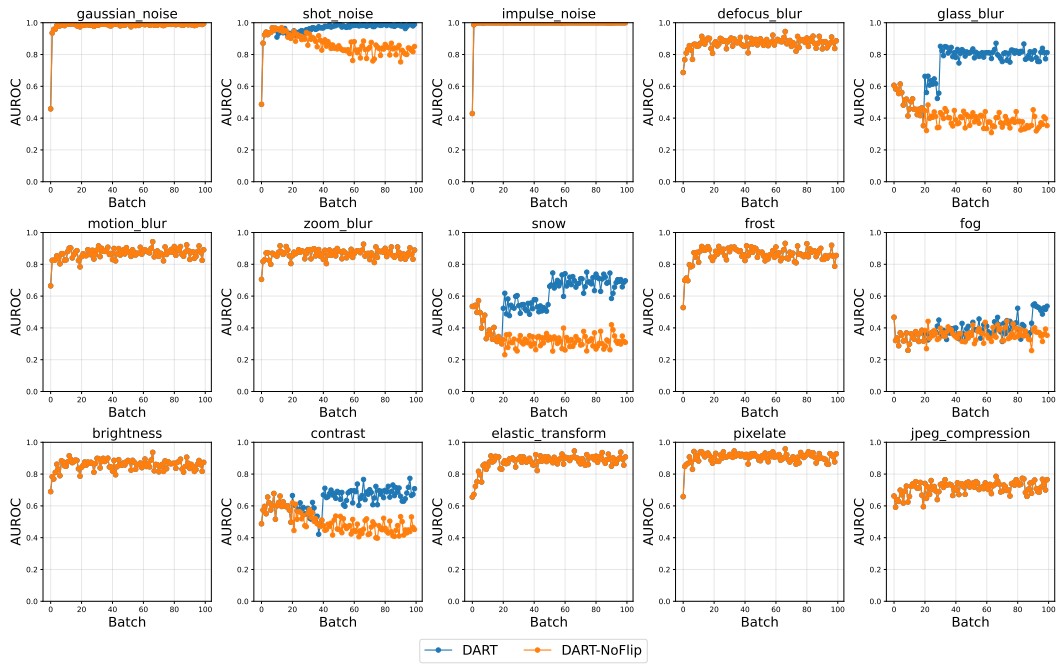

Figure 12: Impact of Flip Correction over time (CIFAR-100-C vs. Textures-C)

## J  IMPACT OF MULTI-LAYER FUSION

We extend Figure 7a to report, for all 15 corruption types, the AUROC of each single-layer variant (Block1, Block2, Block3, FC) and compare them against full *DART* with multi-layer fusion, using CIFAR-100-C as csID and Textures-C as csOOD. See Figure 13 for all results.

This extended analysis reveals that the best-performing layer is highly shift-dependent: under noise-type corruptions such as Gaussian or impulse noise, deeper layers suffer larger degradation, whereas under corruptions such as motion blur and brightness, earlier layers are more severely affected and later layers remain relatively more informative. As a result, relying on any single fixed layer for OOD detection is brittle when the covariate shift type is unknown a priori. In contrast, the fused *DART* score

achieves both the highest mean performance and the most stable behavior: averaged over all corruption types, *DART* not only outperforms every single-layer variant, but also exhibits substantially smaller variation than the strongest single-layer baseline (Block3), with standard deviation 0.1661 versus 0.2582 for Block3. These results quantitatively support our claim that multi-layer fusion is crucial for robust OOD detection under unpredictable covariate shift.

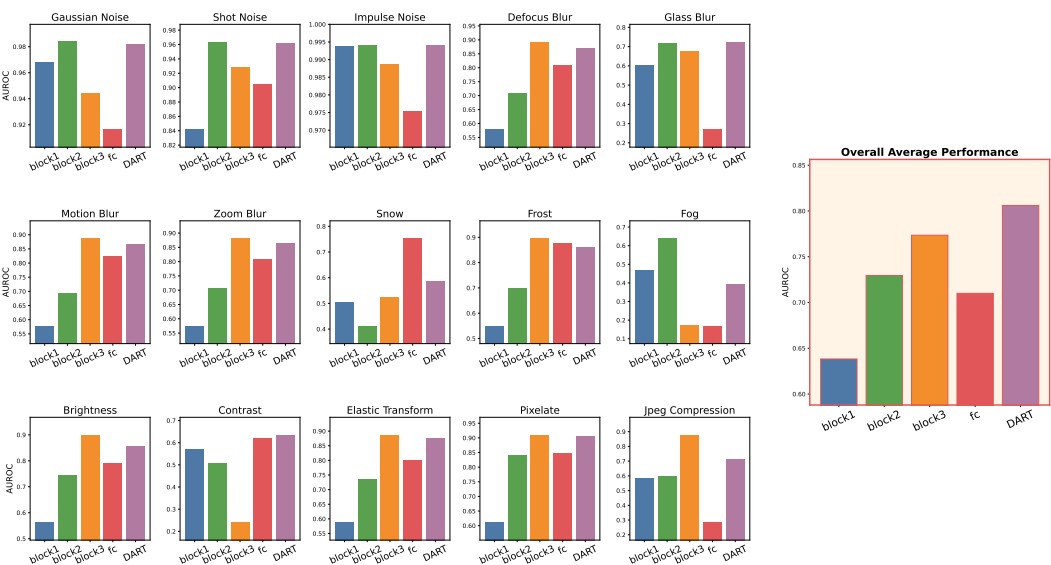

Figure 13: Impact of Multi-layer Fusion (CIFAR-100-C vs. Textures-C)

## K  ANALYSIS ON COLLAPSING SCENARIO

Although very rarely, the discriminative axis tracking of *DART* can sometimes collapse at a particular layer. We further investigated the collapsing cases where perfect alignment was not achieved and identified two distinct failure modes: large angle drift and axis flip. See Figure 14 for for the evolution of the angle in each case.

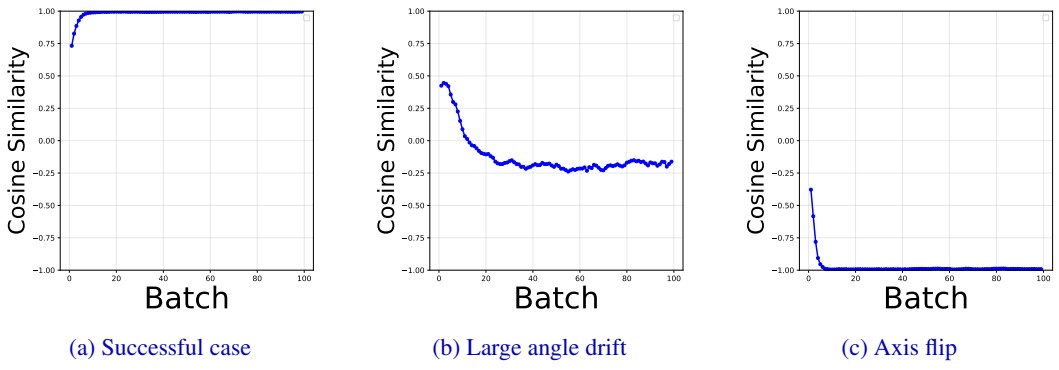

(a) Successful case    (b) Large angle drift    (c) Axis flip

Figure 14: Visualization of Collapsing Scenario

Large angle drifts (Figure 14b) were primarily observed in early layers (e.g., Defocus Blur and Contrast at Layer 0), which we attribute to the limited linear separability of features at this stage; notably, this issue resolves naturally in deeper layers as features become more discriminative.

On the other hand, axis flipping (Figure 14c) was observed in deeper layers (e.g., "Glass Blur" at Layer 5, "Contrast" at Layers 3 and 4). We attribute this primarily to the limitations of the baseline score (MSP), which serves as the reference for prototype initialization and flip detection. Under severe corruptions, the baseline performance degrades significantly, yielding a noisy reference signal

that leads to incorrect initialization or a failure to detect directional flips. This observation implies that the stability of the discriminative axis depends on the quality of the reference score, and employing a more robust reference signal could potentially resolve these flipping issues. Crucially, despite these isolated local instabilities, we emphasize that our Multi-layer Fusion strategy effectively mitigates these risks. By aggregating decisions across multiple layers, *DART* compensates for occasional drifts or flips occurring in individual layers, ensuring robust overall performance. Consequently, even in scenarios where specific layers struggle to align, the ensemble model maintains an AUROC greater than 0.9 across all covariate shifts, validating the practical effectiveness of our approach.

## L    VISUALIZATION OF ROC CURVES

We visualize the ROC curves of our method and the baselines across several evaluation settings. As shown in Figure 15, *DART* achieves lower FPR in this high-TPR region even when overall AUROC is comparable to baselines. Since real-world deployment requires maintaining high ID acceptance while minimizing false alarms, FPR@95TPR better captures the performance that matters in practice.

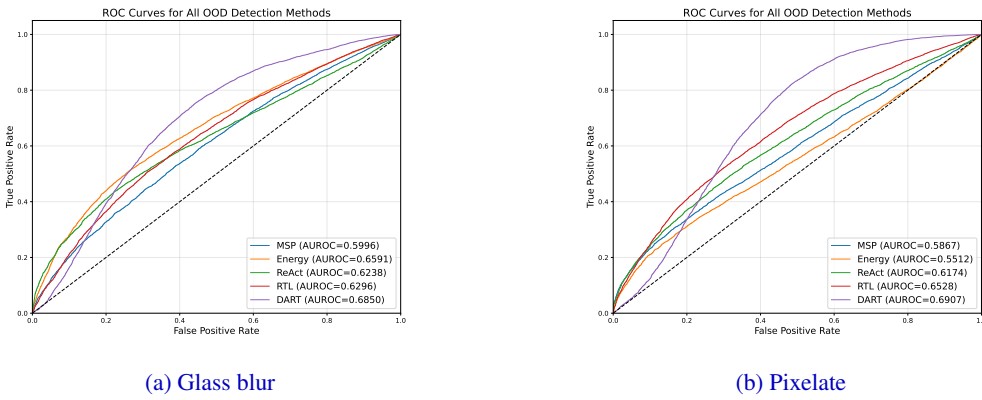

(a) Glass blur                                      (b) Pixelate

Figure 15: ROC curves for CIFAR-100 vs. LSUN under different corruptions

