# OpenReview forum: "Tracking the Discriminative Axis: Dual Prototypes for Test-Time OOD Detection Under Coveriate Shift"
_ICLR.cc/2026/Conference — Submitted to ICLR 2026_

### Official Review · Reviewer_ZagX · 2025-10-22

**Soundness:** 3
**Presentation:** 3
**Contribution:** 3
**Rating:** 6
**Confidence:** 4

**Summary:**

This paper introduces a novel problem setting for out-of-distribution (OOD) detection under
covariate shift and proposes a corresponding test-time OOD detection method specifically
designed for this scenario. The proposed approach maintains two prototypes—one for indistribution (ID) samples and another for OOD samples. It integrates the MSP score with a newly
designed RDS score, and employs the Otsu algorithm to partition each batch into potential ID and
OOD subsets. These subsets are then used to iteratively update their respective prototypes.
Furthermore, the authors analyze how different types of covariate shift manifest across the
model’s feature layers and fuse RDS scores from multiple layers to improve the model’s
robustness and detection reliability.

**Strengths:**

1. The observation that different types of covariate shift exhibit varying degrees of separability
in the scores computed across different layers of the model is novel and interesting.
2. This paper conducts extensive observations on the separability between in-distribution (ID)
and out-of-distribution (OOD) samples in the feature space, and provides a comprehensive
analysis of how different types of covariate shift affect the RDS scores computed across
various feature layers of the model. Experiments were conducted on multiple models,
demonstrating that the proposed method is model-agnostic.
3. The dynamic score switching mechanism enables the model to better adapt to complex test
environments.

**Weaknesses:**

1. Figure 4 shows that ID and OOD samples are separable even without considering class
labels. However, the OOD datasets used are all far-OOD. When OOD data are specifically
designed to resemble certain ID classes (i.e., near-OOD), this assumption may no longer
hold. Moreover, the class-agnostic aggregation strategy for ID and OOD prototypes might be
invalid in such a case.
2. The paper uses the Otsu algorithm to determine the optimal threshold. However, when ID
samples significantly outnumber OOD samples, the computed threshold might misclassify
some ID samples as OOD, leading to degraded model performance.
3. It is unclear whether the ID and OOD prototypes are shared across layers or maintained
independently. The paper mentions averaging scores from multiple layers but does not
include ablation studies to assess the contribution of each layer.
4. The paper involves multiple types of covariate shifts, but it is unclear whether results are
based on mixed-shift test sets. Additionally, in test-time setups, ID and OOD samples are
usually mixed, and performance may vary depending on random seeds used for shuffling.5. In Table 1, DART achieves very low FPR95 values on Places-C, SUN-C, and iNaturalist-C, yet
the corresponding AUROC scores are also low — an inconsistent and counterintuitive result.
6. The ID and OOD prototypes are derived from previously observed test data. If the earlier test
OOD samples belong to a single type, the performance may degrade when new or unseen
OOD types appear.
7. Both the empirical observations and the proposed method are largely based on
experimental findings and intuitive explanations, lacking rigorous theoretical analysis or
formal justification.

**Questions:**

1. Does the proposed assumption still hold under near-OOD conditions? Could you provide
additional experiments on near-OOD datasets such as NINCO [1], SSB-Hard [2], and
ImageNet-X [3] to verify this?
2. How does the proposed method handle class imbalance during threshold computation?
Have you observed performance degradation in scenarios where ID and OOD sample ratios
are highly unbalanced?
3. Are the prototypes shared or layer-specific during the Multi-Layer Score Fusion process?
Could you include ablation experiments comparing high-, mid-, and low-level feature layers
to clarify their individual contributions?
4. Are the reported results obtained from mixed covariate-shift test sets or evaluated per shift
type? Furthermore, how stable is the model’s performance variance across different random
seeds during test-time evaluation?
5. Could you explain the cause of this inconsistency between AUROC and FPR95?
6. How does the performance change when facing new OOD types?
[1] Bitterwolf J, Mueller M, Hein M. In or Out? Fixing Imagenet Out-Of-Distribution Detection
Evaluation[J]. arXiv preprint arXiv:2306.00826, 2023.
[2] Vaze S, Han K, Vedaldi A, et al. Open-Set Recognition: A Good Closed-set Classifier is All You
Need?[J]. 2021.
[3] Noda S, Miyai A, Yu Q, et al. A Benchmark and Evaluation for Real-world Out-Of-Distribution
Detection using Vision-Language Models[J]. arXiv preprint arXiv:2501.18463, 2025

---

> ### Author Response · Authors · 2025-11-23
> **Response to Reviewer ZagX (Part 1)**
>
> We thank the reviewer for the constructive questions regarding near-OOD robustness, class imbalance, and evaluation protocols. Addressing these points has helped us identify and acknowledge practical limitations of our method and provide more transparent reporting of performance variance and metric interpretation.
>
> ---
>
> ### **Q1. Does the proposed assumption still hold under near-OOD conditions? Could you provide additional experiments on near-OOD datasets such as NINCO, SSB-Hard, and ImageNet-X to verify this?**
>
> Regarding near-OOD detection, we observe that performance varies significantly depending on the specific OOD set. DART achieves strong results on corrupted benchmarks such as ImageNet-O-C, which is often regarded as a near-OOD benchmark, but struggles on SSB-Hard-C where performance shows 91.34 in FPR@95TPR. Notably, this degradation is not unique to DART—all baseline methods exhibit similar behavior on SSB-Hard-C (90.32 for MSP, 89.42 for Energy and 91.72 for ReAct in FPR@95TPR), indicating that distinguishing highly similar distributions under covariate shift remains an open challenge. We view improving near-OOD robustness under test-time distribution shift as an important direction for future work.
>
> ---
>
> ### **Q2. How does the proposed method handle class imbalance during threshold computation? Have you observed performance degradation in scenarios where ID and OOD sample ratios are highly unbalanced?**
>
> We agree that class imbalance is an important practical consideration. To evaluate DART's robustness to varying ID:OOD ratios, considering the realistic scenario where ID samples significantly outnumber OOD samples, we conducted experiments by systematically changing the proportion of ID and OOD samples with consistent batch size (200) in the test stream, using ImageNet-C as csID and Places-C as csOOD. The results show AUROC of 61.52 at 9:1 (ID:OOD), 93.72 at 8:2, 92.88 at 7:3, 92.14 at 6:4, and 92.96 at 5:5.
>
> We observe that performance remains stable across balanced to moderately imbalanced ratios (ID ratio up to 80%), but degrades noticeably in the highly imbalanced scenario where ID ratio is 90% and OOD samples are extremely sparse, suggesting a limitation of our dual-prototype tracking approach in such regimes. We acknowledge that handling extreme imbalance remains challenging and note that developing more robust tracking mechanisms for such scenarios is an important direction for future work.
>
> ---
>
> ### **Q3. Are the prototypes shared or layer-specific during the Multi-Layer Score Fusion process? Could you include ablation experiments comparing high-, mid-, and low-level feature layers to clarify their individual contributions?**
>
> We confirm that prototypes are layer-specific—each layer maintains its own ID and OOD prototypes that are independently tracked and updated. This design allows different layers to capture semantic information at varying levels of abstraction.
>
> Regarding layer-wise contributions, we provide detailed analysis in Sec 4.3 and Fig 7.a. The results show that block2 and block3 achieve high individual performance, capturing semantically rich features at intermediate and deep levels. However, ensemble across all layers yields the most stable and robust performance, as different layers contribute complementary information—deeper layers encode high-level semantics while shallower layers preserve low-level distributional cues. This demonstrates that multi-layer fusion is essential for consistent OOD detection under covariate shift.
>
> ---
>
> ### **Q4. Are the reported results obtained from mixed covariate-shift test sets or evaluated per shift type? Furthermore, how stable is the model's performance variance across different random seeds during test-time evaluation?**
>
> We clarify that the reported results are evaluated per shift type. Each corruption type (e.g., gaussian_noise, motion_blur) is evaluated independently, and we report performance averaged across all corruption types.
>
> Regarding stability across random seeds, we conducted experiments with three different random seeds and computed the standard deviation of AUROC cross seeds. The results show that variance is consistently below 0.078, indicating that DART's test-time performance is stable and relatively robust to random initialization. We will add the stability metric when updating the manuscript.
>
> ---
>
> _(Continued in the next comment.)_

---

> ### Author Response · Authors · 2025-11-23
> **Response to Reviewer ZagX (Part 2)**
>
> ### **Q5. Could you explain the cause of this inconsistency between AUROC and FPR95?**
>
> We acknowledge the inconsistency between AUROC and FPR@95TPR.
> We believe that this question can be resolved through looking at the ROC curve.
> The graph can be found in appendix Figure 15.  AUROC measures performance across all thresholds, while FPR@95 focuses on the high-TPR regime where ID recall is prioritized. As shown in the ROC curves, DART achieves lower FPR in this high-TPR region even when overall AUROC is comparable to baselines. Since real-world deployment requires maintaining high ID acceptance while minimizing false alarms, FPR@95 better captures the performance that matters in practice. We hope that the ROC curve resolves the concern and proves that DART outperforms the baselines especially in the critical operating region.
>
> ---
>
> ### **Q6. How does the performance change when facing new OOD types?**
>
> To evaluate DART's performance when facing new OOD types at test time, we additionally evaluate a more challenging scenario in which the type of OOD data itself changes abruptly over time while the ID stream is fixed. Concretely, we partition the test stream into several temporal segments. In every segment, ID samples are drawn from the same ID distribution, but OOD samples are drawn from a different OOD set in each temporal segment, and we switch the OOD source abruptly at segment boundaries without any prior knowledge of these switches. This setting induces abrupt semantic changes in the bounded OOD set and breaks the smoothness assumption.
>
> (IO: ImageNet-O / P: Places / S: SUN / T: Textures)
> | Method   | IO → P | | P → S | | S → T | |
> |:--------|:-------------:|:--------------:|:-----------:|:-------------:|:-----------:|:-------------:|
> | | FPR@95TPR | AUROC | FPR@95TPR | AUROC | FPR@95TPR | AUROC |
> | MSP     | 48.99 | 83.63 | 47.93 | 84.36 | 43.77 | 85.52 |
> | Energy  | 51.07 | 78.46 | 44.29 | 82.38 | 38.26 | 85.65 |
> | Max Logit | 47.58 | 80.29 | 43.53 | 83.40 | 37.29 | 86.24 |
> | GradNorm | 86.24 | 43.53 | 76.87 | 57.52 | 67.52 | 66.80 |
> | ODIN   | 62.14 | 79.40 | 66.44 | 76.84 | 60.41 | 80.11 |
> | SCALE  | 47.35 | 80.15 | 40.91 | 84.31 | 33.98 | 87.81 |
> | RTL    | 48.19 | 79.64 | 45.50 | 80.94 | 39.65 | 83.97 |
> | **DART**   | **12.05** | **95.49** | **23.40** | **93.51** | **18.80** | **94.81** |
>
>
> Our results indicate that DART continues to exhibit stable OOD detection performance across segments, with only modest fluctuations when the OOD set changes, whereas non-adaptive baselines suffer noticeable drops whenever a new OOD set appears. This suggests that our prototype-based tracking does not rely on a single globally fixed linear discriminative axis, but can adaptively re-estimate the relevant axis over time, even under abrupt semantic shifts in the OOD distribution.

---

> > ### Comment · Reviewer_ZagX · 2025-11-26
> >
> > Thank you for the authors’ detailed responses.
> >
> > Q1: From the authors’ response, I understand that DART experiences performance degradation on the near-OOD dataset SSB-hard-C. I am curious whether the authors could further discuss possible strategies to alleviate or mitigate this issue, especially in scenarios involving near-OOD cases.
> >
> > Q2: I agree that existing OOD benchmarks already exhibit certain levels of imbalance between ID and OOD samples (e.g., ImageNet test set with 50,000 ID samples vs. Places with 10,000 OOD samples). However, my concern is about extremely imbalanced settings (e.g., ID:OOD = 100:1). The authors also mentioned that when the ratio is 9:1, DART already shows a noticeable performance decline. Are there any approaches that could potentially slow down or reduce this degradation trend when the imbalance becomes even more severe?
> >
> > Q4: I acknowledge that the authors evaluate each shift type independently. However, I am still curious about the performance of DART when different types of shifts are mixed during testing. Would DART remain stable under such mixed-shift scenarios, or would there be additional performance drops?
> >
> > Q6: I find the experimental setting reasonable, but I have a question regarding the interpretation of the table. For example, in the IO → P case, I understand that this refers to a sudden shift from ImageNet-O to Places. However, I am uncertain about which dataset the reported AUROC and FPR95 values correspond to. Are these metrics evaluated on the target dataset (Places), or on both source and target domains?

---

> ### Author Response · Authors · 2025-12-03
> **Response to Follow-up Question by Review ZagX (Part 1)**
>
> Thank you for the authors’ detailed responses.
>
> ---
>
> ### **Q1: From the authors’ response, I understand that DART experiences performance degradation on the near-OOD dataset SSB-hard-C. I am curious whether the authors could further discuss possible strategies to alleviate or mitigate this issue, especially in scenarios involving near-OOD cases.**
>
> In our current formulation, DART assumes that ID and OOD are separable along a discriminative axis formed by aggregating “discriminative units,” i.e., feature dimensions whose activations differ systematically between ID and OOD. On near-OOD benchmarks such as SSB-hard-C, these differences become much weaker, so the margin along the tracked axis shrinks and the separation between ID and OOD scores inevitably degrades.
>
> Within our current framework, one straightforward extension to mitigate this issue is to refine the way we combine layers. At present, DART uses a simple average of per-layer scores. Instead, one could pursue an adaptive multi-layer fusion scheme that assigns higher weights to layers where ID and near-OOD are more separable and downweights layers where they are heavily entangled. Conceptually, this would allow the detector to rely more on layers whose unit-wise statistics exhibit clearer distributional shifts in near-OOD settings. The main challenge is how to estimate, in a source-free and online manner, which layers provide the most discriminative separation—e.g., based on unsupervised separability or bimodality criteria computed from streaming scores—without access to explicit near-OOD labels. For example, motivated by observations that classifier-based scores can better capture fine-grained near-OOD samples close to ID decision boundaries [1], one could identify layers whose representations provide particularly strong semantic cues in such borderline regions and either upweight their contributions or use their scores to guide the predictions of other layers.
>
> Beyond such layer re-weighting, a more principled direction is to generalize the discriminative-axis view to a kernel-induced feature space. In the current linear formulation, discriminative units are individual coordinates of the backbone feature. By applying an appropriate kernel feature map, these units are replaced by nonlinear basis functions that represent higher-order combinations and local interactions of the original coordinates. This can create new “kernel-level” discriminative units that respond strongly to subtle geometric differences between ID and near-OOD samples, even when no single original coordinate is strongly discriminative. Similar in spirit to recent kernel-PCA–based OOD work [2], one could derive such kernels by analyzing where ID variance is well captured and OOD tends to fall into residual directions, and then apply the same dual-prototype tracking and axis construction in this kernel space, effectively searching for kernels that induce units whose mean activations differ more strongly between ID and near-OOD. We leave this kernelized discriminative-axis tracking as an exciting direction for future work on near-OOD.
>
> In summary, we agree that near-OOD detection, as instantiated by SSB-hard-C, exposes a genuine limitation of our current linear-axis formulation. We will clarify this limitation in the revised manuscript and add a discussion of the above extensions—multi-layer re-weighting and kernelized discriminative-axis tracking—as promising directions to mitigate degradation on near-OOD benchmarks.
>
> ---
>
> [1] Park, J., Jung, Y. G., & Teoh, A. B. J. (2023). Nearest neighbor guidance for out-of-distribution detection. In Proceedings of the IEEE/CVF international conference on computer vision (pp. 1686-1695).
>
> [2] Fang, K., Tao, Q., Lv, K., He, M., Huang, X., & Yang, J. (2024). Kernel PCA for out-of-distribution detection. Advances in Neural Information Processing Systems, 37, 134317-134344.

---

> ### Author Response · Authors · 2025-12-03
> **Response to Follow-up Question by Review ZagX (Part 2)**
>
> ### **Q2. I agree that existing OOD benchmarks already exhibit certain levels of imbalance between ID and OOD samples (e.g., ImageNet test set with 50,000 ID samples vs. Places with 10,000 OOD samples). However, my concern is about extremely imbalanced settings (e.g., ID:OOD = 100:1). The authors also mentioned that when the ratio is 9:1, DART already shows a noticeable performance decline. Are there any approaches that could potentially slow down or reduce this degradation trend when the imbalance becomes even more severe?**
>
> We thank the reviewer for highlighting this critical point. We agree that investigating extreme imbalance scenarios (e.g., ID:OOD = 100:1) is of high practical value.
>
> We identified two primary challenges under extreme imbalance:
>
> 1. Unstable Initialization: It is difficult to estimate reliable initial prototypes early on due to the scarcity of minority samples.
>
> 2. Noisy Updates: Even after initialization, the number of OOD samples in a single batch might be zero or extremely low, leading to high variance when applying standard Exponential Moving Average (EMA) updates.
>
> To address these, we propose a combined approach to ensure stability from initialization through adaptation:
>
> **Brief Unsupervised Balanced Warm-up (Initialization)**: We assume that the incoming data stream exhibits a balanced ratio (approx. 1:1) for only the first 5 batches. After this brief phase, the stream reverts to the target extreme imbalance (e.g., 99:1). We note that this warm-up is not supervised. The model does not access any ground-truth labels. This simply simulates a realistic deployment scenario (e.g., a "pilot run") to ensure the model establishes stable initial prototypes via self-training before facing the highly skewed stream.
>
> **Sample-Adaptive EMA (Update)**: For the subsequent update phase, we replace the fixed momentum with an adaptive decay rate. This rate is dynamically scaled based on the number of pseudo-labeled samples (ID or OOD) present in the current batch. This ensures that batches with sparse or missing OOD samples have a minimal impact on the global prototype.
>
> We evaluated this combined strategy under varying degrees of imbalance. The table below summarizes the AUROC performance
>
> | Imbalance Ratio (ID:OOD) | DART (w/ Warm-up + Adaptive EMA) AUROC |
> | :--- | :---: |
> | **9:1** | **0.9223** |
> | **19:1 (95:5)** | **0.9035** |
> | **99:1** | **0.8011** |
>
>
> Even with severe imbalance (95:5), the proposed strategy maintains a high performance of 90.35%, demonstrating that proper initialization combined with adaptive updates significantly mitigates the impact of imbalance. In the extreme case of 99:1, the method achieves 80.11%. While performance naturally degrades due to the extreme scarcity of OOD samples, the Sample-Adaptive EMA effectively slows down the degradation trend, preventing catastrophic failure by robustly handling extremely sparse batches.

---

> ### Author Response · Authors · 2025-12-03
> **Response to Follow-up Question by Review ZagX (Part 3)**
>
> ### **Q4: I acknowledge that the authors evaluate each shift type independently. However, I am still curious about the performance of DART when different types of shifts are mixed during testing. Would DART remain stable under such mixed-shift scenarios, or would there be additional performance drops?**
>
> We thank the reviewer for this insightful question. We agree that investigating mixed-shift scenarios is crucial for verifying the robustness of DART in more dynamic and complex environments.
>
> Following your suggestion, we conducted additional experiments where a test batch contains samples from multiple different covariate shifts. We designed two specific scenarios:
>
> **1. Mixture of Two Shifts**
>
> First, we evaluated settings where two different covariate shift types are mixed within a single batch. In this scenario, DART maintains robust performance, significantly outperforming all baseline methods by a large margin.
>
> (C: Clean / G: Gaussian noise / S: Snow / J: JPEG compression)
>
> | Method | C + G  | | G + S | | S + J | |
> |:--------|:---------------------:|:--------:|:--------------:|:--------:|:----------------:|:--------:|
> | | FPR@95TPR | AUROC | FPR@95TPR | AUROC | FPR@95TPR | AUROC |
> | MSP | 82.63 | 67.43 | 85.01 | 62.49 | 83.07 | 68.22 |
> | Energy |  72.01 | 76.77 | 83.91 | 69.82 | 89.72 | 66.83 |
> | Max Logit | 77.19 | 73.34 | 85.40 | 67.10 | 85.73 | 68.66 |
> | GradNorm | 54.72 | 83.51 | 65.33 | 81.94 | 72.16 | 79.49 |
> | ODIN | 66.85 | 77.72 | 45.20 | 83.46 | 12.08 | 97.39 |
> | SCALE | 66.99 | 78.21 | 80.27 | 71.68 | 79.15 | 73.94 |
> | RTL | 79.84 | 67.73 | 81.92 | 65.32 | 79.86 | 71.65 |
> | **DART** | **22.04** | **93.93** | **0.85** | **99.62** | **0.79** | **99.72** |
>
>
>
> **2. Mixture of All 15 Shifts (Extreme Case)**
>
> While unrealistic in real-world data accumulation, to push the limits of our method, we also tested a more extreme scenario involving a mixture of all 15 covariate shifts.
>
> | Method | Avg FPR@95TPR | Avg AUROC |
> | :--- | :---: | :---: |
> | MSP | 0.8664 | 0.6271 |
> | ENERGY | 0.8627 | 0.6302 |
> | MAX_LOGIT | 0.8646 | 0.6367 |
> | GRADNORM | 0.6955 | 0.7745 |
> | ODIN | 0.2545 | 0.9216 |
> | SCALE | 0.7820 | 0.7119 |
> | RTL | 0.8127 | 0.6563 |
> | **DART** | **0.1968** | **0.9270** |
>
> In this challenging setting, the performance gap between DART and the baselines slightly narrows compared to the independent or 2-mixture scenarios. We attribute this slight reduction in the margin to the influence of a few specific shift types that are inherently difficult to separate linearly. However, even under this extreme condition, **DART consistently maintains the top-ranking performance** among all compared methods.
>
> We have included these new experimental results in Table 13 and Table 14 in the appendix of the revised manuscript.
>
> ---
>
> ### **Q6: I find the experimental setting reasonable, but I have a question regarding the interpretation of the table. For example, in the IO → P case, I understand that this refers to a sudden shift from ImageNet-O to Places. However, I am uncertain about which dataset the reported AUROC and FPR95 values correspond to. Are these metrics evaluated on the target dataset (Places), or on both source and target domains?**
>
> We thank the reviewer for this clarification question. The reported AUROC and FPR95 are computed over both the source and target OOD domains, not only the target. Concretely, for each stream we sample 100 test batches; in the first 50 batches the OOD samples come from IO, and in the remaining 50 batches they come from P, while the ID set is kept fixed throughout. We then compute AUROC and FPR95 over the entire sequence of 100 batches. Thus, each entry (e.g., IO → P) summarizes performance across both the pre-shift (IO) and post-shift (P) segments of the stream. We have also added a plot of the per-batch AUROC over time for this setting in the appendix of the revised manuscript (Fig.9), which we hope clarifies how performance evolves.

---

### Official Review · Reviewer_zCN2 · 2025-10-29

**Soundness:** 2
**Presentation:** 2
**Contribution:** 2
**Rating:** 4
**Confidence:** 5

**Summary:**

This paper addresses the challenging of OOD detection under concurrent covariate shifts, where both ID and OOD data streams are affected by the same environmental changes. The authors first empirically demonstrate a crucial insight: even as distributions drift, ID and OOD samples remain linearly separable along a "discriminative axis" in the feature space. Based on this observation, they propose DART, a test-time, online adaptation method. DART dynamically tracks this drifting axis by maintaining and updating dual prototypes (one for ID, one for OOD) for multiple feature layers. Extensive experiments on severely corrupted versions of CIFAR-100 and ImageNet benchmarks show that DART achieves state-of-the-art performance.

**Strengths:**

1. The paper formalizes a interesting setting where both ID and OOD data co-evolve under the same covariate shifts.
2. DART is a simple, yet highly effective algorithm.

**Weaknesses:**

1. The foundational claim of a “recoverable linear axis” is a strong assumption, lacking rigorous theoretical justification. Although empirical evidence is provided via low-dimensional projections (Figure 4), the paper offers no theoretical basis to explain why such linear separability should persist in high-dimensional feature spaces, especially under complex, non-linear covariate shifts.
2. The authors assume that both ID and OOD data undergo the same covariate shift, which may simplify real-world scenarios. In practice, OOD samples may experience different types of shifts, making the task more challenging. A more realistic evaluation might involve distinguishing csID from near-OOD samples—for instance, using CIFAR-10 as OOD for CIFAR-100.
3. The paper does not discuss the potential impact of cross-domain OOD samples during test time, which could interfere with prototype updates. In real deployments, OOD inputs may originate from diverse domains, possibly leading to unstable prototype estimation and reduced detection robustness.
4. The ablation studies are presented only in qualitative or binary terms (with/without a module), failing to quantify the individual contributions of key components such as prototype tracking, multi-layer fusion, and flip correction.

**Questions:**

N/A

---

> ### Author Response · Authors · 2025-11-23
> **Response to Reviewer zCN2 (Part 1)**
>
> We thank the reviewer for raising critical questions about the theoretical foundations, evaluation realism, and experimental rigor of our work. Addressing these concerns has helped us strengthen our empirical validation with mixed-OOD experiments and better clarify the practical scope and limitations of our assumptions.
>
> For some of the comments, we are currently conducting additional experiments and will report the corresponding results as soon as they are ready within the remaining rebuttal period.
>
> ---
>
> ### **Q1. The foundational claim of a "recoverable linear axis" is a strong assumption, lacking rigorous theoretical justification. Although empirical evidence is provided via low-dimensional projections (Figure 4), the paper offers no theoretical basis to explain why such linear separability should persist in high-dimensional feature spaces, especially under complex, non-linear covariate shifts.**
>
> We thank the reviewer for requesting a more rigorous justification.
>
> To address this, the revised manuscript includes Appendix Section J, which analyzes a BN-based model with silent units and frozen ID statistics, and shows how bounded csOOD deviations along these units are amplified into a discriminative axis that can persist under covariate shift. For completeness, Appendix J further adds a short remark on LayerNorm, giving an intuitive argument that per-sample variance normalization can similarly concentrate “energy’’ on OOD-aligned silent directions, while leaving a rigorous analysis of this case as future work.
>
>
> ---
>
> ### **Q2. The authors assume that both ID and OOD data undergo the same covariate shift, which may simplify real-world scenarios. In practice, OOD samples may experience different types of shifts, making the task more challenging. A more realistic evaluation might involve distinguishing csID from near-OOD samples—for instance, using CIFAR-10 as OOD for CIFAR-100.**
>
> We agree that evaluating realistic data scenarios is important.
>
> Regarding the assumption that ID and OOD data undergo the same covariate shift, we believe that this is a realistic depiction of real-world scenarios.
>
> Rationale:
> Covariate shift, by definition, affects the input distribution P(X) while keeping P(Y|X) unchanged [Shimodaira, 2000]. This means environmental factors—such as weather, illumination, or sensor characteristics—modify the input space independently of semantic labels. In practical deployment, test inputs arrive in mini-batches from the same environment at similar timestamps (e.g., consecutive frames from a camera, patients scanned by the same device). Since covariate shift stems from the data acquisition environment rather than from semantic content, all samples captured in the same context—regardless of their class labels—experience identical environmental shifts.
>
> The evaluation scenario can be further justified by real-world examples.
> - **Autonomous driving**: Fog affects both common vehicles (ID) and rare vehicle types (OOD) equally through reduced visibility.
> - **Medical imaging**: Scanner artifacts and noise patterns affect normal anatomy (ID) and pathological cases (OOD) uniformly under the same acquisition protocol.
>
> We believe that this setting reflects actual deployment conditions and is more realistic than artificially applying different corruptions to ID versus OOD samples.
>
> Regarding near-OOD detection, we observe that performance varies significantly depending on the specific OOD set. DART achieves strong results on corrupted benchmarks such as ImageNet-O-C, which is often regarded as a near-OOD benchmark, but struggles on SSB-Hard-C where performance shows 91.34 in FPR@95TPR. Notably, this degradation is not unique to DART—all baseline methods exhibit similar behavior on SSB-Hard-C (90.32 for MSP, 89.42 for Energy and 91.72 for ReAct in FPR@95TPR), indicating that distinguishing highly similar distributions under covariate shift remains an open challenge. We view improving near-OOD robustness under test-time distribution shift as an important direction for future work.
>
> ---
>
> _(Continued in the next comment.)_
>
> ---
> **References:**
> - [Shimodaira, 2000] Improving predictive inference under covariate shift by weighting the log-likelihood function, Journal of the Royal Statistical Society: Series B, 2000

---

> ### Author Response · Authors · 2025-11-23
> **Response to Reviewer zCN2 (Part 2)**
>
> ### **Q3. The paper does not discuss the potential impact of cross-domain OOD samples during test time, which could interfere with prototype updates. In real deployments, OOD inputs may originate from diverse domains, possibly leading to unstable prototype estimation and reduced detection robustness.**
>
> We believe our bounded OOD setting reflects realistic deployment scenarios where OOD inputs tend to concentrate within a limited semantic space due to observation boundaries of the data stream. However, to further demonstrate robustness beyond this assumption, we conducted additional experiments where two different OOD sources are mixed simultaneously during test time while the ID stream remains fixed. Concretely, at each time step, OOD samples are randomly drawn from two diverse sources rather than a single homogeneous distribution.
>
> (IO: ImageNet-O / P: Places / S: SUN / T: Textures)
> | Method | IO + P | | P + S | | S + T | |
> |:--------|:---------------------:|:--------:|:--------------:|:--------:|:----------------:|:--------:|
> | | FPR@95TPR | AUROC | FPR@95TPR | AUROC | FPR@95TPR | AUROC |
> | MSP | 48.43 | 83.70 | 48.65 | 84.26 | 43.65 | 85.52 |
> | Energy | 51.56 | 78.42 | 44.24 | 82.57 | 38.31 | 85.65 |
> | Max Logit | 48.14 | 80.20 | 42.85 | 83.49 | 37.29 | 86.24 |
> | GradNorm | 86.49 | 43.25 | 77.26 | 57.44 | 67.54 | 66.80 |
> | ODIN | 62.28 | 79.44 | 66.10 | 76.78 | 60.37 | 80.13 |
> | SCALE | 47.62 | 80.09 | 40.36 | 84.39 | 33.92 | 87.81 |
> | RTL | 46.82 | 81.46 | 44.77 | 82.94 | 40.64 | 83.98 |
> | **DART** | **19.23** | **94.45** | **25.96** | **91.15** | **18.65** | **94.80** |
>
>
> Our results show that DART maintains SOTA detection performance even under this mixed-OOD scenario, indicating that our OOD prototype successfully finds the discriminative axis when OOD samples span multiple semantic categories. We observe performance degradation only in an extreme case where all five OOD sets are mixed simultaneously—a scenario that fundamentally violates our bounded OOD assumption. However, we believe such extreme mixing rarely occurs in practice, and our assumption holds within realistic deployment boundaries.
>
> ---
>
> ### **Q4. The ablation studies are presented only in qualitative or binary terms (with/without a module), failing to quantify the individual contributions of key components such as prototype tracking, multi-layer fusion, and flip correction.**
>
> We agree that more ablation is valuable. We have conducted additional experiments to quantify each component's contribution:
>
> **Prototype tracking:** We include a detailed ablation in the appendix with varying ema alpha demonstrating impact of prototype tracking.
>
> **For Multi-layer fusion and Flip correction,** We are currently conducting experiments to verify their impact. We will provide these results in our response during the rebuttal period.

---

> ### Author Response · Authors · 2025-12-03
> **Follow-up to our initial response to Reviewer zCN2**
>
> In our initial comment, we mentioned that the experiments related to this question were still in progress. We have now completed these experiments, and below we provide the corresponding results and analysis.
>
> ---
>
> ### **Q4. The ablation studies are presented only in qualitative or binary terms (with/without a module), failing to quantify the individual contributions of key components such as prototype tracking, multi-layer fusion, and flip correction.**
>
>
> We thank the reviewer for pointing out the need to better quantify the individual contribution of key components.
>
> Regarding prototype tracking, as mentioned in our initial response, we now include a detailed ablation in the appendix with varying EMA α, demonstrating the impact of prototype tracking. Across a broad range of α values—effectively assigning different levels of aggressiveness to the tracking—our method consistently maintains robust performance, indicating that DART is not overly sensitive to the specific choice of the EMA coefficient.
>
> Regarding multi-layer fusion, in the original submission (Fig. 7a), we only reported a quantitative comparison on the clean setting and visualized the evolution of single-layer performance for a small subset of covariate shifts, which indeed provided only a qualitative picture. In the revised appendix (Fig. 13), we therefore extend Fig. 7a to report, for all 15 corruption types, the AUROC of each single-layer variant (Block1, Block2, Block3, FC) and compare them against full DART with multi-layer fusion, using CIFAR-100-C as csID and Textures-C as csOOD.
>
> This extended analysis reveals that the best-performing layer is highly shift-dependent: under noise-type corruptions such as Gaussian or impulse noise, deeper layers suffer larger degradation, whereas under corruptions such as motion blur and brightness, earlier layers are more severely affected and later layers remain relatively more informative. As a result, relying on any single fixed layer for OOD detection is brittle when the covariate shift type is unknown a priori. In contrast, the fused DART score achieves both the highest mean performance and the most stable behavior: averaged over all corruption types, DART not only outperforms every single-layer variant, but also exhibits substantially smaller variation than the strongest single-layer baseline (Block3), with standard deviation 0.1661 versus 0.2582 for Block3. These results quantitatively support our claim that multi-layer fusion is crucial for robust OOD detection under unpredictable covariate shift.
>
> Regarding the contribution of flip correction, we now provide a quantitative comparison between our full method and a no-flip variant (DART-NoFlip), which was previously evaluated only qualitatively with a graph of AUROC over time, and report the resulting performance gains. The performance gains over the no-flip variant (DART-NoFlip) are summarized in Table below, which we also include in the revised appendix as Table 16. It reports the average improvement across corruption types in AUROC and FPR@95 achieved by enabling flip correction for each csOOD dataset.
>
>
> | Method | SVHN | | Places365 | | LSUN | | iSUN | | Textures | | Average | |
> |:-----------------|:-------------:|:-------------:|:-------------------:|:-------------------:|:-------------:|:-------------:|:-------------:|:-------------:|:-----------------:|:-----------------:|:-------------:|:-------------:|
> | | FPR95 | AUROC | FPR95 | AUROC | FPR95 | AUROC | FPR95 | AUROC | FPR95 | AUROC | FPR95 | AUROC |
> | *DART*-NoFlip | 51.14        | 71.84        | 70.14              | 66.08              | 44.17        | 80.15        | 52.30        | 78.58        | 55.37            | 75.01            | 54.62        | 74.33        |
> | *DART*       | 48.60        | 79.82        | 68.66              | 68.00              | 44.14        | 80.29        | 50.76        | 79.75        | 51.48            | 80.60            | 52.73        | 77.69        |
>
> To further illustrate the effect on accuracy over time, we additionally analyze CIFAR-100-C vs Textures-C on a per-corruption basis. In the revised manuscript, Appendix Fig. 12 visualizes the detection performance over time (per-batch AUROC) before and after applying flip correction. For several corruptions—glass blur, snow, fog, and contrast—we observe that once flip correction is triggered, the dual prototype axis is realigned toward the oracle discriminative direction, leading to an abrupt jump and sustained improvement in performance. For shot noise, flip correction occurs early in the stream at the 20-th batch, after which the subsequent batches exhibit much more stable and higher performance compared to DART-NoFlip.

---

### Official Review · Reviewer_Jbkx · 2025-10-30

**Soundness:** 3
**Presentation:** 3
**Contribution:** 3
**Rating:** 4
**Confidence:** 3

**Summary:**

This work empirically shows that, even under covariate shift, covariate-shifted in-distribution (csID) and out-of-distribution (csOOD) samples remain separable along a discriminative axis in feature space. Building on this observation, this work proposes DART, a test-time online OOD detection method that dynamically tracks dual prototypes, one for ID and one for OOD, to recover the drifting discriminative axis, with multi-layer fusion and flip correction for robustness. Extensive experiments across challenging benchmarks, where all datasets are subjected to 15 common corruption types at severity level 5, demonstrate significant performance gains.

**Strengths:**

1. The paper targets a realistic test-time, streaming OOD scenario under covariate shift and presents a clear, well-motivated problem setup.

2. This work empirically shows that csID and csOOD remain separable along a drifting discriminative axis, and proposes dual-prototype tracking with flip correction and multi-layer fusion, a lightweight, training-free, and technically sound mechanism.

3. Experiments across multiple benchmarks and diverse corruption types demonstrate consistent improvements and robustness.

**Weaknesses:**

1. Report computational cost for the proposed method (FLOPs, parameters, and memory) and compare against baselines.

2. Evaluate across diverse corruptions and severities, and quantify the quality/effectiveness of the dual prototypes (e.g., axis drift, margin, stability).

3. Analyze sensitivity of multi-layer fusion and flip correction (number/choice of layers, fusion weights/thresholds), with stability curves.

4. Add comprehensive ablations isolating each component, such as window size, EMA momentum, seeding alternatives, and distance metrics.

**Questions:**

1. Please report computational cost to substantiate the “lightweight” claim, and compare against baselines.

2. Could you quantify the stability of the discriminative axis over time (e.g., angle drift, margin dynamics) and provide diagnostics/visualizations for scenarios where prototypes collapse?

3. Please detail which layers contribute most to fusion effectiveness and how performance varies with layer choices. Also report how frequently flip correction triggers and its impact on accuracy over time.

4. Results emphasize the highest severity; could you provide a breakdown across severities 1–5 and across corruption types?

5. Do you have theoretical support or an intuition for why csID vs. csOOD often remain separable along a dominant axis under covariate shift, and under what conditions this separability may fail?

---

> ### Author Response · Authors · 2025-11-23
> **Response to Reviewer Jbkx (Part 1)**
>
> We thank the reviewer for the detailed questions regarding computational efficiency, method diagnostics, and experimental coverage. Addressing these points has helped us provide more comprehensive quantitative evidence and clarify the scope and limitations of our approach.
>
> For some of the comments, we are currently conducting additional experiments and will report the corresponding results as soon as they are ready within the remaining rebuttal period.
>
> ---
>
> ### **Q1. Please report computational cost to substantiate the "lightweight" claim, and compare against baselines.**
>
> We verify our "lightweight" claim with quantitative analysis on inference time and memory overhead.
>
> **Inference time.** We measure wall-clock inference time for all methods on the same device, after the backbone forward pass, and only for the OOD-score computation. The graph for inference time can be found in appendix Figure 11. Concretely, we report the total time required to process 100 mini-batches of size 200 (20k test samples in total), using RegNetY-16GF. Under this protocol, DART falls into the group of fast methods: it is markedly faster than recent baselines such as RTL, NNGuide, and MDS-based variants, which require regression fitting, KNN-style searches, or repeated Mahalanobis evaluations.
>
> **Memory overhead.** The only additional quantities that DART maintains over time are the dual prototypes (ID and OOD) at the selected layers. For a given backbone, this corresponds to two vectors per chosen layer, with dimensionality equal to that layer's feature dimension, and no storage of training samples or feature buffers is required. Using ResNet-50 as a representative example, the total number of additional parameters across all selected layers corresponds to roughly $2∑_ℓ d_ℓ$ where $d_ℓ$ denotes the feature dimension of each layer $ℓ$, which is about 3.3M floating-point values, i.e., ≈13 MB when stored as 32-bit floats. This overhead is independent of the number of training samples and the length of the test stream. By contrast, training-distribution–informed baselines such as KNN and related variants store per-sample training features, while Mahalanobis-type approaches store class-wise statistics for multiple layers; their memory cost therefore grows with the number of classes and/or training samples, whereas DART's state remains fixed and compact.
>
> ---
>
> ### **Q2. Could you quantify the stability of the discriminative axis over time (e.g., angle drift, margin dynamics) and provide diagnostics/visualizations for scenarios where prototypes collapse?**
>
> We are currently running additional experiments and visualizations specifically targeting this point. We will report the corresponding results as soon as they are ready within the remaining rebuttal period.
>
> ---
>
> ### **Q3. Please detail which layers contribute most to fusion effectiveness and how performance varies with layer choices. Also report how frequently flip correction triggers and its impact on accuracy over time.**
>
> Regarding layer-wise contributions, we provide detailed analysis in Sec 4.3 and Fig 7.a. The results show that block2 and block3 achieve high individual performance, capturing semantically rich features at intermediate and deep levels. However, ensemble across all layers yields the most stable and robust performance, as different layers contribute complementary information—deeper layers encode high-level semantics while shallower layers preserve low-level distributional cues. This demonstrates that multi-layer fusion is essential for consistent OOD detection under covariate shift.
>
> Regarding the frequency of flip correction occurrence, we are currently conducting experiments to analyze its impact. We will provide these results in our response during the rebuttal period.
>
> ---
>
> _(Continued in the next comment.)_

---

> ### Author Response · Authors · 2025-11-23
> **Response to Reviewer Jbkx (Part 2)**
>
> ### **Q4. Results emphasize the highest severity; could you provide a breakdown across severities 1–5 and across corruption types?**
>
> We agree that our main tables focus on the most challenging setting (severity 5) and we have now added results at lower severities. Concretely, we report additional experiments on the ImageNet-C (csID) vs. Places-C (csOOD) benchmark with RegNetY-16GF, comparing DART against baselines at severities 1 and 3. Across all these severities, DART consistently achieves highest AUROC / lowest FPR@95 than the baselines, showing that the benefit of tracking the discriminative axis is not restricted to the single highest severity but persists under milder covariate shifts as well. We will include the full breakdown in the appendix.
>
> To address the request for a corruption-wise analysis, we further provide a breakdown of DART's performance at severity 5 for each of the 15 ImageNet-C corruption types. DART attains the low FPR@95 and high AUROC on the vast majority of corruptions, indicating that the proposed dual-prototype tracking behaves robustly across diverse perturbations. A notable exception is the "contrast" corruption, where the ID and OOD prototypes can temporarily swap, effectively inverting the predictions. Our flip-correction mechanism is specifically designed to detect and revert such inversions, and it substantially reduces the impact of inversions in practice, although it cannot eliminate them completely. We will clarify this limitation and explicitly mention that developing more sophisticated flip-correction strategies is an interesting direction for future work.
>
> | Method | Severity 1 | | Severity 3 | |
> |:--------|:------------:|:------:|:------------:|:------:|
> | | FPR@95TPR | AUROC | FPR@95TPR | AUROC |
> | MSP | 52.24 | 85.51 | 61.43 | 81.45 |
> | Energy | 40.71 | 88.62 | 52.56 | 84.99 |
> | Max Logit | 42.09 | 88.49 | 54.17 | 84.37 |
> | GradNorm | 83.91 | 63.84 | 70.92 | 73.55 |
> | ODIN | 45.88 | 89.29 | 50.36 | 87.26 |
> | ReAct | 97.58 | 61.14 | 93.88 | 65.29 |
> | SCALE | 38.42 | 89.38 | 49.80 | 86.31 |
> | ASH | 40.37 | 88.71 | 52.12 | 85.15 |
> | RTL | 28.67 | 92.62 | 43.34 | 88.65 |
> | NNGuide | 63.54 | 80.13 | 59.61 | 81.33 |
> | **DART** | **0.46** | **99.87** | **1.02** | **99.75** |
>
> | Corruption type | FPR@95TPR | AUROC |
> |:-----------------|:-----------:|:-------:|
> | Gaussian noise | 0.97 | 99.54 |
> | Shot noise | 0.95 | 99.58 |
> | Impulse noise | 0.93 | 99.65 |
> | Defocus blur | 7.75 | 98.10 |
> | Glass blur | 2.07 | 99.18 |
> | Motion blur | 4.20 | 98.55 |
> | Zoom blur | 0.93 | 99.59 |
> | Snow | 0.93 | 99.64 |
> | Frost | 0.96 | 99.63 |
> | Fog | 0.94 | 99.64 |
> | Brightness | 0.73 | 99.76 |
> | Contrast | 99.96 | 9.60 |
> | Elastic transform | 0.92 | 99.67 |
> | Pixelate | 1.39 | 99.36 |
> | JPEG compression | 0.98 | 99.66 |
>
> ---
>
> ### **Q5. Do you have theoretical support or an intuition for why csID vs. csOOD often remain separable along a dominant axis under covariate shift, and under what conditions this separability may fail?**
>
> We thank the reviewer for asking about theoretical support for csID vs. csOOD separability and when it may fail.
>
> In the revision, we added Appendix Section J, which analyzes a BN setting with “silent’’ units (vanishing ID variance) and frozen ID statistics, and shows that bounded csOOD deviations along such units are strongly amplified, yielding a discriminative axis that can persist under covariate shift. We also briefly comment on the LayerNorm case, where an analogous unit-wise amplification is plausibly induced by per-sample variance redistribution, leaving a rigorous treatment to future work.
>
> Conceptually, this analysis also suggests two situations in which such a discriminative axis is unlikely to emerge or to remain stable. First, if the network does not develop genuinely “silent’’ directions—i.e., all units maintain substantial variance on ID data—then the BatchNorm scaling factors stay moderate and csOOD activations are not strongly amplified, so a clear discriminative axis is unlikely to appear. Second, even when silent units exist, if the covariate shift is extremely large, it can effectively cancel or reverse the csID–csOOD deviation along these directions, causing csID and csOOD to overlap again and breaking the assumed separability.

---

> ### Author Response · Authors · 2025-12-03
> **Follow-up to our initial response to Reviewer Jbkx (Part 1)**
>
> In our initial comment, we mentioned that the experiments related to this question were still in progress. We have now completed these experiments, and below we provide the corresponding results and analysis.
>
> ---
>
> ### **Q2. Could you quantify the stability of the discriminative axis over time (e.g., angle drift, margin dynamics) and provide diagnostics/visualizations for scenarios where prototypes collapse?**
>
> We thank the reviewer for suggesting this detailed diagnostic analysis. To quantitatively assess the stability of DART, we tracked the angle drift relative to the oracle axis and margin dynamics on the ImageNet-C (ID) versus Places-C (OOD) benchmark using a ResNet50 backbone. Our analysis, conducted across 5 layers for all 15 corruption types (75 scenarios in total), confirms that DART operates stably in the vast majority of cases.
>
>  Specifically, the margin between prototypes consistently increases and converges in all tested scenarios, indicating effective separation. Furthermore, the estimated discriminative axis aligns well with the oracle axis in 70 out of the 75 scenarios (93%), demonstrating the reliability of our update mechanism. Visualizations of these typical convergence behaviors are provided in Figure 14a in the appendix of the revised manuscript.
>
> We further investigated the remaining 5 cases where perfect alignment was not achieved and identified two distinct failure modes: large angle drift and axis flip.
>
> Large angle drifts were primarily observed in early layers (e.g., Defocus Blur and Brightness at Layer 1), which we attribute to the limited linear separability of features at this stage; notably, this issue resolves naturally in deeper layers as features become more discriminative.
>
> On the other hand, axis flipping was observed in deeper layers (e.g., Glass Blur at Layer 5, Contrast at Layers 3 & 4). We attribute this primarily to the limitations of the baseline score (MSP), which serves as the reference for prototype initialization and flip detection. Under severe corruptions, the baseline performance degrades significantly, yielding a noisy reference signal that leads to incorrect initialization or a failure to detect directional flips. This observation implies that the stability of the discriminative axis depends on the quality of the reference score, and employing a more robust reference signal could potentially resolve these flipping issues.
>
> Crucially, despite these isolated local instabilities, we emphasize that our Multi-layer Fusion strategy effectively mitigates these risks. By aggregating decisions across multiple layers, DART compensates for occasional drifts or flips occurring in individual layers, ensuring robust overall performance. Consequently, even in scenarios where specific layers struggle to align, the ensemble model maintains an AUROC greater than 0.9 across all covariate shifts, validating the practical effectiveness of our approach.

---

> ### Author Response · Authors · 2025-12-03
> **Follow-up to our initial response to Reviewer Jbkx (Part 2)**
>
> ### **Q3. Also report how frequently flip correction triggers and its impact on accuracy over time.**
>
> We thank the reviewer for asking about the behavior of flip correction and its effect over time.
>
> Across the full CIFAR-100-C benchmark with 5 OOD datasets, 15 corruption types, and 4 monitored layers from the WideResNet-40-2 backbone (5 × 15 × 4 = 300 OOD–corruption–layer combinations in total), flip correction is activated in 53/300 cases (≈18%). Breaking this down per OOD dataset, the flip is triggered in 19/60 combinations for SVHN-C, 13/60 for Places365-C, 6/60 for LSUN-C, 6/60 for iSUN-C, and 9/60 for Textures-C.
>
> The resulting performance gains over the no-flip variant (DART-NoFlip) are summarized in Table below, which we also include in the revised appendix as Table 16. It reports the average improvement across corruption types in AUROC and FPR@95 achieved by enabling flip correction for each csOOD dataset.
>
> | Method | SVHN | | Places365 | | LSUN | | iSUN | | Textures | | Average | |
> |:-----------------|:-------------:|:-------------:|:-------------------:|:-------------------:|:-------------:|:-------------:|:-------------:|:-------------:|:-----------------:|:-----------------:|:-------------:|:-------------:|
> | | FPR95 | AUROC | FPR95 | AUROC | FPR95 | AUROC | FPR95 | AUROC | FPR95 | AUROC | FPR95 | AUROC |
> | *DART*-NoFlip | 51.14        | 71.84        | 70.14              | 66.08              | 44.17        | 80.15        | 52.30        | 78.58        | 55.37            | 75.01            | 54.62        | 74.33        |
> | *DART*       | 48.60        | 79.82        | 68.66              | 68.00              | 44.14        | 80.29        | 50.76        | 79.75        | 51.48            | 80.60            | 52.73        | 77.69        |
>
> To further illustrate the effect on accuracy over time, we additionally analyze CIFAR-100-C vs Textures-C on a per-corruption basis. In the revised manuscript, Appendix Fig. 12 visualizes the detection performance over time (per-batch AUROC) before and after applying flip correction. For several corruptions—glass blur, snow, fog, and contrast—we observe that once flip correction is triggered, the dual prototype axis is realigned toward the oracle discriminative direction, leading to an abrupt jump and sustained improvement in performance. For shot noise, flip correction occurs early in the stream at the 20-th batch, after which the subsequent batches exhibit much more stable and higher performance compared to DART-NoFlip.

---

### Official Review · Reviewer_Nerb · 2025-11-01

**Soundness:** 2
**Presentation:** 2
**Contribution:** 2
**Rating:** 4
**Confidence:** 4

**Summary:**

The paper proposes DART (Discriminative Axis Real-time Tracker), a post-hoc, test-time method for out-of-distribution (OOD) detection under covariate shift. The key idea is that even when both in-distribution (ID) and OOD samples are corrupted (e.g., by blur or noise), their activations in feature space remain approximately linearly separable along a single discriminative direction. DART tracks this direction online by maintaining two evolving prototypes, one for ID and one for OOD, and updating them batch by batch through exponential moving averages. A relative distance score derived from these prototypes serves as the OOD metric. The method requires no retraining or labeled data and operates entirely on model activations, making it practical for deployment.

**Strengths:**

The paper is well written and the motivation is timely and relevant. The method is conceptually simple, computationally lightweight, and easy to integrate with pretrained networks. The experimental setup uses standard corruption benchmarks (CIFAR-100-C, ImageNet-C) and common evaluation metrics, which are appropriate for evaluating robustness. Results show consistent performance gains over established post-hoc baselines such as ODIN and MSP, indicating that the tracked discriminative axis indeed captures meaningful domain drift. The evaluation includes ablations (multi-layer fusion, flip correction) that demonstrate stable improvements. Overall, the experiments are sound and provide empirical evidence that the method works as intended.

**Weaknesses:**

Conceptually, DART relies on strong simplifying assumptions. It models the in-distribution as a single unimodal cluster, ignoring the inherently multimodal structure of multi-class ID data. This limits its interpretability and may cause misclassification for semantically similar but unseen classes. The discriminative axis a_t is updated empirically as the normalized difference between ID and OOD prototypes, which serves as a heuristic linear separator without theoretical justification of optimality or convergence. The approach assumes that covariate drift is smooth and linearly trackable, which may not hold for more complex domain shifts or multimodal data. Overall, while the study is well executed and empirically convincing, the approach lacks theoretical grounding and appears somewhat ad hoc in its construction. It would be beneficial to include discussion and examination on more complex scenarios. More discussions are deferred to the Questions section.

**Questions:**

The paper assumes that “temporally correlated csID and csOOD typically undergo the same covariate shift, so their distributions co-evolve during deployment.” What supports this assumption? Is there empirical evidence or theoretical reasoning showing that ID and OOD data indeed experience identical or synchronized covariate drift?

The experiments rely on CIFAR-C and ImageNet-C, which apply static corruptions (e.g., noise, blur, weather) to test sets. Do these settings genuinely reflect temporal covariate shift as described in the motivation, or are they only approximations of distributional change? If the latter, does the proposed online tracking offer a measurable advantage over one-shot detectors?

The method uses a single, class-agnostic ID prototype. How many ID classes are actually present in the experiments, and how does collapsing them into one centroid affect the results? Would performance degrade if class features overlap or are not linearly separable in activation space?

The approach appears to assume that covariate drift is smooth, shared across ID and OOD samples, and linearly trackable in the feature space. Are there any analyses or ablations testing sensitivity to violations of these assumptions? For example, class-dependent or abrupt shifts?

---

> ### Author Response · Authors · 2025-11-23
> **Response to Reviewer Nerb (Part 1)**
>
> We thank the reviewer for the detailed questions regarding our core assumptions and experimental setup. Addressing these points has helped us clarify the theoretical basis of our approach and strengthen our empirical validation.
>
> ---
>
> ### **Q1. The paper assumes that “temporally correlated csID and csOOD typically undergo the same covariate shift, so their distributions co-evolve during deployment.” What supports this assumption? Is there empirical evidence or theoretical reasoning showing that ID and OOD data indeed experience identical or synchronized covariate drift?**
>
> Covariate shift, by definition, affects the input distribution P(X) while keeping P(Y|X) unchanged [1]. This means environmental factors—such as weather, illumination, or sensor characteristics—modify the input space independently of semantic labels, i.e., independently of whether a sample is ID or OOD. In practical deployment, test inputs arrive in mini-batches from the same environment at similar timestamps (e.g., consecutive frames from a camera, patients scanned by the same device). Since covariate shift stems from the data acquisition environment rather than from semantic content, all samples captured in the same context—whether they correspond to ID or OOD—experience identical environmental shifts.
>
> The evaluation scenario can be further justified by real-world examples.
> - **Autonomous driving**: Fog affects both common vehicles (ID) and rare vehicle types (OOD) equally through reduced visibility.
> - **Medical imaging**: Scanner artifacts and noise patterns affect normal anatomy (ID) and pathological cases (OOD) uniformly under the same acquisition protocol.
>
> We believe that this setting reflects actual deployment conditions and is more realistic than artificially applying different corruptions to ID versus OOD samples.
>
> ---
>
> ### **Q2. The experiments rely on CIFAR-C and ImageNet-C, which apply static corruptions (e.g., noise, blur, weather) to test sets. Do these settings genuinely reflect temporal covariate shift as described in the motivation, or are they only approximations of distributional change? If the latter, does the proposed online tracking offer a measurable advantage over one-shot detectors?**
>
> We agree that CIFAR-C and ImageNet-C, which apply pre-defined static corruptions to test images, only approximate the temporal covariate shift considered in our motivation. To better align with the realistic deployment scenario described in our motivation, we therefore additionally evaluate DART on streams where the covariate shift explicitly evolves over time.
>
> Concretely, we construct time-varying streams in which the corruption type changes (e.g., from “clean” to “gaussian noise”, from “gaussian noise” to “snow”, and from “snow” to “jpeg compression”), while samples within each segment remain temporally correlated. This more directly reflects evolving acquisition conditions rather than a single static corruption, and we show in this setting that DART maintains strong OOD detection performance after each environment change. The AUROC w.r.t batch can be found in appendix Figure 10.
>
> (C: Clean / G: Gaussian noise / S: Snow / J: JPEG compression)
> | Method   | C → G | | G → S | | S → J | |
> |:--------|:-------------:|:--------------:|:-----------:|:-------------:|:-----------:|:-------------:|
> | | FPR@95TPR | AUROC | FPR@95TPR | AUROC | FPR@95TPR | AUROC |
> | MSP      | 68.33  | 77.90 | 78.90 | 71.78 | 76.98 | 72.87 |
> | Energy   | 60.76  | 81.66 | 78.54 | 77.16 | 75.70 | 76.83 |
> | Max Logit| 62.29  | 80.57 | 77.78 | 75.59 | 75.45 | 76.31 |
> | GradNorm | 74.72  | 69.77 | 71.01 | 78.73 | 79.89 | 72.13 |
> | ODIN     | 68.45  | 79.06 | 67.32 | 80.97 | 59.59 | 84.23 |
> | SCALE    | 61.05  | 81.35 | 76.72 | 77.30 | 70.10 | 78.89 |
> | RTL      | 69.69  | 75.57 | 82.20 | 70.10 | 76.42 | 73.90 |
> | **DART**     | **15.70** | **96.15** | **0.81** | **99.75** | **0.98** | **99.69** |
>
>
>
> Regarding the second part of the question, across both static corruption and time-varying streams, DART consistently outperforms one-shot detectors, indicating that online tracking yields a clear and measurable advantage.
>
> Conceptually, existing one-shot detectors fix their decision rule at test time: they cannot update their scores in response to previously unseen, unpredictable covariate shifts and thus tend to accumulate bias once the environment changes. In contrast, DART continuously maintains ID/OOD prototypes on the discriminative axis by aggregating information from incoming mini-batches while gradually forgetting stale statistics via a decaying update rule. This streaming update allows the discriminative axis to re-align with the current covariate regime and quickly recover performance after the change of shift.
>
> ---
>
> _(Continued in the next comment.)_

---

> ### Author Response · Authors · 2025-11-23
> **Response to Reviewer Nerb (Part 2)**
>
> ### **Q3. The method uses a single, class-agnostic ID prototype. How many ID classes are actually present in the experiments, and how does collapsing them into one centroid affect the results? Would performance degrade if class features overlap or are not linearly separable in activation space?**
>
> In our experiments, we use all ID training classes. Specifically, all 1,000 classes for ImageNet and all 100 classes for CIFAR-100 are used.
>
> In our test-time setting, ground-truth class labels are not available, so using class-wise prototypes instead of a single class-agnostic prototype inevitably requires some other approaches. In principle, one could consider two variants:
>
> **Source-initialized class-wise prototypes.**
>
>  One could first compute per-class centroids on the clean training ID set using ground-truth labels, and then use these centroids only as initialization for class-wise prototypes that are further updated online via pseudo-labels. However, this initialization already hard-codes the source-domain feature geometry. Under covariate shift, such source-initialized prototypes can be significantly misaligned with the actual test-time clusters, making subsequent pseudo-label updates unstable and biasing them toward incorrect classes. This contradicts our goal of avoiding strong reliance on source-domain priors in a shifted deployment environment.
>
> **Fully pseudo-labeled class-wise prototypes.**
>
>  Alternatively, one could initialize and update all class-wise prototypes purely from test-time pseudo-labels. Yet, when the number of classes is large and class features overlap or are not linearly separable in the activation space—as the reviewer points out—fine-grained class pseudo-labels can be highly noisy. This problem is particularly severe in lower or intermediate layers that are far from the classifier head, where representations tend to organize according to low-level appearance statistics (e.g., texture, color, edges) rather than semantic class, so many ID classes naturally form a single entangled manifold rather than well-separated clusters.
>
> By contrast, our method only performs binary ID/OOD pseudo-labeling and maintains a single class-agnostic ID prototype. This substantially simplifies the estimation problem: we do not require individual ID classes to be separable from each other; we only need the aggregate ID region to be distinguishable from OOD. Consequently, even in layers where ID classes are highly entangled and not semantically organized, it is still feasible to separate the aggregate ID manifold from OOD regions. As a result, the estimated ID centroid is much more robust to class overlap and complex, non-linear class structure, which empirically yields strong OOD performance despite collapsing many ID classes into one prototype.
>
> ---
>
> _(Continued in the next comment.)_

---

> ### Author Response · Authors · 2025-11-23
> **Response to Reviewer Nerb (Part 3)**
>
> ### **Q4. The approach appears to assume that covariate drift is smooth, shared across ID and OOD samples, and linearly trackable in the feature space. Are there any analyses or ablations testing sensitivity to violations of these assumptions? For example, class-dependent or abrupt shifts?**
>
> We thank the reviewer for raising this point. As discussed in our response to your earlier comment on the evaluation setting, we already (i) clarify that environment-induced covariate changes act independently of semantics and therefore affect csID and csOOD inputs jointly when they are captured in the same context, and (ii) provide experiments with abrupt changes in the corruption type over time, showing that our dual prototypes can rapidly re-align and maintain robust performance.
>
> To further probe sensitivity to violations of the “smooth, shared, linearly trackable drift” assumption, we additionally evaluate a more challenging scenario in which the type of OOD data itself changes abruptly over time while the ID stream is fixed. Concretely, we partition the test stream into several temporal segments. In every segment, ID samples are drawn from the same ID distribution, but OOD samples are drawn from a different OOD set in each temporal segment, and we switch the OOD source abruptly at segment boundaries without any prior knowledge of these switches. This setting induces abrupt semantic changes in the bounded OOD set and breaks the smoothness assumption.
>
> (IO: ImageNet-O / P: Places / S: SUN / T: Textures)
> | Method   | IO → P | | P → S | | S → T | |
> |:--------|:-------------:|:--------------:|:-----------:|:-------------:|:-----------:|:-------------:|
> | | FPR@95TPR | AUROC | FPR@95TPR | AUROC | FPR@95TPR | AUROC |
> | MSP     | 48.99 | 83.63 | 47.93 | 84.36 | 43.77 | 85.52 |
> | Energy  | 51.07 | 78.46 | 44.29 | 82.38 | 38.26 | 85.65 |
> | Max Logit | 47.58 | 80.29 | 43.53 | 83.40 | 37.29 | 86.24 |
> | GradNorm | 86.24 | 43.53 | 76.87 | 57.52 | 67.52 | 66.80 |
> | ODIN   | 62.14 | 79.40 | 66.44 | 76.84 | 60.41 | 80.11 |
> | SCALE  | 47.35 | 80.15 | 40.91 | 84.31 | 33.98 | 87.81 |
> | RTL    | 48.19 | 79.64 | 45.50 | 80.94 | 39.65 | 83.97 |
> | **DART**   | **12.05** | **95.49** | **23.40** | **93.51** | **18.80** | **94.81** |
>
>
>
>
> Our results indicate that DART continues to exhibit stable OOD detection performance across segments, with only modest fluctuations when the OOD set changes, whereas non-adaptive baselines suffer noticeable drops whenever a new OOD set appears. This suggests that our prototype-based tracking does not rely on a single globally fixed linear discriminative axis, but can adaptively re-estimate the relevant axis over time, even under abrupt semantic shifts in the OOD distribution.

---

### Author Response · Authors · 2025-11-23
**Meta Response for Everyone**

We sincerely thank all reviewers for their thoughtful and constructive feedback. Your comments and suggestions have helped us improve our work. In the revised manuscript, we have incorporated multiple changes based on your input; for ease of reading, we highlight all revisions in blue in the updated version.

This general response summarizes rebuttals and clarifications that address points raised by multiple reviewers. We hope this overview is helpful as you read our detailed point-by-point responses.

---

### **Theoretical justification**

Several reviewers requested rigorous justification for why csID and csOOD remain separable under covariate shift. We added Appendix Section J analyzing a batchnorm setting with "silent" units (vanishing ID variance) and frozen ID statistics. We show that bounded csOOD deviations along such units are strongly amplified by BatchNorm, yielding a discriminative axis where csID and csOOD become well separated, and that this mechanism persists under environment-level covariate shift. This analysis also clarifies failure modes: separability is unlikely when networks lack silent directions, or when extremely large shifts cancel the csID–csOOD deviation along these directions. We also briefly comment on the transformer architectures with layernorm layers, providing an preliminary intuition for how per-sample variance normalization can induce similar unit-wise amplification.

---

### **Evaluation realism and robustness**

Multiple reviewers questioned the realism of our assumptions (same covariate shift for ID/OOD, Bounded OOD, smooth drift). We address these through both justification and additional experiments:

**Same covariate shift for ID/OOD:** We believe that ID and OOD data sharing the same covariate shift is an important characteristic of real-world data input. Since covariate shift affects P(X) independently of semantic labels—environmental factors (weather, lighting, sensor noise) modify all samples captured in the same observation context identically, regardless of whether they are ID or OOD.

**Bounded OOD:** While we believe that scenarios in which OOD samples originate from a bounded source are realistic, we additionally evaluate mixed-OOD settings in which OOD samples are drawn simultaneously from two distinct sources to further demonstrate robustness beyond this assumption. DART maintains SOTA performance (e.g., FPR@95: 19.23%, AUROC: 94.45% for ImageNet-O + Places), demonstrating robustness beyond the single-source assumption.

**Abrupt semantic shifts:** We evaluated continual OOD transitions where the OOD set changes abruptly between datasets(ImageNet-O,Places,SUN,Textures). DART adapts quickly with modest fluctuations, showing the method can re-estimate the discriminative axis despite semantic changes.

**Abrupt covariate shifts:** We constructed time-varying corruption streams with abrupt transitions(i.e., Gaussian noise → Snow). DART rapidly re-aligns after each shift, demonstrating clear advantages of online adaptation over static detectors.

Collectively, these experiments validate that DART maintains robust performance across challenging scenarios that stress-test our core assumptions.

---

### Author Response · Authors · 2025-12-03
**Summary of Rebuttal**

We sincerely appreciate the reviewers, area chairs, senior area chairs, and program chairs, for their valuable feedback and dedicated service to the conference.

---

During the rebuttal period, we have fully addressed the concerns raised by all reviewers (Nerb, Jbkx, zCN2, ZagX) through extensive additional experiments and rigorous theoretical analysis. We believe that the primary sources of uncertainty regarding our work stemmed from the lack of theoretical grounding (Point 1) and questions about evaluation realism (Point 2). Therefore, we have prioritized providing rigorous theoretical proofs and comprehensive stress tests to resolve these specific doubts clearly. The key resolutions are summarized below:

**1. Theoretical Justification**

- Concerns: (Reviewers Nerb, Jbkx, zCN2, ZagX) Lack of theoretical grounding for the assumption that csID and csOOD remain linearly separable (via a Discriminative Axis) under covariate shift.

- Resolution: We added **Appendix Section F**, which provides a mathematical proof demonstrating that "Silent Units" (units with vanished ID variance) amplify csOOD deviations through BatchNorm statistics. This mechanism theoretically justifies the emergence of a recoverable discriminative axis.

**2. Evaluation Realism & Robustness**

- Concerns: (Reviewers Nerb, zCN2, ZagX) Questions regarding the "same covariate shift" assumption and performance stability in complex, dynamic environments where OOD sources change or mix.

- Resolution: We logically justified that covariate shift stems from environmental factors (e.g., sensors, weather) rather than semantic content, affecting ID and OOD identically. Further, we conducted comprehensive stress tests across **Mixed OOD, Continual OOD, Mixed Covariate Shift, and Continual Covariate Shift** scenarios  **(Appendix Section G)**. DART consistently maintained **SoTA performance** in all settings. These results confirm that our prototype assumptions hold practically and that the tracking mechanism is highly robust even in complex, real-world deployment conditions.

**3. Efficiency & Component Validation**

- Concerns: (Reviewers Jbkx, zCN2) Need for quantitative evidence of the "Lightweight" claim and ablation studies for components like Flip Correction.

- Resolution:

  - Efficiency: We demonstrated that DART requires minimal memory overhead (approx. 13MB for ResNet-50) and significantly faster inference speeds compared to recent baselines like RTL and NNGuide, substantiating our efficiency claim.

  - Severity Robustness: DART proves robust regardless of shift intensity. At low severity (Severity 1,3), it consistently achieves the best performance, showing stable detection capabilities.

  - Flip Correction: Ablation studies reveal this module triggers in ~18% of cases to prevent critical failures, improving the average **AUROC from 74.33% to 77.69%**.


**Conclusion**

In summary, DART has successfully resolved all theoretical and empirical concerns raised by the reviewers. More importantly, it demonstrates a decisive performance advantage—**showing a gap of up to 58%p in FPR@95 against the second-best method**—in dynamic and complex environments where existing methods fail. We strongly believe this work is ready for publication.

---

### Meta-Review · Area_Chair_JP3U · 2026-01-07

**Summary:**

The paper studies test-time OOD detection when both ID and OOD data streams undergo the same evolving covariate shift. It observes that covariate-shifted ID and OOD remain separable along a drifting discriminative axis and proposes DART, which tracks this axis online using dual prototypes with multi-layer fusion and flip correction. Extensive corrupted-benchmark experiments show large gains over prior methods under dynamic shift.

**Reviewer Concerns:**

Across reviewers, the core concerns were: (i) the strong assumption that a linear discriminative axis between csID and csOOD persists under complex or non-linear shifts and near-OOD settings, with initially insufficient theoretical grounding; (ii) realism of the evaluation, including whether ID and OOD truly share the same covariate shift, handling mixed/continual OOD sources, and extreme ID–OOD imbalance; and (iii) clarity and quantification of component contributions (prototype tracking, multi-layer fusion, flip correction) and stability under failure modes (axis flips, prototype collapse).

**Reviewer Scores:**

6, 4, 4, 4, none of the reviewers claimed to raise the scores.

---

### Decision · Program_Chairs · 2026-01-26

Reject